# Calibration of a Water Vapour Raman Lidar using GRUAN-certified Radiosondes and a New Trajectory Method

Shannon Hicks-Jalali[1], Robert J. Sica[1,2], Alexander Haefele[2,1], and Giovanni Martucci[2]

[1]Department of Physics and Astronomy, The University of Western Ontario, London, Canada
[2]Federal Office of Meteorology and Climatology MeteoSwiss, Payerne, Switzerland

*Correspondence to:* Shannon Hicks-Jalali, hicks.shannonk@gmail.com

**Abstract.**

Raman lidars have been designated as potential candidates for trend studies by NDACC and GRUAN; however, for such studies improved calibration techniques are needed as well as careful consideration of the calibration uncertainties. Trend determinations require frequent, accurate, and well-characterized measurements. However, water vapour Raman lidars produce a relative measurement and require calibration in order to transform the measurement into mixing ratio, a conserved quantity when no sources or sinks for water vapour are present. Typically, the calibration is done using a reference instrument such as a radiosonde. We present an improved trajectory technique to calibrate water vapour Raman lidars based on the previous work of Whiteman et al. (2006), Leblanc and Mcdermid (2008), Adam et al. (2010), and Herold et al. (2011), who used radiosondes as an external calibration source, and matched the lidar measurements to the corresponding radiosonde measurement. However, they did not consider the movement of the radiosonde relative to the air mass and fronts. Our trajectory method is a general technique which may be used for any lidar, and only requires that the radiosonde report wind speed and direction. As calibrations can be affected by a lack of co-location with the reference instrument, we have attempted to improve their technique by tracking the air parcels measured by the radiosonde relative to the field-of-view of the lidar. This study uses GCOS Reference Upper Air Network (GRUAN) Vaisala RS92 radiosonde measurements and lidar measurements taken by the MeteoSwiss RAman Lidar for Meteorological Observation (RALMO), located in Payerne, Switzerland from 2011-2016 to demonstrate this improved calibration technique. We compare this technique to the traditional radiosonde-lidar calibration technique which does not involve tracking the radiosonde and use the same integration time for all altitudes. Both traditional and our trajectory methods produce similar profiles when the water vapour field is homogeneous over the 30 min calibration period. We show that the trajectory method reduces differences between the radiosonde and lidar by an average of 10% when the water vapour field is not homogeneous over a 30 min calibration period. We also calculate a calibration uncertainty budget that can be performed on a nightly basis. The calibration uncertainty budget includes the uncertainties due to phototube paralysis, aerosol extinctions, the assumption of the Ångstrom exponent, and the radiosonde. The study showed that the radiosonde was the major source of uncertainty in the calibration at 4% of the calibration value. This trajectory method showed small improvements for RALMO's calibration, but would be more useful in for stations in different climatological regions, or when non-co-located radiosondes are the only available calibration source.

## 1 Introduction

Water vapour is one of the main contributors to the greenhouse effect due to its ability to absorb infrared radiation efficiently. Water vapour has high temporal and spatial variability, making it difficult to characterize its influence on the atmosphere (Ross and Elliott, 1996; Trenberth et al., 2005; Kämpfer, 2013). When conducting climatological studies, ground-based lidars have an advantage over satellite-borne instruments in that they have the ability to provide frequent measurements from the same location. Lidar measurements are particularly useful for creating statistically significant water vapour trends throughout the troposphere, as they are able to make long term and frequent measurements (Whiteman et al., 2011b). Minimizing the uncertainty in the measurements is critical in order to establish a valid trend. A large component of a lidar measurement's uncertainty budget is its calibration constant. Water vapour lidars measure relative profiles, and therefore require a calibration to convert the measurements into physical units (here mixing ratio). Refining the calibration process is critical to detect the small changes anticipated in the trend analysis. Several Raman lidar calibration techniques have been developed over the years, including internal, external, and a hybrid of internal and external methods.

Internal calibration techniques require no external reference instrument. They can account for the entire optical path in the lidar system to find the water vapour calibration constant. In essence, all optical transmittance, quantum efficiencies of the detectors, Raman cross-sections, the geometric overlap, and their associated uncertainties must be quantified and accounted for. Some of these can be derived simultaneously using the white light calibration discussed in Leblanc and Mcdermid (2008). The white light technique is advantageous in that it can accurately track changes in the calibration constant. However, the calibration is incapable of detecting shifts in spectral separation units, and is not able to accurately detect the cause of calibration changes unless multiple lamps in different locations are used (Whiteman et al., 2011a). Venable et al. (2011) improved the technique by using a scanning lamp instead of a stationary lamp. The limiting factor in the white lamp calibration technique is the degree to which we know the molecular cross-sections, which have uncertainties on the order of 5% (Avila et al., 2004; Venable et al., 2011). While internal calibration offers many advantages, it is impractical for many systems, such as lidars that use multiple mirrors (Dinoev et al., 2013; Godin-Beekmann et al., 2003) or large-aperture mirrors such as the rotating liquid mercury mirror of The University of Western Ontario's Purple Crow Lidar (Sica et al., 1995).

The standard external method involves comparing the lidar and a reference instrument; typically the reference instrument is a radiosonde (Melfi, 1972; Whiteman et al., 1992; Ferrare et al., 1995) but microwave radiometers or GNSS satellites may also be used (Han et al., 1994; Hogg et al., 1983; Foth et al., 2015; David et al., 2017). External calibrations are often

preferable because there is no need to characterize every system component and the uncertainties in the Raman cross sections do not contribute. However, the accuracy of the external calibration is dependent on the accuracy of the reference instrument. Radiosondes are widely used calibration instruments, as they have high spatial resolution, are routinely available, and widely available. Uncertainties for the Vaisala RS92 relative humidity measurements vary between 5 to 15% depending on the time of day (Miloshevich et al., 2009; Dirksen et al., 2014). To minimize the calibration uncertainties induced by biases in the radiosonde reference, the GCOS (Global Climate Observing System) Reference Upper-Air Network (GRUAN) has established a robust correction algorithm for the Vaisala RS92 radiosondes, as RS92 radiosondes are the most frequently used calibration radiosondes (Dirksen et al., 2014). GRUAN RS92 relative humidity profiles have been shown to be 5% more moist than uncorrected RS92 relative humidity profiles, while reducing the relative humidity uncertainties by up to 2% (Dirksen et al., 2014).

Hybrid internal-external methods, which also attempt to minimize variations in the sampled air mass due to the balloon's horizontal motion, have also been implemented by Leblanc and Mcdermid (2008) and Whiteman et al. (2011a). In these hybrid techniques, the white light calibration lamp is used to monitor the efficiency of the lidar optical paths, but is supplemented with radiosondes for the absolute calibration value. The hybrid technique will monitor relative changes in the calibration constant, but must be supplemented periodically with an external calibration (Leblanc and Mcdermid, 2008).

For any external calibration where the lidar and the calibration instrument do not share a common field-of-view, variations in water vapour cause an additional uncertainty in the calibration that is often not quantified in the uncertainty budget. A portion of the calibration uncertainty when using radiosondes can occur from the radiosonde's lack of co-location with the lidar, hereafter the "representation" uncertainty. This paper attempts to resolve the co-location problem and minimize the representation uncertainty by using a tracking technique that expands upon those discussed in Whiteman et al. (2006), Leblanc et al. (2012), Adam et al. (2010). The co-location problem can be particularly acute for calibration via a radiosonde, as the radiosonde takes approximately 30 min to reach the tropopause at mid-latitudes, during which time the radiosondes in this study traveled a minimum of 4 km from the lidar's field-of-view (assumed here to be the zenith, which is typically how water vapour lidars are operated). The distance traveled by the radiosonde has little effect on a calibration measurement if the air mass being sampled is horizontally homogeneous. However, this is not necessarily the case, and when we calibrate while on the edge of an airmass, or the air mass simply is not horizontally uniform, then the water vapour field may change dramatically over the distances the radiosonde travels. Lidar stations which have the resources to use daily radiosondes may not see this as much of a hindrance; however, if the station relies on infrequent calibration campaigns then the campaign calibration results are dependent on the air masses which are sampled.

As in any atmospheric calibration method, it is important that the instruments involved measure the same air mass. To improve the coincidence for periods where calibration is required but the atmospheric water vapour content is changing, we have developed an improved lidar-radiosonde calibration technique that utilizes the position of the radiosonde and the wind speed and direction measured by the radiosonde. The wind speed and direction measurements allow us to track the air parcels as measured by the radiosonde with respect to the position relative to the lidar. If the air is within a 3 km radius around the lidar, we use the corresponding times and lidar scans for calibration. A lidar "scan" refers to a 1 minute (1800 shots) raw measurement profile. We have implemented the technique using 76 nighttime GRUAN RS92 radiosonde flights from 2011 to 2016. The GRUAN sondes represent the best characterized sonde measurements available in terms of calibration and uncertainty budget Dirksen et al. (2014). Daytime calibrations were not tested due to the significantly reduced signal-to-noise (SNR) in daylight measurements and the inability to reach above 5 km effectively with the lidar. We will illustrate the method using measurements from the MeteoSwiss RAman Lidar for Meteorological Observing (RALMO) in Payerne, Switzerland (Dinoev et al., 2013; Brocard et al., 2013) on July 22nd, 2017 corresponding to the 00:00 UTC GRUAN RS92 radiosonde launch.

Section 2 will outline the measurements used in the study. Sections 3 and 4 discusses the methodologies of the traditional and trajectory methods. Sections 5 and 6 will compare the new trajectory method with the traditional calibration technique and their respective uncertainties. Sections 7 and 8 will summarize the results and discuss their implications and the next steps forward.

## 2  Calibration Measurements

### 2.1  Radiosonde Measurements

The MeteoSwiss Payerne research station launches Vaisala GRUAN RS92 radiosondes within 100 m of RALMO bi-weekly (every other week). A subset of these radiosondes are processed by GRUAN because not every RS92 flight before 2019 was GRUAN-compliant. GRUAN requires that radiosondes undergo several pre-flight checks and calibrations, which are detailed in Dirksen et al. (2014). These calibrations are needed to correct radiation and systematic relative humidity biases in the radiosonde temperature, pressure, and relative humidity profiles. This study uses the official GRUAN RS92 radiosonde product to minimize and accurately calculate the calibration uncertainty and the contribution from the radiosonde. All radiosonde measurements from 2011 to 2016 taken by the RS92 Vaisala sondes were processed by the GRUAN correction software (Dirksen et al., 2014). Radiosondes prior to October 2011 were RS92 radiosondes, but were not processed by GRUAN because

they were not compatible with the GRUAN requirements listed in Dirksen et al. (2014). All radiosonde measurements were interpolated onto the lidar resolution grid (3.75 m).

The radiosonde water vapour mixing ratios are calculated using the GRUAN-corrected relative humidity profiles and the Hyland and Wexler 1983 formulae for the saturation vapour pressure (Hyland and Wexler, 1983). By convention, the relative humidity measurements are assumed to be over water for all altitudes. A total of 76 GRUAN RS92 nighttime flights were initially used to conduct this analysis, however, due to clouds and lack of coincident lidar measurements, only 24 flights were used for calibration.

## 2.2 Lidar Measurements

Lidar measurements in this study were made using RALMO. RALMO was built at the École Polytechnique Fédérale de Lausanne (EPFL) for operational meteorology, model validation, and climatological studies and is operated at the MeteoSwiss Station in Payerne, Switzerland ($46.81°$ N, $6.94°$ E, 491 m a.s.l.). RALMO was designed to be an operational lidar and therefore was designed to have high accuracy, temporal measurement stability, and minimal altitude-based corrections (Dinoev et al., 2013; Brocard et al., 2013). RALMO operates at 355 nm with a nominal pulse energy of 300 mJ and a repetition frequency of 30 Hz. Measurements are recorded for one minute (1800 laser shots) with a 3.75 m height resolution from both the nitrogen (407 nm) and water vapour (387 nm) Raman scattering channels. RALMO runs day and night with an average of 50% uptime from 2008 - 2017. RALMO downtime is due to the presence of fog, clouds below 800 m, or precipitation (40%) as well as repairs/routine maintenance (10%).

The lidar measurements are processed for calibration in several steps. First, we select $\pm 2$ hours of 1-minute lidar profiles around the launch time of the radiosonde. While two hours was chosen as an arbitrary time range to allow for scan selection, in practice the method rarely selects scans more than 30 min before or after the launch. The 1 min scans are filtered to remove scans with high backgrounds above 0.01 photon counts/bin/s. We assume clouds are present if the nitrogen SNR is less than 1 at 13 km. If a cloud is present, the scan is masked and removed from the calibration. The calibration is conducted at the lidar's native altitude resolution in order to provide as many data points as possible and to avoid smoothing out small features.

## 3 The Traditional Method

The "traditional" method for calibrating water vapour lidars is done by integrating a fixed number of lidar profiles as a function of height starting at a time which is coincident with the radiosonde launch and then calculating a linear weighted least-squares fit between the radiosonde and lidar measurements to determine the calibration constant (Melfi, 1972; Whiteman et al., 1992).

The altitudes over which the fit is conducted are either fixed (e.g. always 1 - 5 km), or the optimal altitude region may be determined by calculating the correlation between the radiosonde and the lidar measurements (Dionisi et al., 2010; Whiteman et al., 2012). For the purposes of this paper, we refer to the traditional method as using 30 min of integration with a weighted least-squares fit over altitudes determined by the correlation coefficient which minimizes the variance of the fit's residuals.

## 3.1 Calculation of the Water Vapour Mixing Ratio for RALMO Measurements

The water vapour mixing ratio ($w$) for RALMO is calculated from the saturation- and background-corrected lidar signals using the water vapour Raman lidar equation (Melfi, 1972; Whiteman et al., 1992; Whiteman, 2003):

$$w(z) = C_w \frac{N_{\mathrm{H_2O}}(z)}{N_{\mathrm{N_2}}(z)} \frac{\Gamma_{\mathrm{N_2}}(z)}{\Gamma_{\mathrm{H_2O}}(z)} \tag{1}$$

where $N_{\mathrm{H_2O,N_2}}(z)$ is the background- and saturation-corrected water vapour and nitrogen photon signals as a function of altitude ($z$) and $\Gamma_{\mathrm{H_2O,N_2}}(z)$ is the total Raman-backscatter transmissions for the water vapour and nitrogen channels, including molecular and particulate scattering. The molecular transmission values are calculated using the GRUAN-corrected temperature and pressure profiles from the corresponding radiosonde and the Rayleigh cross-sections are determined using the formulae from Nicolet (1984).

Whiteman (2003) discussed the necessity of accounting for aerosol transmissions, as the presence of aerosols can create uncertainties in the lidar profiles of up to 4%, depending on the aerosol load. Therefore, to minimize this effect on our calibration constants, we have calculated the aerosol extinctions using the RALMO backscatter ratio product which is calculated by taking the ratio of the elastic backscatter signal to the sum of the pure rotational Raman signals (Whiteman, 2003). Similarly to the method followed in Sica and Haefele (2016), we calculate the extinction profile ($\alpha_{aer}(z)$) using the following equation:

$$\alpha_{aer}(z) = LR(z)(\beta_{mol}(z)(BSR(z) - 1)), \tag{2}$$

where $LR(z)$ is the assumed lidar ratio profile, $\beta_{mol}(z)$ is the molecular backscatter profile taken from the NCEP model, and $BSR(z)$ is the backscatter ratio profile. The lidar ratio profile is a step function with a constant value of 50 in the boundary layer and 20 in the free troposphere. The height of the boundary layer is estimated using the backscatter ratio profile. The assumed lidar ratios are climatological values which have been based on the typical aerosols detected using the co-located Precision

Filter Radiometer (PFR). The aerosol transmissions for water vapour and nitrogen were calculated using the following:

$$\Gamma_{aer,X} = e^{-\tau} = \exp\left[-\left(\frac{\lambda_X}{\lambda_0}\right)^A \int \alpha_{aer}(z)dz\right] \tag{3}$$

Where $\Gamma_{aer,X}(z)$ is the aerosol transmission profile for a given molecule $X$ (e.g. $N_2$ or $H_2O$), and the optical depth is $\tau_X$. The wavelength for a particular channel is $\lambda_X$, while $\lambda_0$ is the reference extinction profile, which is 354.7 nm for the elastic channel. The Ångstrom exponent, $A$, is assumed constant with altitude. The Ångstrom exponent is also measured during the daytime using the co-located PFR. However, it is not calculated daily as it requires stable, cloud-free conditions to get an accurate calculation. Since it is not always available, we fit the sum of a 6 and 12 month sinusoid to the Ångstrom exponent time series over measurements from 01 January 2012 until 31 December 2015, with 2014 removed due to a faulty sensor. The fitted sinusoid was then used as the values for the Ångstrom exponents. The standard deviation of the residuals was $\pm 0.34$ and was used as the uncertainty for the Ångstrom exponents. The uncertainty in the calibration due to our assumptions of the aerosol extinction and the Ångstrom exponent are 0.1 and 0.4% ,respectively, and are included in the uncertainty budget for the calibration constant discussed in Sect. 6.

RALMO uses a polychromator with a bandpass of 0.3 nm (Simeonov et al., 2014). The central wavelengths of RALMO's water vapour and nitrogen channels were chosen to minimize temperature dependence of the Raman cross-section. Dinoev et al. (2013) showed that the nitrogen channel had a relative change in transmitted intensity of 0.4% per 100 K and the water vapour channel intensity changed by roughly 1% when varied between $-60°C$ and $+40°C$.

The calibration constant $C_w$ is defined as:

$$C_w = 0.781 \frac{M_{H_2O}}{M_{Air}} \frac{\eta_{N_2}}{\eta_{H_2O}} \frac{O_{N_2(z)}}{O_{H_2O}(z)} \frac{\sigma_{N_2}(T(z))}{\sigma_{H_2O}(T(z))} \frac{F_{N_2}(T(z))}{F_{H_2O}(T(z))} \tag{4}$$

(Whiteman, 2003).

The calibration constant contains all scaling constants and unknown factors, such as the fraction of nitrogen molecules in air, $0.781$ , the molecular weights of water and dry air ($M_{H_2O,Air}$), the system efficiency of the nitrogen and water vapour channels ($\eta_{N_2,H_2O}$), the overlap function for both channels ($O_{N_2,H_2O}(z)$),the Raman cross-section for each molecular species ($\sigma_{N_2,H_2O}(T(z))$), and the temperature dependency of the Raman cross-section ($F_{N_2,H_2O}(T(z))$). In RALMO's case, the differential overlap is designed to be unity (Dinoev et al., 2013; Simeonov et al., 2014).

## 3.2 Correlated and Weighted Least Squares Fitting of the Lidar Water Vapour Mixing Ratio to the Radiosonde Water Vapour Mixing Ratio

After calculating the uncalibrated mixing ratio profiles from the ratio of the two lidar signals (Section 2.2), we use a correlated and weighted least squares fit to normalize the lidar profile to the radiosonde and find the calibration constant (Dionisi et al., 2010; Whiteman et al., 2012). The radiosonde relative humidity profile is transformed into water vapour volume mixing ratio using the standard WMO conversion of Hyland and Wexler 1983 saturation vapour pressure formulae (World Meteorological Organization (WMO), 2014; Hyland and Wexler, 1983). The calibration range extends from the surface (491 m ASL) to roughly 7 km ASL depending on the profile. The bottom limit is the first lidar altitude bin at the surface, and the final calibration altitude is determined by the SNR and integration limits we impose. We remove scans at all altitudes where the trajectory spent less than 5 min in the lidar region due to their SNR values being less than 2. This cutoff typically results in the calibration region extending to 7 to 8 km altitude. To ensure that the calibration constant is not biased by a vertical displacement of the air parcel between the lidar and the radiosonde volume, we require the resulting uncalibrated lidar and the radiosonde mixing ratio profile to have a correlation coefficient which minimizes the variance of the fit's residuals and must be higher than 0.75. Several fits are made using correlation coefficient thresholds between 0.75 and 0.9 and the fit with the minimum variance in the residuals is chosen for the final calibration constant.

A moving window of 300 m is run over both the radiosonde and lidar profile, and the cross correlation between the two profiles inside each window is determined. To reduce the effect of noise on the cross correlation, both profiles are smoothed beforehand with a boxcar filter of 101.5 m width. In less than one-third of the cases, when the radiosonde leaves the lidar region early, or the wind is such that the air is spending less than 5 min in the lidar region, a large portion of the profile may be cut off.

If the correlation between the radiosonde and lidar mixing ratios within each window is higher than the correlation threshold then that window's altitude range is accepted for calibration (Dionisi et al., 2010; Whiteman et al., 2012). If there is less than 900 m of data available for calibration at the end of the correlation process, then we do not use that night for calibration as it does not have enough data with which to accurately calibrate. This criterion caused 2 out of the 76 nights to be rejected. While the correlation is calculated on the smoothed profiles, the fit is done by using the native resolution of the lidar inside the accepted calibration windows with a requirement of at least 243 points. The least squares fit is conducted over all of the points selected in the cross-correlation procedure. Each fitting point is weighted by the inverse of the sum of the variances of the water vapour mixing ratio percent uncertainty and the average radiosonde mixing ratio uncertainty. The lidar water vapour mixing ratio statistical uncertainty ($\sigma_{w,stat}$) is propagated from the water vapour and nitrogen channel statistical uncertainties using Eq. 1 (Melfi, 1972). The radiosonde mixing ratio percent uncertainties ($\sigma_{MR,Radiosonde}$) are calculated using the total relative

humidity, temperature, and pressure uncertainties reported for each GRUAN flight and propagating through the Hyland and Wexler (1983) equations for saturation vapour pressure to calculate mixing ratio while assuming the relative humidity to be over water (Dirksen et al., 2014; Immler et al., 2010). However, the GRUAN processing occasionally does not report pressure uncertainties below 15 km. Therefore, it was necessary to create a nightly average pressure uncertainty profile which was used on nights where pressure uncertainties were not reported. The variation in the pressure uncertainties was on the order of 0.01%, therefore this assumption is justified. The calibration constant is then determined by using a one-parameter weighted least-squares fit of the form shown in Eq. 1.

## 4   The Radiosonde Trajectory Method

The radiosonde trajectory method begins with the same procedure as the traditional method, where each of the scans is filtered for clouds or abnormally large background levels, as is discussed in Sect. 2.2. However, instead of choosing the first 30 scans after the radiosonde launch, the scans are chosen based on the radiosonde's movement with respect to the air mass and the wind direction measured by the radiosonde.

First, we use the latitude and longitude of the radiosonde, as calculated by the on-board GPS system, as the initial position for air parcel tracking. The air parcel is then tracked backwards from the radiosonde position and is assumed to have traveled in a straight line. We then transform the coordinates onto a Euclidean grid with the lidar located at the origin using the local flat Earth approximation, which is appropriate when distances are shorter than 20 km (Daidzic, 2017; Smart, 1977). We do not consider the vertical movement of the air parcel in this method. Users may need to consider other distance conversion methods, such as the haversine conversion (Smart, 1977), if utilizing radiosondes which are not launched from the same site as the lidar station. Additionally, it may be more appropriate in such cases to use a wind field measurement or model to conduct the trajectory calculation.

RALMO's field-of-view projects to a circle of approximately 1 m diameter at 5 km altitude, an area too small for most trajectories to pass directly through. Therefore, it was necessary to construct a region of assumed horizontal homogeneity in which the water vapour mixing ratio is constant. In order to maintain significant lidar SNR, we defined the homogeneous region, hereafter called the "homogeneous lidar region", to be a circle around the lidar. In order to maintain a SNR in the water vapour channel greater than 2 above 7 km altitude for the majority of the cases, we defined the homogeneous lidar region to be a circle of 3 km radius centered around the lidar. The size of the homogeneous region was chosen by varying the radius from a range of 1 - 25 km and finally increasing it to infinity. Radii below 3 km resulted in SNRs smaller than 2 below 7 km and in some cases halved the SNR of the water vapour channel at altitudes below 5 km, which decreased the altitude coverage for

the calibration and increased the noise in the primary calibration region. While radii above 3 km resulted in SNRs larger than 2 above 7 km, the water vapour profiles started to exhibit biases due to using too long integration times at certain altitudes and losing small features which had previously been visible. The 3 km radius provided the most altitude coverage with profiles closest to the radiosonde measurements and is the best compromise.

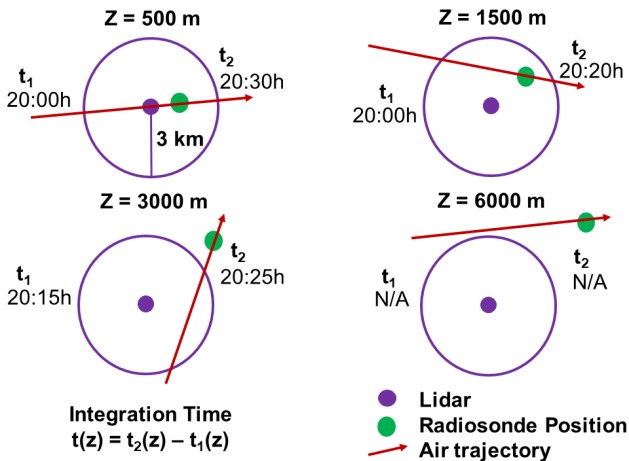

**Figure 1.** Trajectory calculation and scan selection hypothetical example. The purple circle around the lidar has a 3 km radius and represents the region in which we assume the humidity field is horizontally homogeneous. The green dot is the radiosonde position, the purple dot is the lidar position, and the red arrow is the air parcel trajectory. The variable $z$ refers to altitude, $t_1$ is the entry time, and $t_2$ is the exit time from the 3 km radius. The integration time, $t(z)$, is the total time that the air parcel spends inside the homogeneous region. When the air parcel trajectory does not intersect with the circle, then no data is available for calibration.

Fig. 1 shows how air parcels will always be "seen" by the lidar if the radiosonde remains inside the 3 km radius, whereas any air measured outside the radius may not intersect with the lidar region. If the trajectories do not enter the region, we do not use these altitudes for calibration. The entry and exit times from the homogeneous region mark the first and final scans used to calculate the lidar water vapour mixing ratios, with a maximum of 30 min of integration in order to accurately compare with the traditional technique, which uses a standard 30 min summation across all altitudes (Dinoev et al., 2013; Leblanc et al., 2012;

Whiteman et al., 1992; Melfi, 1972). The standard thirty minute integration is the average time it takes a radiosonde to reach the tropopause at mid-latitudes, and therefore generally covers the primary calibration altitudes. If the total time spent inside the homogeneous region exceeds 30 min, we take $\pm15$ min around the time of closest approach to the lidar. The variation of the integration length with altitude is shown in Fig. 2. The integration time will decrease with altitude for two reasons: higher wind speeds and the air parcel trajectories may intersect with the outer edges of the homogeneous region and are therefore inside

for shorter time spans (Fig. 1). The decrease in integration time with altitude will also change depending on the rate at which the radiosonde moves away from the lidar. The majority of nights had integration times less than 5 min above 7 km. However,

if the wind is strong at a particular altitude, sharp decreases in integration times may be seen as the radiosonde moves quickly away. It is also possible to see the integration times decrease and then increase again as the radiosonde drifts in and out of the homogeneous lidar region.

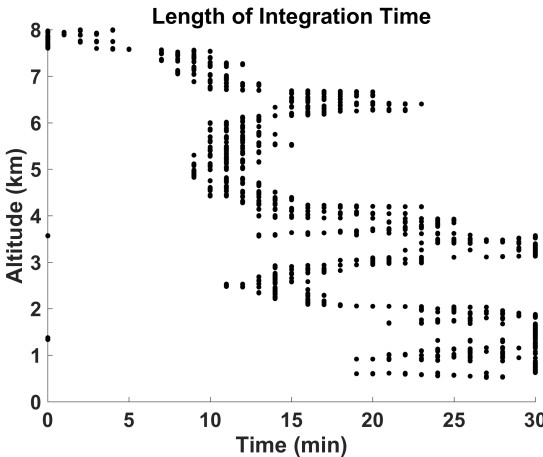

**Figure 2.** Example integration times from 22 July 2015. The lidar water vapour integration period is determined by the length of time the air parcels spend inside the homogeneous region. The integration time will decrease with altitude due to higher wind speeds. The maximum integration time is 30 min, in order to properly compare with the traditional analysis.

Once the appropriate scans have been chosen by the trajectory analysis, they are integrated to form the raw water vapour and nitrogen profile. The same procedure as in the traditional method (Sect. 3.1 and Sect. 3.2) is then followed to calculate the ratio of the two channels' profiles, find the appropriate calibration regions, and derive the calibration constant. The final calibrated water vapour profile for 22 July 2015 is shown in Fig. 3. The correlation algorithm selected 84% of the profile above 1.5 km to use for the calibration while regions with high variability were excluded from the calibration. The calibrated profile closely follows the radiosonde profile, with differences fluctuating between 5% and 20% over all altitudes. The uncertainty of the slope from the weighted fit is the uncertainty in the calibration constant due to measurement noise. The accuracy to which we know the calibration constant will be discussed further in Section 6.

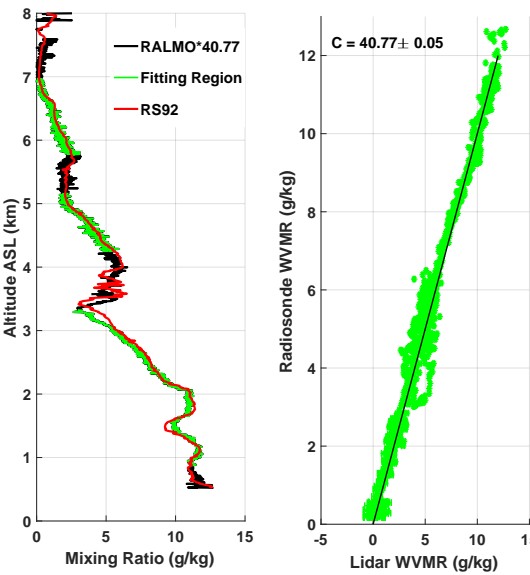

**Figure 3.** Left: The final trajectory-calibrated profile for 22 July 2015. The lidar profile is in black, the radiosonde is in red. The correlation calibration regions are shown by the overlaid green points. Right: The least square fit of the green points in the left panel. The uncertainty of the calibration constant is the standard error of the slope calculated from the weighted least squares fit.

For clarity on the similarities and differences between the traditional and trajectory methods, a flow chart of the calibration process for both methods is shown in Fig. 4. The main difference is the selection of the appropriate scans for calibration at the beginning. Both methods use the same process for choosing the appropriate calibration regions and calculating the calibration constant.

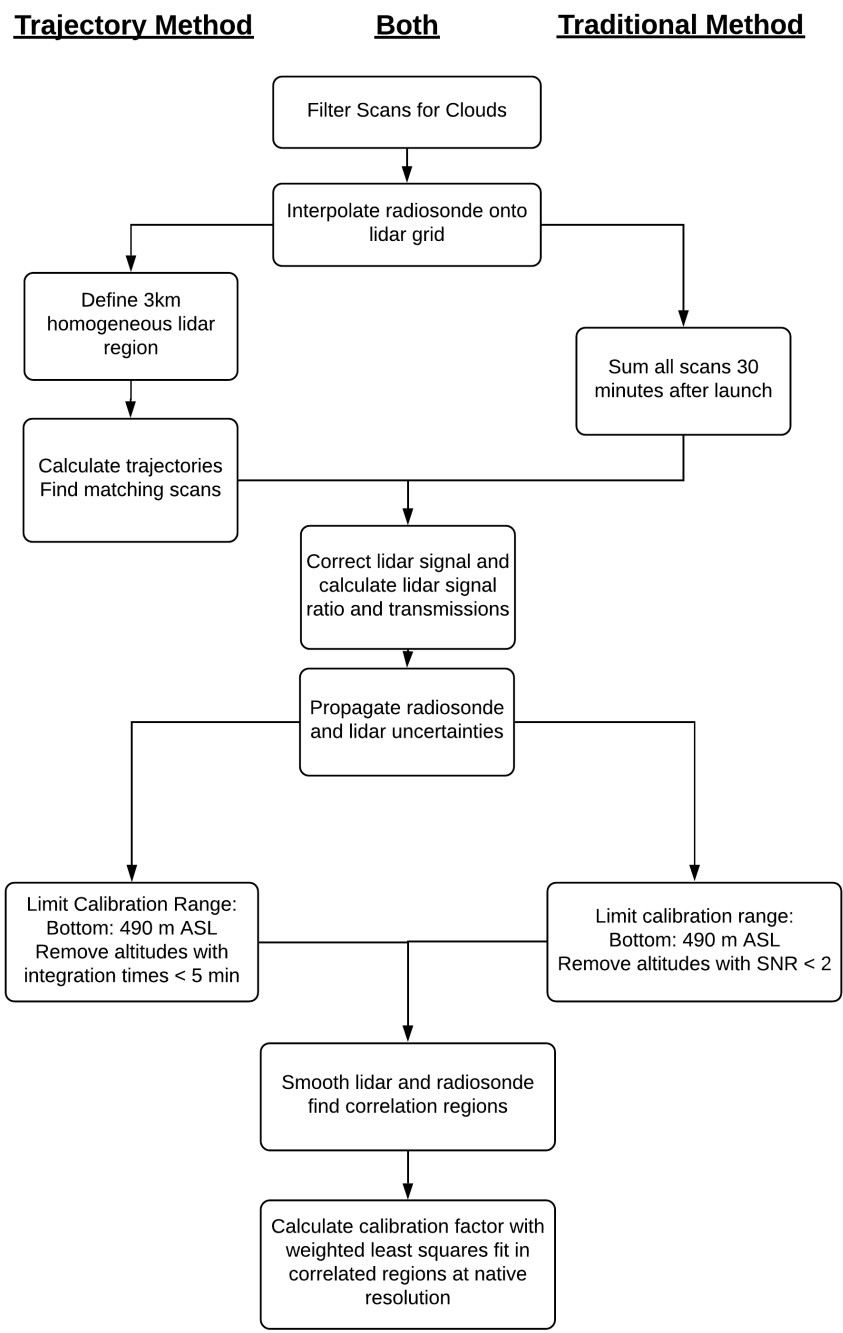

**Figure 4.** Flowchart of the steps to calibrate the RALMO lidar by the Trajectory Method (left) and the Traditional Method (right).

## 5    Comparing the Traditional and Trajectory Methods

We applied the trajectory technique to 76 nights between January 2011 and December 2016 in which 31 were removed due to lack of lidar measurements during the radiosonde launch window, primarily due to precipitation or routine maintenance. From the 45 remaining nights, the trajectory calibration and traditional method automatically removed 8 nights due to abnormally

5   high background values above 0.01 counts/bin/s. An additional 13 nights were removed from both the trajectory and traditional calibrations due to low signal-to-noise levels (below 1 SNR) and and the presence of clouds. The filtering process removed all of the nighttime flights from 2008 to 2011 due to significant cloud cover coincident with the radiosonde launch. A final list of the nights with their calibration constants is shown in Table 1.

While comparing the calibration constants from the two methods, it became apparent that we could separate them into two

10  groups when observing the water vapour mass mixing ratio contours over the course of the calibration period. One set of nights exhibited water vapour fields which were horizontally homogeneous around the lidar over the course of the 30 min calibration period, and were thereby dubbed "homogeneous" nights. The second set of nights showed movement of water vapour layers over 100 m in altitude over the course of 30 min, and were called "heterogeneous" nights. Table 1 has been divided into the two categories and shows the calibration constants for each night and calibration technique.

| Homogeneous | $C_{trad} \pm \Delta C_{trad}\%$ | $C_{traj} \pm \Delta C_{traj}\%$ | Abs(Difference) | Abs(Percent Difference) | Comments |
|---|---|---|---|---|---|
| 2011.10.05 | $38.61 \pm 4.5$ | $38.42 \pm 5.4$ | 0.19 | 0.49 | |
| 2012.07.18 | $39.67 \pm 4.5$ | $39.75 \pm 4.5$ | 0.08 | 0.20 | |
| 2012.08.09 | $40.42 \pm 4.7$ | $40.12 \pm 4.7$ | 0.3 | 0.75 | |
| 2012.08.29 | $39.23 \pm 4.4$ | $39.32 \pm 4.4$ | 0.09 | 0.23 | |
| 2012.12.13 | $40.30 \pm 5.3$ | $40.57 \pm 5.3$ | 0.27 | 0.67 | |
| 2013.04.24 | $41.77 \pm 4.8$ | $41.49 \pm 4.8$ | 0.28 | 0.67 | |
| 2013.06.05 | $41.77 \pm 4.4$ | $41.67 \pm 4.4$ | 0.1 | 0.24 | |
| 2014.01.23 | $41.31 \pm 4.5$ | $41.36 \pm 4.6$ | 0.05 | 0.12 | |
| 2014.03.21 | $40.39 \pm 5.5$ | $38.31 \pm 5.2$ | 2.08 | 5.42 | Trajectory calibration includes points below 1 km and Traditional does not. |
| 2014.07.18 | $40.05 \pm 5.2$ | $39.93 \pm 5.2$ | 0.12 | 0.30 | |
| 2015.11.11 | $41.50 \pm 3.7$ | $41.72 \pm 3.6$ | 0.22 | 0.53 | |
| 2016.03.09 | $45.42 \pm 4.5$ | $45.67 \pm 4.5$ | 0.25 | 0.55 | |
| 2016.08.24 | $43.95 \pm 6.2$ | $44.13 \pm 6.3$ | 0.18 | 0.41 | |
| **Heterogeneous** | $C_{trad} \pm \Delta C_{trad}\%$ | $C_{traj} \pm \Delta C_{traj}\%$ | **Abs(Difference)** | **Abs(Percent Difference)** | **Comments** |
| 2012.02.29 | $36.50 \pm 4.8$ | $35.83 \pm 4.8$ | 0.67 | 1.87 | |
| 2012.05.25 | $37.94 \pm 4.6$ | $37.11 \pm 4.6$ | 0.83 | 2.24 | |
| 2012.07.27 | $39.90 \pm 5.1$ | $39.39 \pm 5.1$ | 0.51 | 1.29 | |
| 2013.06.18 | $40.72 \pm 4.5$ | $39.68 \pm 4.6$ | 1.04 | 2.62 | |
| 2015.06.26 | $41.05 \pm 3.8$ | $41.16 \pm 3.8$ | 0.11 | 0.26 | Calibration done over the same homogeneous regions. |
| 2015.07.22 | $39.99 \pm 3.9$ | $41.33 \pm 3.8$ | 1.34 | 3.24 | |
| 2016.03.23 | $45.09 \pm 5.2$ | $45.41 \pm 5.2$ | 0.32 | 0.71 | |
| 2016.04.07 | $38.97 \pm 2.7$ | $39.95 \pm 2.7$ | 0.98 | 2.45 | |
| 2016.09.09 | $44.37 \pm 4.5$ | $43.72 \pm 4.5$ | 0.65 | 1.49 | |
| 2016.10.06 | $44.30 \pm 3.8$ | $44.59 \pm 3.8$ | 0.29 | 0.65 | |
| 2016.11.17 | $46.53 \pm 3.5$ | $48.07 \pm 3.4$ | 1.54 | 3.20 | |

**Table 1.** A comparison of the calibration constants of all nights used in this study. The table is broken into two sections- homogeneous and heterogeneous nights. Column 1 is the date on which the radiosonde was launched. Column 2 or $C_{trad}$ is the traditional calibration constant with its total percent uncertainty. Column 3 or $C_{traj}$ is the trajectory method calibration constant with its percent uncertainty. Column 4 is the absolute value of the difference between the two constants. Column 5 is the absolute value of the percent difference of the two constants with respect to the traditional calibration constant. Column 6 is for comments regarding the differences.

For the homogeneous nights, we hypothesized that if the water vapour field is stable for long periods of time and experiences very little change over the distance traveled by the radiosonde, then the radiosonde and the lidar should measure roughly similar water vapour content. Therefore, we should see small differences between the traditional and trajectory methods' calibration constants. While both methods should produce similar profiles and calibration constants on homogeneous nights, the two may not share the same calibration constants due to using different lidar scans (Fig. 5). The traditional method uses all profiles from the radiosonde launch to 30 min after launch, which are all scans inside the two dashed red lines in Fig. 5. The trajectory technique will choose the appropriate calibration scans based on each air parcel's trajectory and its position of closest approach shown by the scans between the magenta dots. Consequently, the trajectory method will not include measurements from altitudes where the air parcel trajectories do not intersect with the homogeneous lidar region.

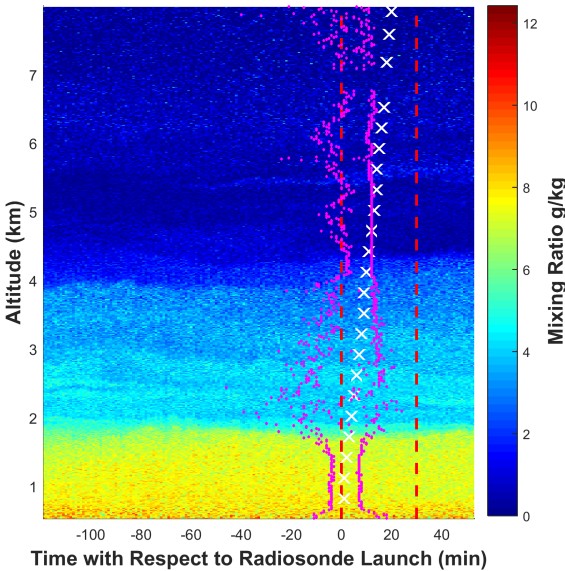

**Figure 5.** Lidar water vapour mixing ratio measurements on 2013-06-05 00:00 UTC. The time axis is measured relative to the radiosonde launch. The traditional method uses all scans between the two red dashed lines. The trajectory method uses all measurements between the magenta dots. The white "x" markers show the height of the radiosonde with time.

A subset of the homogeneous nights is shown in Fig. 6. The first column shows the percent difference from the mean water vapour profile over the 2 hours shown, with the calibration time between the two dashed red lines. The second and third columns are the percent difference between the calibrated lidar water vapour mixing ratio measurements and the radiosonde measurements for the traditional and trajectory techniques, respectively. The pink regions in the figures of the second and third

5    columns are the calibration regions used for each method. Both methods use similar calibration regions, due to the fact that the methods produce similar lidar water vapour profiles. The strength of the trajectory technique is shown by the reduction of the number of regions with large differences between the lidar and the radiosonde in the traditional method (see the second and third columns of Fig.6). For example, on 18 July 2012, the large difference between the radiosonde and the lidar at approximately 3.8 km is reduced by 20% in the trajectory method. The large difference is caused by the appearance of a water vapour layer

10    halfway through the calibration technique and can be seen in the water vapour contour. Sharp features, as shown on 9 August 2012 at 4 km are produced in both methods due to the sudden stratification of the water vapour layers. The large 10% difference between the radiosonde on 9 August 2012 at 2 km is also reduced by the trajectory method by 5%. On 24 April 2013, there is a large and increasing difference between the radiosonde and lidar measurements above 4 km with a slope of roughly 5% difference per kilometers altitude. This increasing difference between the radiosonde and lidar measurements is reduced in the

trajectory method to a constant bias of 5%; however, the variability of the difference between the sonde and lidar profile is larger than for the traditional method. While the trajectory method does reduce the bias in the traditional method on this night at altitudes above 4 km, it does produce larger variability at the same altitudes. The increase in variability of the difference is due to the smaller integration times at those altitudes due to the distance of the radiosonde from the lidar. Indeed, on 24 April 2013 the radiosonde is 4 km away from the lidar at 4 km altitude and 12 km away at 8 km altitude. Fast winds and larger distances from the lidar decrease the time the air spends in the lidar region and decreases the chances of intersection which results in shorter integration periods. The majority of the homogeneous nights have a percent difference in their calibration constants of less than 1% (Table 1). However, one night (21 March 2014) showed large differences and this is due to using different calibration regions in the trajectory method. The average percent difference in the homogeneous calibration constants is $0.43 \pm 0.21\%$ when not considering the anomalous night, but increases to $0.81 \pm 1.4\%$ when they are included.

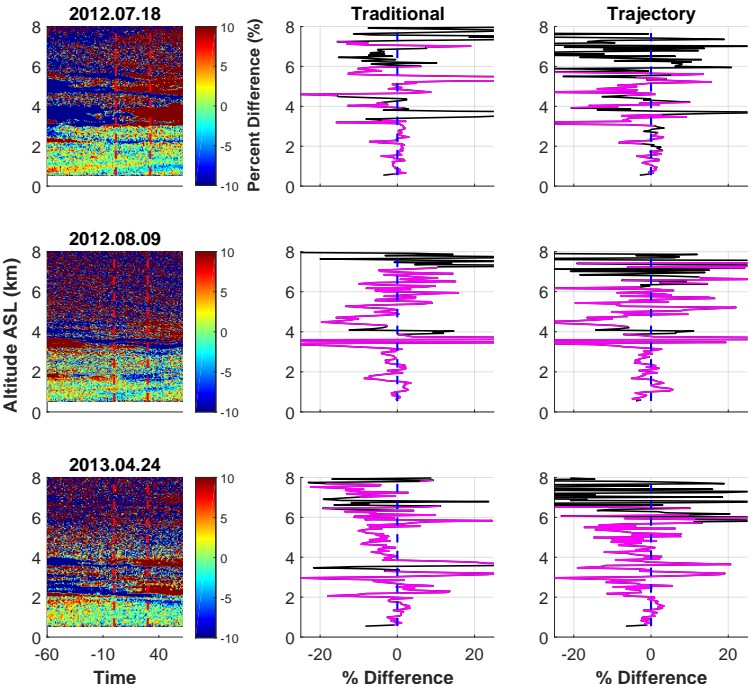

**Figure 6.** A subset of the dates with largely homogeneous conditions showing the differences between the traditional and trajectory calibration techniques. The first column is the percent difference from the mean water vapour mixing ratio profile over the two hours and averaged to 15 m altitude bins. The first red line is the time when the radiosonde was launched. The second red line is 30 min after radiosonde launch and indicates the last profile used for the traditional method. The second column is the percent difference between the radiosonde and the profile produced using the traditional method. The third column is the percent difference between the radiosonde and the profile produced by the trajectory method. Magenta regions are regions where the correlation between the radiosonde and the lidar are above 90%. During homogeneous conditions, the trajectory and traditional methods show good agreement, with similar percent differences with respect to the radiosonde. Large spikes are regions where the lidar and the radiosonde disagree on layer heights.

When the water vapour field is horizontally heterogeneous, meaning water vapour layers moved over 100 m in altitude over the course of the 30 min traditional calibration period, the trajectory method should better represent the air sampled by the radiosonde than the traditional technique. Layers on the order of several hundred meters thickness can change in altitude over this period, resulting in water vapour mixing ratios changing over 30% at a given height.

Similarly to the homogeneous nights, a subset of the heterogeneous nights is shown in Fig. 7. The contour of the percent difference from the mean water vapour profile for each night shows water vapour layers which change rapidly over the course of the 30 min calibration period (column 1 of Fig. 7). These rapid changes produce large differences in the radiosonde and lidar mixing ratio profiles if the movement of the radiosonde with respect to the air mass is not taken into account. These differences can be on the order of 15% - 20%, as is shown in the second column of Fig. 7, particularly on the night of 22 July 2015. Large differences on this night on the order of ± 10% are reduced to less than 5% by the trajectory method (third column of Fig. 7). Both methods on that night produce sharp differences at 3.8 km due to the sharp change in water vapour content. Above 4 km there is a constant bias between the radiosonde and the lidar in the traditional method of 10% which is reduced to 5% in the trajectory technique. On 27 July 2012 there are large difference features present throughout the entire percent difference profile for the traditional method on the order of 10%. These are similarly reduced to less than 5%, with the exception of the larger spike at 1.5 km caused by the sharp change in water vapour concentration. The night of 25 May 2012 shows less variation than the other two nights, but does have large differences at 4.2 km and 2 km. The higher feature is reduced from -25% difference to 10% by the trajectory method, while the percent difference in the lower feature changes from +20% to +10%.

Similarly to the homogeneous profiles, we do see an increase in noise at the higher altitudes of the trajectory method profiles. This is again caused by the large drift in the radiosonde's position, as well as wind speed. However, below 4 km we do see that the trajectory method reduces the differences between the radiosonde and the lidar by up to 15%.

The differences between the calibration constants on the heterogeneous nights is larger than the homogeneous nights due to the difference in calibration regions (Table 1). The average difference in the calibration constants on heterogeneous nights is $1.82 \pm 1.02$ from the traditional method calibration constant. One night out of the 11 in the heterogeneous nights showed very small differences in the calibration constant despite structural changes throughout the calibration period. This night used similar calibration regions that were also stable over the course of the calibration in both methods. If this night is not included, then the average difference becomes $1.92 \pm 0.93\%$.

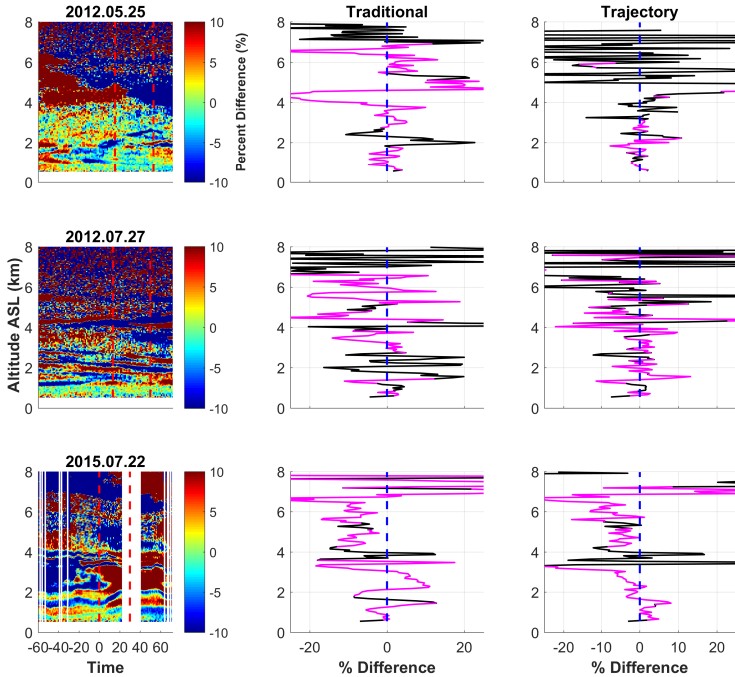

**Figure 7.** A subset of the dates with largely heterogeneous conditions showing the differences between the traditional and trajectory calibration techniques. This figure follows the same format as Fig. 6 and shows that when the water vapour field changes over the 30 min traditional calibration period, the traditional water vapour profile can look significantly different from the radiosonde. The trajectory method produces a profile with a smaller percent difference with respect to the radiosonde.

The average and the standard deviation of all percent difference profiles with the radiosonde from the trajectory and traditional method profiles are shown in Fig. 8. The average trajectory bias oscillates around 1%, but the variability increases above 4.5 km. This is due to the shorter integration times and smaller SNRs at higher altitudes (Fig. 8). The average traditional bias also oscillates around -0.7%, however, the average profile deviates farther from the center than the trajectory method (Fig. 8). The standard deviation of the ensemble of percent difference profiles between both calibration methods and the radiosonde shows that the trajectory method has 10-15% less variability with respect to the radiosonde profile above 2 km. Below 2 km the traditional and trajectory methods produce similar profiles on average, with similar consistency. In summary, the trajectory method shows a similar absolute bias to the radiosonde but with the opposite sign compared to the traditional method. The variability of the differences between the lidar and the radiosonde is 10 - 15% smaller in the trajectory method than it is in the traditional method between 2 and 4 km altitude, but is the same below 2 km and above 4 km.

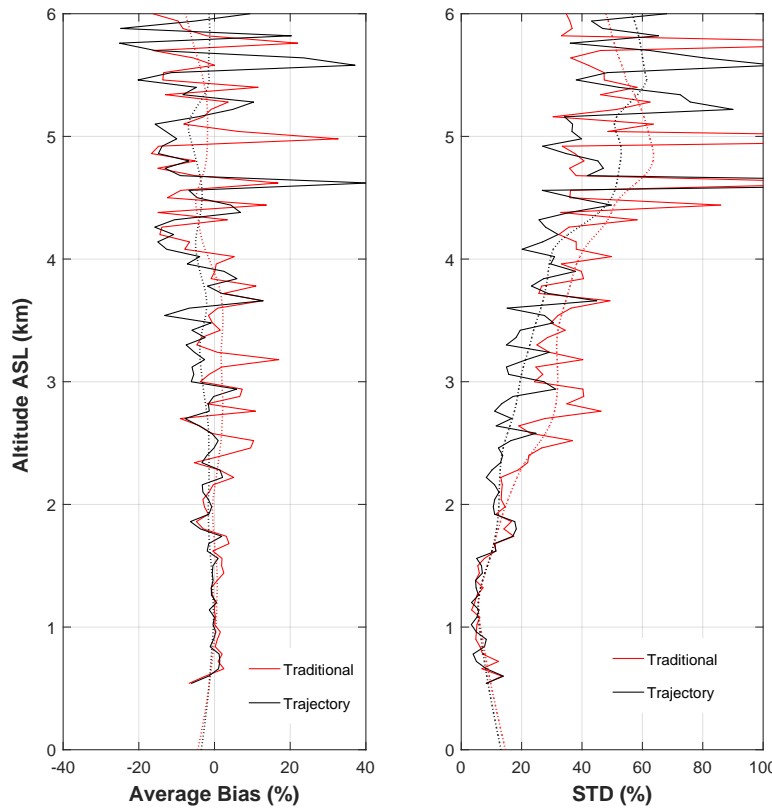

**Figure 8.** Left panel: The average bias between the radiosonde and the trajectory method calibrated profiles at 25 m vertical resolution for both the trajectory (black) and traditional methods (red). Dotted lines are further smoothed results using a running average of 75 m for both methods. Right panel: The standard deviation of all trajectory percent difference profiles at 25 m resolution for both trajectory (black) and traditional (red) methods. Dashed lines are the further smoothed results by a running average of 75 m for both methods.

## 6   Lidar Calibration Uncertainties for Trajectory and Traditional Methods

The standard practice for determining the uncertainty of the calibration constant has been to conduct extensive calibration campaigns and assume that the calibration value does not change over the campaign period and then measure the variability of the constant (Ferrare et al., 1995; Turner et al., 2002; Whiteman et al., 2006; Leblanc and Mcdermid, 2008; Dionisi et al., 2010; David et al., 2017). The variability of the constant is then assumed to be the uncertainty and the calibration constant is not changed until the next campaign when multiple radiosondes (or other reference calibration instruments) are available for calibration. The assumption that the calibration constant does not change over long periods of time introduces another source of uncertainty into water vapour measurements, which is often unknown until the next calibration period. Uncertainties calculated

during a campaign period vary between 4 and 5% of the calibration constant during the calibration period, but do not account for the individual sources of contribution nor do they typically account for the variability in the calibration constant beyond the campaign period.

Accounting for drift or changes in the calibration constant and its uncertainty is extremely important for long term trend analyses, since such a drift/change could easily be larger than the uncertainty of the calculated trend (Whiteman et al., 2011b). Many systems have now taken this into account by conducting daily or semi-daily calibration measurements either using an internal, hybrid, or external calibration. Taking more frequent calibration measurements with uncertainties calculated for each calibration then turns a systematic uncertainty component of a trend analysis into a random uncertainty component, particularly if the uncertainty of the calibration constant is recalculated with each calibration.

Previous studies have shown that the largest uncertainty is typically the uncertainty of the reference instrument (Leblanc and Mcdermid, 2008). It was not until recently that such detailed uncertainty budgets became available routinely for radiosonde measurements. The GRUAN radiosonde products are the first radiosonde profiles to have a published uncertainty budget for each measurement as a function of altitude (Dirksen et al., 2014). By using the GRUAN radiosonde product, we are now able to calculate the uncertainty in the calibration constant due to the radiosonde's uncertainties.

We investigated five major sources of uncertainty in the determination of the calibration constant for both methods: the lidar statistical, GRUAN radiosonde mixing ratio, dead time, aerosol extinction, and Ångstrom coefficient uncertainties. The uncertainty in the calibration constant, the lidar statistical uncertainties, and dead time were identified as the major sources of uncertainty in RALMO water vapour measurements by Sica and Haefele (2016), who also retrieved aerosol extinction, Ångstrom exponents, and their associated uncertainties. In the traditional method, the mixing ratio requires the uncertainty due to aerosol extinction and the Ångstrom exponent to be evaluated, as discussed in Whiteman (2003) and Kulla and Ritter (2019). The GRUAN radiosonde water vapour mixing ratio uncertainties were calculated using the reported GRUAN total uncertainties (combined statistical and systematic) for pressure, temperature, and relative humidity and by propagating through the Hyland and Wexler 1983 formula for saturation vapour pressure (Hyland and Wexler, 1983; Dirksen et al., 2014). We use an average pressure uncertainty profile calculated from all the nights when the pressure uncertainty is not reported for less than one-third of the nights. The radiosonde relative humidity uncertainties vary between 5% and 10% RH in the troposphere. The pressure uncertainties are on the order of $10^{-3}$ hPa in the troposphere, and the total temperature uncertainty varies between 0.1 and 0.3 K in the troposphere. The radiosonde mixing ratio uncertainties are linearly interpolated onto the lidar's 3.75 m resolution grid for the uncertainty determination.

The lidar mixing ratio statistical uncertainties are propagated through Eq. 1 using the random uncertainties from both the water vapour and nitrogen signals. The lidar statistical uncertainties from the trajectory method are smaller than the radiosonde uncertainties below 3 km but are larger than the radiosonde uncertainties, varying from 10% to 20% at and above 4 km from profile to profile.

Both the lidar statistical and radiosonde uncertainties were used as the weights for the least squares fit performed in Sect. 3.2, defined by Eq. 5 (Bevington and Robinson, 2003).

$$C_w = \frac{\sum_{i=1}^{K} \frac{R_i L_i}{\sigma_i^2}}{\sum_{i=1}^{K} \frac{L_i^2}{\sigma_i^2}}, \tag{5}$$

where $C_w$ is the calibration constant, $K$ is the number of points used in the fit, $R_i$ are the radiosonde mixing ratio points used in the calibration, and $L_i$ are the saturation and transmission corrected ratio of water vapour and nitrogen signals, and $\sigma_i$ are

the weights. Using the variance of the residuals of the least-squares fit, one can calculate the uncertainty in the fit, or "fitting uncertainty". This fitting uncertainty is the result of the amount of photon counting noise in the lidar measurements, and can be treated as the uncertainty in the calibration due to the lidar photon counting statistics. The fitting uncertainty is calculated using the standard equations for the slope of a line (Bevington and Robinson, 2003). The average trajectory method fitting uncertainty is 0.4% of the average calibration constant. The average fitting uncertainty for the traditional method is 0.3% of the

average calibration constant. The traditional method has smaller fitting or statistical uncertainties than the trajectory method due to the larger number of scans used per altitude, on average, compared to the trajectory method. The fitting uncertainty does not encompass the entire uncertainty of the calibration constant, since it is due only to the photon counting noise.

   The calibration of a lidar using a radiosonde is limited primarily by the accuracy of the radiosonde measurement. The uncertainty of the water vapour calibration constant due to lidar's random uncertainty and radiosonde's total uncertainty (both

systematic and random) was determined using the uncertainty propagation in Eq. 6 (JCGM, 2008).

$$U_{C_w} = \sqrt{\Sigma_{n=1}^{N} \Sigma_{m=1}^{N} \frac{\partial C_w}{\partial X_n} \frac{\partial C_w}{\partial X_m} cov(X_n, X_m)} \tag{6}$$

where $X$ is the measurement vector including both the radiosonde and lidar measurements (e.g. $X = [L_i, ... R_i, ...]$) used to calculate the calibration constant from Eq. 5 with length $N = 2K$. We make several assumptions in Eq. 6. First, by definition, the covariance of a radiosonde or lidar measurement uncertainty with itself is simply the variance. Second, we assume that

the lidar photon counting uncertainties are uncorrelated with each other. Third, we assume that the radiosonde measurement

uncertainties are uncorrelated with lidar measurement uncertainties. Lastly, we assume that the radiosonde measurement uncertainties are correlated with each other with a correlation coefficient of $r = 1$. Choosing $r$ equal to unity implies that we are assuming complete correlation and therefore the maximum possible uncertainty. With these assumptions, Eq. 6 becomes:

$$U_{C_w} = \sqrt{\Sigma_{i=1}^{K}(\frac{\partial C_w}{\partial R_i})^2 U_R^2 + \Sigma_{i=1}^{K}(\frac{\partial C_w}{\partial L_i})^2 U_L^2 + 2\Sigma_{i=1}^{K-1}\Sigma_{j=i+1}^{K}\frac{\partial C_w}{\partial R_i}\frac{\partial C_w}{\partial R_{i+1}}r_{ij}U_i U_j} \tag{7}$$

where $U_{L,R}$ are the corresponding lidar and radiosonde mixing ratio uncertainties, and $U_{i,j}$ are the uncertainties corresponding to the measurement vector $X$. The derivatives are calculated from Eq. 5. Note that the second term in Eq. 7 is the uncertainty due to the lidar's photon counting uncertainty. This term is the same as the fitting uncertainty discussed in the previous paragraph, and the values agree with each other within a tenth of a percent. The combined uncertainty in the calibration constant due to the radiosonde and lidar uncertainties is an average of 4% for both the trajectory and traditional techniques with signal levels

below 15 MHz which is to be expected since the traditional technique tends to sample the same volume of air as the trajectory as shown in Figure 5.

The dead time uncertainty can be large for RALMO, particularly during the daytime. Thus, Eq. 6 must be modified to account for this contribution when it is present. The dead time uncertainty is propagated through Eq. 1 assuming a non-paralyzable system and using Eq. 6. For RALMO, we assume a deadtime uncertainty ($U_\gamma$) of 5% or 0.2 ns, which was the

standard deviation of the retrieved dead times for all of these nights when using the Optimal Estimation method of Sica and Haefele (2016). The average calibration constant uncertainty due to dead time uncertainty is then $0.3\%$ of the calibration value for both the trajectory and traditional techniques, about equal to the fitting uncertainty.

The uncertainty in the calibration constant due to the uncertainty in the extinction profile is calculated using Eq. 6. The uncertainties for the extinction were assumed to be 100% to determine an upper-limit uncertainty contribution. However, the

derivatives of the calibration constant with respect to the individual extinction values were so small that the uncertainty contribution from the extinction was consistently less than 0.01% for all cases. The larger uncertainty component in the extinction is the calibration uncertainty due to the assumption of the Ångstrom exponent. The uncertainties in the Ångstrom exponent values were estimated by detrending the time series of these measurements from 2011-2015 using a summation of a 6 and 12 month sinusoid to the Ångstrom exponent measurements. The standard deviation of the fit's residuals was 0.34. The uncertainty in the

calibration constant due to the uncertainty in the Ångstrom exponent was then calculated to be $0.4 \pm .5\%$. While on average it is only an order of magnitude larger than the extinction uncertainty component, the Ångstrom exponent contributes more to the uncertainty when more aerosols are present. The maximum contribution from the Ångstrom exponent was 1.8% on 23 March

2016 due to the presence of a stronger aerosol layer. The rest of the nights had either no aerosols present or weakly interacting aerosol layers resulting in lower uncertainty contributions from the Ångstrom exponent.

Another possible contributor to the total calibration constant uncertainty is the overlap function. RALMO is designed to have no differential overlap in the water vapour and nitrogen channels and an overlap ratio between the nitrogen and water vapour signals of unity. However, a small differential overlap could result from chromatic aberration from the protective windows and edge filters (Dinoev et al., 2013). The average total uncertainty of both the trajectory and traditional calibration constant is 4.5%, with the majority from the radiosonde's contribution.

We also compared the average of the nightly calibration results to the standard deviation of the entire 24–night RALMO calibration time series used in this study. The RALMO system is known to have differential aging of its photomultipiers which causes the calibration to drift (Simeonov et al., 2014). A linear fit was made to the calibration time series and then removed to calculate the standard deviation of the calibration over 6 years. The standard deviations for both the traditional and trajectory time series were 4.5%, thereby agreeing with the average nightly uncertainty. The results from calculating the standard deviation of the time series shows that the typical methods used in calibration campaigns will generally give the same result as taking the average uncertainty of the individual uncertainties. However, we would suggest that taking the individual uncertainties is a better approach for long term analysis and maintaining consistency throughout a time series of measurements.

## 7 Summary

We have presented a new method, using GRUAN-corrected radiosondes, to calibrate Raman-scattering water vapour lidar systems that incorporates geophysical variability into the determination of the calibration constant. The trajectory method tracks the air parcels measured by the radiosonde and matches them with the appropriate lidar measurement time; thus, the integration time varies with height. We compared this method to the traditional lidar calibration technique where we sum 30 min of lidar measurements and fit them to a radiosonde profile.

The difference between the traditional and trajectory method calibration coefficients is due to the difference in 1-minute lidar scans selected by the methods, as well as the difference in correlation regions used to determine the calibration coefficient from these profiles. We found that when the water vapour field is homogeneous, the traditional method and trajectory method profiles will produce similar profiles, with slight differences due to the correlation regions included. The homogeneous nights had an average difference of 0.4% from the traditional calibration constant value. In contrast, the heterogeneous nights, or nights with significant structural changes over the 30 min traditional calibration period, had an average difference of 2% with respect to the traditional constant. We have also shown that using trajectories to track the air sampled by the radiosonde more accurately

reproduces the radiosonde profile when the water vapour field is variable and decreases the percent difference between the lidar and radiosonde measurements by 5 - 10%. In summary, we found the following:

1. The traditional and trajectory methods agree when the water vapour field is homogeneous during the radiosonde flight. The average difference between their calibration constants (when not considering the single outlier) was $0.43 \pm 0.21\%$.

2. The trajectory method provides a better fit with the radiosonde when the water vapour field changes appreciably over the time of the radiosonde flight. For these cases the calibration constants calculated by the trajectory method resulted in an average of $1.92 \pm 0.93\%$ difference with the traditional method calibration constants.

3. The trajectory method produces a smaller average bias between the radiosonde and the lidar than the traditional method between 2 and 4 km (Fig. 8). Adding points above 4 km does not change the calibration constant significantly as the photon counting uncertainty becomes large at these altitudes.

4. The combined lidar statistical and radiosonde mixing ratio uncertainties contribute an average of 4.5% uncertainty in the calibration constant determination for both calibration methods, where the radiosonde mixing ratio uncertainty is the dominating factor.

5. The uncertainty in the calibration coefficient due to the uncertainty in dead time contributes an average of 0.3% in the calibration coefficient for a 5% dead time uncertainty.

6. The uncertainty in the calibration constant due to the uncertainty in the extinction is less than 0.01%. The uncertainty in the calibration constant due to the Ångstrom exponent's uncertainty is larger and is on average 0.4%, but can reach higher than 1% when strongly-attenuating aerosol layers are present.

7. The average statistical uncertainty in the calibration constants produced by the trajectory technique is 0.4%, as opposed to the traditional method uncertainty of 0.3%. However, the fitting uncertainty is negligible relative to the uncertainty of the calibration constant due to the uncertainty of the radiosonde measurements.

8. The uncertainties calculated by the standard deviation of the trajectory and traditional method time series were both 4.5%, which is consistent with the total uncertainties calculated using Eq. 6.

A summary of the uncertainty components for both methods in the calibration constant is shown below in Table 2.

| Parameter | Parameter Uncertainty | Avg Uncertainty in Calibration Constant |
|---|---|---|
| Lidar Photon counting | 5 - 40% | <0.5% |
| Sonde Mixing Ratio | 0.5-40% | 4% |
| Dead time | 5% | 0.3% |
| Extinction | 100% | < 0.01% |
| Ångstrom Exponent | 0.34 | 0.4% |
| Total Uncertainty | - | 4.5% |

**Table 2.** Components of the calibration uncertainty, their inherent uncertainty, and their contribution to the uncertainty of the calibration constant for both the trajectory and traditional methods. The uncertainty contributions are the same for both methods since the only difference between the two methods is the selection of lidar measurements.

## 8 Discussion and Conclusions

The trajectory calibration technique attempts to more realistically represent the physical processes taking place during a radiosonde - lidar calibration, by ensuring the radiosonde and lidar sample the same air mass. This tracking method was built upon the methods suggested in Whiteman et al. (2006), Leblanc and Mcdermid (2008), Adam et al. (2010), and Herold et al. (2011). Similarly to the techniques discussed in these studies, we match the measurements at each altitude with the radiosonde. However, Whiteman et al. (2006) assumed a horizontally homogeneous and uniformly translating atmosphere and did not consider varying wind speed and direction. In Whiteman et al. (2006) the integration time was varied with altitude in order to keep the random uncertainty below 10%, however, the position of the air parcels was not considered. This technique was ultimately found to be not as accurate as other methods, and was later improved upon using the correlation comparisons in Whiteman et al. (2012). Our method does not assume a uniformly translating atmosphere, however, we do consider a homogeneous region around the lidar and the integration time is varied as a function of the time the air parcels spend inside the homogeneous region. Leblanc and Mcdermid (2008) used four methods to match the radiosonde and the lidar measurements: 1) no matching, summing 2 hours of lidar profiles 2) using all lidar scans before the radiosonde reaches 10 km, about 30 min of scans, similar to our "traditional method", 3) only altitudes with minimum water vapour variability over 2 hours are used to calibrate, and 4) only using scans which were coincident with the radiosonde altitude - similar to the Whiteman et al. (2006) "Track" technique and our trajectory method. However, method 4 did not track the air parcels as we did. Leblanc et al. (2012) found that the second method provided the smallest variation in their calibration constant, but did mention that the other methods produced very close results and could be used as well.

Another way to attempt to correct for the movement of the air mass or radiosonde is to follow the methods in Dionisi et al. (2010); Whiteman et al. (2012) which look for regions of high correlation as in these regions it is more likely the radiosonde and lidar are sampling the same air mass. The traditional method, using the correlation algorithm, does provide similar calibra-

tion constants to the trajectory method on homogeneous nights. However, using the combined correlation algorithm with the trajectory tracking can provide more regions of high correlation for the calibration, particularly on heterogeneous nights when the air mass changes rapidly as it passes over the lidar. In some cases, of course, it may not provide more regions for calibration particularly if the wind speeds are high and the radiosonde quickly leaves the 3 km homogeneous lidar region.

Using our new trajectory method has several advantages over the traditional technique. The first advantage is that the method presents an automatic and new scheme to calibrate with non-co-located radiosondes. The trajectory method does not rely on the radiosonde's location, but instead relies of the direction of the air measured by the radiosonde. The trajectory method will automatically find the appropriate calibration times as a function of altitude for the lidar. Lidar stations may then be able to use radiosondes launched farther away more effectively, thus allowing more frequent calibrations over the year as well as reducing

the need for expensive calibration campaigns. Lidar stations who use our technique with radiosondes located several kilometers away may find it necessary to expand their "lidar region" to greater than 3 km. Secondly, this method allows for calibration if the water vapour field changes rapidly in space and time, allowing more nights to be used for calibration when they would otherwise be discarded due to large differences between the traditional lidar profile and the radiosonde. Lidars with drifts or fluctuations in their calibration constant that may require many calibrations might also find this technique useful. Additionally,

frequent and accurate lidar calibrations are critical for detecting water vapour trends and small changes in water vapour. We consider the representation uncertainty to be greatly reduced in the trajectory method because we are now considering the location of the radiosonde relative to the lidar. Lastly, this technique provides an automatic, objective and quantitative method of determining acceptable calibration nights. This method could conceivably be expanded to work with ozonesondes or tracking other conserved quantities such as aerosols. We have not attempted to expand this technique, but leave it up to others who may

find it useful. The method could also be further expanded to work with wind field measurements that include vertical wind speeds.

Future studies using this technique could possibly show an improvement using a cone instead of a cylinder for the homogeneous lidar region. The trajectory method works better between 2 and 4 km but worse than the traditional method above. Using a cone could increase the integration time for the higher altitudes where less trajectories tend to intersect the homogeneous

lidar region cylinder, improving the comparison. We initially tested a cone, and set its radius using the horizontal correlation lengths of water vapour using the wind speeds measured by the radiosonde. However, this scheme produced a cone which was too variable in size to be useful for calibration. We explored using a cone the same size as the lidar's field-of-view, but it allowed so few trajectories to intersect the cone that no calibration could be performed. Varying the cylinder size did not significantly change the shape of the profile above 4 km, but reduced the noise. Reducing the noise of the profile is important;

however, as the lidar measurements only contribute on the order of 0.1% of the uncertainty to the calibration constant, it would not provide much benefit for RALMO. Using a cone could prove advantageous for other sites which exhibit higher wind speeds than Payerne. However, for sites which use radiosondes which are not co-located with the lidar, it would not be as beneficial as a cylinder. We would encourage others who might implement this method to try a cone to see if it significantly improves their results.

A significant new aspect of our study is using calibrated GRUAN radiosondes whose analysis includes a complete uncertainty budget. The full uncertainty budget shows the radiosonde measurement is the dominant uncertainty source as compared to the uncertainty in the regression line on an individual night derived from the uncalibrated lidar measurements and sonde. The uncertainty in the lidar measurements, the dead time, and the extinction components contribute an order of magnitude smaller uncertainty than the radiosonde. However, the Ångstrom exponent can contribute on the same order of magnitude uncertainty as the radiosonde if there are strongly-interacting aerosols present during the calibration. Using the GRUAN sondes allows a calibration to be determined with a full uncertainty budget on an individual night, as opposed to requiring a times series of nights to calculate a statistical calibration variation. The uncertainties in our calibration determinations could be reduced using the hybrid method of Leblanc and Mcdermid (2008), as a refinement of our method would be to combine the trajectory calibration with an internal lamp source or some sort of internal calibration technique which could further reduce the variation in the calibration over time.

Eleven of the calibration nights in this study showed significant structural variations in water vapour over the 30 min traditional calibration period. These nights had an average of 2% difference in the calibration constant, which is less than the average calibration uncertainty of 4.5%. Therefore, the trajectory and traditional methods do not produce statistically different calibration values. However, the trajectory method does more accurately reproduce the radiosonde profile than the traditional method between 2 and 4 km and above 4 km the methods do equally well (Fig. 8). The water vapour content below 4 km for the nights in this study was an average of 87% of the total content measured by the radiosonde. Therefore, we believe that calibration should be limited to below 4 km where the signal is highest and the trajectory method performs best. Additionally, the points above 4 km do not make a significant difference in the calibration factor obtained.

The RALMO has an average of 50% uptime over the last 10 years, making it an ideal database for the detection of water vapour trends in the free troposphere. In addition to frequent measurements, trend analyses also require minimal uncertainty and well-characterized retrievals. The aim of this work was to develop a calibration method that characterized the uncertainty of the calibration constant as well as making sure it was physically consistent with the reference instrument. The trajectory calibration

technique will be used in conjunction with an internal calibration method to produce a 10 year water vapour climatology and UTLS trend analysis using RALMO measurements.

*Data availability.* All GRUAN data is accessible on www.gruan.org and access may be requested through them. MeteoSwiss lidar data may be requested by contacting Dr. Alexander Haefele (Alexander.Haefele@meteoswiss.ch).

5 *Author contributions.* Shannon Hicks-Jalali was responsible for developing the trajectory method technique code, applying the technique to the GRUAN radiosondes and RALMO measurements, comparing the method results to the traditional method, deriving and calculating the uncertainty budget for each calibration, and manuscript preparation. This work will be used as part of her doctoral thesis. R. J. Sica was responsible for supervision of the doctoral thesis, contributions to manuscript preparation. Alexander Haefele had the original idea for a trajectory method, was also responsible for surpervision of the thesis, and helped with manuscript preparation. Giovanni Martucci helped 10 with manuscript preparation, provided RALMO and radiosonde data from MeteoSwiss, as well as helpful guidance on the workings of the MeteoSwiss lidar code.

*Acknowledgements.* We would like to thank the GRUAN support team for providing the GRUAN-processed radiosonde measurements. Shannon Hicks-Jalali would also like to thank Ali Jalali who spent time reading the paper and providing helpful scientific discussions and suggestions throughout the entire process. This project has been funded in part by the National Science and Engineering Research Council 15 of Canada through a Discovery Grant (Sica) and a CREATE award for a Training Program in Arctic Atmospheric Science (K. Strong, PI), and by MeteoSwiss (Switzerland).

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
