# Peer review of "Calibration of a Water Vapour Raman Lidar using GRUAN-certified Radiosondes and a New Trajectory Method"

_Atmospheric Measurement Techniques, 2018_

## Referee Comment (RC1)

**Review of «Calibration of a Water Vapour Lidar using a Radiosonde Trajectory Method » Shannon Hicks-Jalali et al.**

The paper by Hicks-Jalali et al. presents a new version of Whiteman et al.'s (2006) Raman lidar calibration method with radiosondes. This new method takes into account horizontal air mass movement. Hicks-Jalali et al. present a detailed description of the uncertainty associated with calibration.

**General comments:**
Shannon Hicks-Jalali compares the calibration results obtained by the « traditional » and the « improved » methods. If we regret that the retrotrajectory work is only done in 2D and not in 3D, the idea is interesting to continue to overcome the problems related to the non-colocation of the reference measurement with that of the lidar. However, I remain unconvinced by the discussion and conclusions of the article which consider that the new method is significantly better (in case of an « heterogeneous » atmosphere) because it is poorly supported by the results of the article in terms of number of calibrated nights (no more) or uncertainties (not less) with the use of both methods, and does not benefit from any validation (comparison of calibrated profiles with a third instrument). It is imperative to rework the discussion and review the method's contributions in a more factual way.

I think the article is not publishable as it stands, the article needs to be reworked to answer the following major remarks (more specific comments follow):
**1.** The structure of the paper needs to be reworked to improve clarity and precision. I suggest to include section 3.2 into current section 2.2. I suggest to improve internal structure of Sect. 4 (see major comments n°4), to remove the Summary section (you should share its content into sections 4, 5, the conclusion and avoid repetitions) and to split the current section 7 into a Discussion section (that you need to develop) and a Conclusions section.

**2.** The « traditional » method is not described simply at the beginning of the article, there is just a list of bibliographical references that the reader must read without knowing which one is used precisely. Explanation loops are given as the article progresses but they arrive too late. A short or detailed description of the so-called « traditional » method should be added to the introduction and/or methodological part.

**3.** In the description of the lidar measurement and the description of the « improved » methodology, a lot of choice in filtering the data are made based on high or low SNR but it is never quantified. Please be more specific on this point.

**4.** Sect. 4 presents many issues:
  *4.1* The structure needs to be reworked so that the reader can have the following elements:
  ○ Presentation of data
  ○ Methodology to differentiate between nights when water vapour is homogeneous and nights when water vapour is heterogeneous
  ○ Presentation of Table 1
  ○ General comment (the current last paragraph)
  ○ Illustration of the different characteristics with Fig. 5 and 7
  ○ Conclusion with Figure 8
  *4.2* Figures:
  ○ they are under used, even not used for specific subplots. Maybe there is too many figures,

- the legends need to be shortened (some analyses of the figures are made in the legend whereas it should be done in the text),

    *4.3* The quantification of biases of « 0 % » in average. I suggest taking absolute values and indicating the sign of the bias.

**5.** It is repeated over and over in the summary and discussion/conclusions that the method is « more accurate », in other words but the results on uncertainty does not quantify this improvement. The discussion/conclusion about the advantages of the method, it must be thorough:

    *5.1* Almost no discussion about the limitations of the method: 32 % of night are calibrated (what about the others?) on 6 years (the first 3 years not being calibratable also)

    *5.2* The benefits presented are based on the theoretical expectations that motivated the implementation of this methodology. The uncertainty is presumably better but this is not reflected in the budget calculation.

*5.3* Discussion should be pushed before opening the perspective of using this dataset to « UTLS climatology over 10 years »: how to do that with 24 nights above 2008-2016, how to calibrate night where this methodology could not be applied? It should have thought because the authors wants to use the whole dataset for trends study.

**Specific comments:**

*Title*

I strongly suggest to refer to RALMO in the title or to find a way to indicate that it is a test of this new methodology of calibration done on one lidar which could potentially be applied to others.

*Abstract – page 1*

*Lines 1-4:* I would keep theses sentences for an introduction because it is too general. At least, please shorten this part.

*Lines 5 & 6:* Reference citations should not be included in this section unless they are essential. Using radiosondes is the most used technique for calibrating so please select a maximum of one reference, I would suggest Whiteman et al. (2006) which is the closest from the one you will use?. Maybe, the method is even better summarized in Whiteman et al. (2012).

*Line 7 « movement of radiosonde »:* I suggest replacing it with « movement of air masses »

*Line 12:* Precise on which period the calibration has been performed (i.e. 2011-2016).

*Lines 14-15:* The authors use « more accurately » but there is no conclusion in the article that quantifies that the uncertainty associated with the new technique is better than the « traditional » one . I suggest replacing « reproduces more accurately » with « reproduces accurately ».

*Lines 16-21:* The summary associated with the uncertainty budget is too detailed. Please replace this part by one value (or range of value) quantifying the total uncertainty associated with the calibration.

*Page 2*

*Line 2:* Please replace « the primary contributor» with « one of the main contributors ».

*Line 3:* Add reference to « ...high temporal and spatial variability »

*Line 4:* I suggest deleting « uniquely ».

*Line 6:* I suggest deleting « more » or please add a reference.

*Line 8:* Replace « take » with « make ».

*Line 9:* Delete « also ».

*Line 10 « Several Raman… external methods »:* I would place this sentence in the following paragraph.

*Lines 17 to 22:* It's too detailed whereas it is not the main subject of the article. I suggest deleting this part.

*Line 29:* Please add the use of GNSS as an external instrument to calibrate Raman lidars and a reference. I suggest: David, L., Bock, O., Thom, C., Bosser, P., and Pelon, J.: Study and mitigation of calibration factor instabilities in a water vapor Raman lidar, Atmos. Meas. Tech., 10, 2745-2758, https://doi.org/10.5194/amt-10-2745-2017, 2017.

*Page 3*

*Lines 1-2 « External...do not contribute »:* I suggest deleting this sentence.

*Line 7 « as RS92 radiosondes are the most frequently used calibration radiosondes Immler et al., 2010 ; Dirksen et al., 2014). »:* I'm not convinced that the main objective of this article was to correct sondes for the calibration of Raman lidars. Please rephrase.

*Line 8:* Replace « Vaisala » with « not corrected »

*Line 9:* Replace « errors » with « uncertainties »

*Lines 9-10 « A portion of… uncertainty »:* I suggest moving this sentence in the next paragraph, line 17 before « This paper attempts to... ».

*Lines 12-15:* I suggest moving the whole paragraph line 27.

*Line 19:* Please explain in few sentences (2-3) what is the « traditional » method, references are not sufficient considering that the improvement of this technique is the main subject of this paper.

*Line 20 «as the radiosonde takes approximately 30 min to reach the tropopause »:* It depends on which latitude the sonde is launched, it will be larger near the equator. Please specify it or add a location.

*Line 21:* Your statement should be supported by a reference or some statistics from your database.

*Line 28 « in order to ensure that the lidar and the radiosonde are measuring the same air »:* It was also the goal of the « traditional » method. Please check.

Line 28: I would suggest deleting « improved ».

*Line 29:* Replace « of the radiosonde and the » with « , ».

*Page 4*

*Line 7:* Add « in Payerne » at the end of the sentence.

*Line 9 « their respective uncertainties »:* The full uncertainty budget for the calibration is not given for the « traditional » method. What about the « representation uncertainty » for example? I will come back to this in more detail in following comments.

*Line 13:* Does the lidar always start working at 0:00UTC? Does « bi-weekly » refer to twice a week or one every two weeks? Please specify it in the text.

*Line 14:* Why is their only « a subset » of these radiosonde processed by GRUAN? Please explain it in the text.

*Line 23:* Please specify that the analysis was conducted on an initial set of 76 flights but in the end only 24 of them were used.

*Page 5*

*Section 2.2:* Precise that the RALMO is operating day and night. Give the effective measurement time (« 50% » in the conclusion, it should be precised earlier) and what explain that 50% of the time is not exploitable.

*Line 2:* The authors explained that the instrument « is designed to be an operational lidar, and as such, needs to have high accuracy, temporal measurement stability, and minimal altitude-based corrections (Dinoev et al., 2013; brocard et al., 2013) ». The study of the instrument's performance made in the bibliographical references and years of operation should determine whether the instrument really has a high accuracy, temporal stability of measurements and profiles that start close to the ground. Please be more specific.

*Lines 4-6 « RALMO operates...scattering channels »:* Move this sentence to the previous paragraph.

*Line 9 « a sufficiently high SNR »:* This is a major issue in this paper. The SNR is often used to select some data and reject others, but no threshold based on bibliographic references or empirical tests specific to this study is defined. You need to clarify this.

*Lines 12 & 13 « After the filtering process...small features »:* I wonder if this sentence should not be found in the description of the calibration methodology. Or at least move the interpolation part of the radiosonde profiles to section 2.1 and specify here that there is no interpolation for lidar profiles.

*Line 15 « 3.1 Tracking air parcels »:* I would move this title at the end of the page. Paragraph line 16 to 20 is more an introduction to sect. 3.

*Page 6 – Figure 1*

I would use « homogeneous region » instead of « homogeneous cylinder » to be more consistent with the text. I would suggest adding letters or numbers to the different steps and refer to them in the text.

*Page 7*

*Line 7:* As simple that it could be, you should provide mathematical explanation/an equation to illustrate your calculation.

*Line 8:* This assumption should be discussed a little more. You might provide a physical discussion about when (or if) this assumption is (would be) realist and its limitation, please cite references that could support this hypothesis or this discussion.

*Line 9:* What do you mean by « We do not explicitely consider the vertical movement of the air parcel in this method »? From your description of the methodology, I understand that you do not consider at all the vertical movement. Please be more specific.

*Line 14:* For my point of view, the « lidar region » refers more to the ~1 m diameter at 5 km that the radius of 3 km choosen after for the sensitivity test. You might use a name more related with the assumed homogeneity of the water vapor like or its use for calibration, « homogeneous lidar region » or « calibration region » as you call them after in the paper.

*Lines 14-15:* What is the decisional parameter? Is this the SNR? Which threshold?

*Lines 16/17/18:* You characterized some « very low SNR », a « large enough SNRs » and « the highest SNR ». Please quantify or explain why you define it this way.

*Page 8 – Figure 2*

Is this a conceptual scheme or a real example? Please precise it in the legend. If it is an example, give the date of the measurement. The second sentence of the legend is also explained in the text. Please replace this sentence with something like « The purple circle corresponds to the lidar region ».

*Page 8*

*Lines 6-7 « The standard thirty minutes...tropopause »:* It sounds quite general in your text but it corresponds to mid-latitudes. It would be a different duration for polar regions or in the tropics. Please be more specific by adding « at mid-latitudes » for example.

*Lines 7-8 « Integrating...by radiosonde »:* I suggest deleting this sentence.

*Lines 9-10:* The analysis of Figure 3 consists in one sentence. Either your analysis is too short, either the figure is not necessary. Please reconcile.

*Page 9 – Figure 3*

*Legend « The integration time…analysis. »:* Please remove these sentence from the legend. It is already explained in the text or is part of the analysis.

*Figure 2 – Figure 3 – Figure 4*
In this section, Figure 2 seems to be a conceptual method, Figure 3 refers to July 21 and Figure 4 to July 22. It is not explained why these 3 dates are choosen for each example. Considering that Section 3 details the method of calibration by sonde, you might choose the same date to illustrate the different aspects of it. If not, please justify why.

*Section 3.2 - Pages 9 and 10*
I suggest moving this section in or after Section 2.2 because it is more about the « Lidar measurement » than about the « Radiosonde Trajectory Method ».

*Page 9*
*Lines 10-11:* I would rephrase this way: « The central wavelengths of the water vapour and nitrogen channels of the RALMO were choosen to minimize temperature dependence. »
*Line 14:* Please add a reference.

*Page 10*
*Lines 4-5 « In RALMO's case, the ratio...2014). »:* Please move this sentence to Sect. 2.2.
*Line 8 « the corrected signal »:* Which correction? Please be more specific.
*Line 9:* Does « the correlated and weighted least squares fitting » correspond to the « traditional » method? If so, why not call it this way in the abstract so that you don't list so many references? It could be included in the introduction in this way as well.
*Line 14 « low SNRs »:* Please specify.
*Lines 15-17:* It is interesting because there is a desire to minimize the bias associated with not taking into account the vertical displacement of air masses.
*Line 20:* I suggest replacing « lidar region » with « calibration region ».

*Section 3.3 – Pages 10 and 11*
It seems to be the traditional method that is mainly described here except that the lidar data are selected as described in Section 3.1. Be careful to distinguish in the text of this part (and even in the entire article) between what is specific to your study and what is traditional. Perhaps the structure you have chosen is a little confusing on this point.

*Page 11*
*Lines 2-10:* I suggest moving this part to Sect. 2.1 or at least be more concise.
*Line 12 « for July 22, 2015 »:* Why did you choose this date instead of July 21 as in Figure 3? That would seem more consistent.
*Line 15:* Please replace « error » with « uncertainty ».
*Line 15:* Please add a reference. What was the range of uncertainty found in the litterature for the « traditional » method?

*Figure 4 – Page 11*
The date is not specified, please do it. The last sentence is not necessary.

*Page 11*
*Line 3:* What do you mean by « repairs »?
*Line 4 « abnormaly high »:* Please quantify or explain.
*Line 6:* I suggest: « and presence of clouds ».
*Lines 6-7 « The filtering process...radiosonde launch »:* This sentence seems to say that there were clouds every night of measurement for all the duration of the measurement. Is that what you mean?

Does this mean that there are 3 years of measurement that cannot be used because of systematic presence of clouds?

*Line 8:* If I understood correctly, between 2008 and 2016, there are only 76 calibratable nights and they are all condensed over the period 2011 and 2016. What solution for 2008-2010? On the other hand, over the 76 nights, with the implementation of this methodology, only ~30% can be calibrated? What about the other nights of measurements? It is essential to discuss these aspects in Section 7.

*Table 1 – Page 13*
*Column « Difference »:* Sign of the difference is missing or it should be specified that it is absolute value.
*Legend:* I suggest deleting the word « calibration » in the second sentence. The text from « Two nights in the homogeneous » to « variability in the water vapour » should not appear in the legend.

*Page 13*
*Line 6 « show a good agreement »:* Please quantify or explain.
*Line 7:* I suggest adding « water vapor conditions around the location of the lidar measurement ».
*Line 8 « as « homogeneous » or « stable » nights in Table 1 »:* Only « homogeneous » is used in Table 1 in comparison with « heteregenous » , « stable » is used in the « Comments column » for heterogeneous nights. Please reconcile.
*Line 9 « an average bias of 0 % »:* Please used absolute value to calculte the average bias. Maybe you should use the standard deviation and  specify if the bias is more positive or negative.

*Page 14*
*Line 1 « The bias on that night is reduced when using the trajectory method »:* What do you mean? Please rephrase.
*Lines 2-3:* Figure 5 is composed of 9 subplots and show results for 3 dates but there is only one sentence that refers to it in the text. You need to analyse your figures or do not put them in the paper.

*Figure 5 – Page 14*
*Legend - 4th sentence:* Why do you mention « White vertical regions » whereas there is none of them in the figure? Please delete this sentence.
*Legend - 7th sentence:* You should use « magenta » instead of « pink » or change the color of the corresponding line in Figure 5.
*Legend - 8 and 9th sentences:* It is part of the analysis and should not appear in the legend. Please delete them.

*Figure 6 – Page 15*
There is no 2012-07-27 measurement in Table 1. Is this the right date? Please check.
*Legend – 1st sentence:* I suggest deleting « launch at 0 min ».

*Page 15*
*Lines 1-5:* This part explains the method to characterize a night as homogeneous or heterogeneous, it should come earlier in this section; before discussing about the results. Please reconcile.
*Line 3:* Figure 7 is quoted before Figure 6 (p16 - l.11) in the text. The order of the figures is not respected.
*Line 3:* As for other figures in the article, Figure 7 is commented in only one sentence.
*Line 8:* See my previous comment on the average bias.
*Line 9:* Please refer to a figure or a table.

*Line 14:* Why « 11 » whereas there are 12 heterogeneous nights in Table 1?

*Figure 7 – Page 16*
Why did you choose these three nights? You must analyze all three of them, when you add a figure it should bring an supplementary information, otherwise it means that the figure is not necessary.
*Legend – 2nd to 6th sentences:* This is already explain in legend of Figure 5, please just leave the part on the white vertical regions and refer to legend of Figure 5.
*Legend – 7 and 8th sentences:* Again, this does not have to appear in the legend.

*Page 16*
*Lines 1-2:* Shouldn't these nights be described as homogeneous in this case? Does this not question the methodology of homogeneous versus heterogeneous characterization?
*Line 4: T*he average bias is around 1 % here whereas it was 0 % earlier (p15 - l. 8). Please reconcile.
*Line 5:* Please specify what threshold you chose to define that the variability increases above 5 km.
*Line 6:* Here you precise the sign of the bias whereas you have not done it earlier, same for the significant number of digits. Please reconcile.
*Line 7:* Please rephrase the sentence.
*Line 8 « better fits »:* Please quantify.
Line 11: Once again the commentary on the figure does not exceed one sentence. Please develop or remove the sentence.

*Figure 8 – Page 17*
I suggest choosing one reference (either the trajectory or the « traditional » method) to avoid the multiplication of the data and to put the average bias and the standard deviation on the same plot.

*Page 17*
*Starting Page 16 -line 10 to Page 17 – line 7:* This is a description of the methodology and should come earlier in the section. Please put this explanation after the introduction of Table 1.
*Lines 3-4:* What do you mean by « the majority »? How many nights? In Table 1, it seems that it is true for 6 nights on 12. This represents half of the data not the majority.
*Lines 4-5:* Please specify the dates .

*Page 18:*
*Lines 2-7:* You repeat yourself. This was already mentioned earlier in the article.
*Lines 17-18:* Please precise how the fitting uncertainty is calculated.

*Page 19*
*Line 2:* Which part is the systematic part versus the statistical part of the uncertainty? This is important to determine given that one of your objectives is to establish trends in the UT/LS.
*Line 5:* Please explain what do you mean by the « measurement vector » with regard to the lidar measurement.
*Line 12:* Please correct the indices of the third term of Eq. (5)
*Lines 13-25:* The comments of the equation is not clear enough. First you should describe each term, precise the source of each uncertainty (radiosonde measurements, lidar (including lidar's photon counting and deadtime and covariance term), quantify the uncertainties before (« 8 % » coming to late in the next section, ?, 5 %, calculation?) and after propagation (values of Table 2) and you should use Table 2 to support this section. Actually, Table 2 comes too late in the paper. You should also precise how is considered your uncertainty on the calibration constant due to the

radiosonde: statistical or systematic? Then you can conclude on the total uncertainty of the calibration constant for which you are found after propagation: an average value of 4%.

*Line 16:* Why do you mentioned that the fitting uncertainty is the same that lidar's photon counting uncertainty. As you do it l.24 to compare it to the deadtime uncertainty? Is there a physical or metrological explanation? The fitting uncertainty seems to be due to the methodology and is different from these two uncertainties.

*Line 23:* I suggest not making a line break here and including it in the previous paragraph.

*Line 23 « we assume a deadtime uncertainty () of 5% or 2 ns »:* How is estimated this uncertainty: literature? test?

*Line 25:* Please remove « including dead time effects ».

*Page 20*

*Line 7 « The calibration time series »:* Do you mean the 24 nights? Why don't you show them on a figure?

*Line 8:* What do you mean by « de-trended »? Please explain.

*Line 8 « over ten years »:* You work on the 2011-2016 (i.e. 6 years), why do you speak about 10 years?

*Line 17 – Page 20 to Line 14 – Page 21:* This « Summary » part should not be a part in itself. These information should be in sections 4 and 5 and in the conclusion. It is necessary to rethink the structure at the end of the article, which for the moment consists in two parts: 6 Summary and 7 Discussion and Conclusions (see comments below).

*Line 23 « due to the lidar profiles »:* Please rephrase.

*Page 21*

*Line 4:* Please change « error » into « uncertainty ».

*Line 4 « is negligible »:* Do you neglect it in your calculation? This uncertainty does not appear in Table 2. Please specify it in the text.

*Line 5:* This part is supposed to be a summary but this value « 8 % » appears for the first time. This should have been discussed earlier.

*All Sect. 6:* This section contains many repetitions and brings many repetitions in relation to the whole article. There are key elements (such as the quantification of the uncertainty on the radiosonde of 8 % or Table 2) that should appear earlier in the article.

*Page 22*

*Line 1:* This article lacks a real discussion part, so it is necessary to restructure it as follows:

      6 Discussion

      7 Conclusions

*Line 2 « has several advantages »:* What about the limitations? The following points should be discussed: no consideration of the vertical movement of air masses, only 6 years calibratable over 9 and only 32% of the 76 nights with exploitable radiosonde profiles according to the method's prerequisites. A real issue is: which method did you use for all other nights of measurements than the 24 selected? Because otherwise it means that over 9 years of data there are only 24 nights of calibrated and therefore usable data? We can't make trends over 24 nights.

*Line 19 « an automatic […] scheme »:* What do you mean? Does this mean that this method has been implemented in a production chain for water vapor profiles of the RALMO lidar?

*Page 23*

*Line 1:* If we read the reasons for the exclusion of measurement nights for the calculation of the calibration coefficient on page 12, there does not seem to be more or less nights used with the new method rather than with the « traditional » method. Please reconcile.

*Line 5:* The uncertainty has not been quantified for the « traditional » method (at least not indicated in the article) or in this article. This is something that is difficult to quantify, but as a result it is difficult to conclude that it has improved it. Indeed from a theoretical point of view it is, but in practice (and in this discussion) you do not relate it to your results. This uncertainty appears in your introduction and in the conclusion part but no word in the section focusing on the uncertainties.

*Lines 8-9 « This method could be conceivably... »:* Yes , it is a good idea for ozone measurements.

*Line 10 « The method could... »:* Yes but it should be presented as a limitation and not only as a perspective because it means that the « representation uncertainty » may not be as negligible as that.

*Line 12:* I suggest starting here Sect. 7 Conclusions.

*Line 15:* How do you assess that? Reference? Maybe Whiteman et al. (2011b)?

*Line 15 « 50 % uptime over ten years »:* Why is this only appearing now? 9 years of measurements -> 76 nights that can be calibrated -> 24 nights calibrated in practice, what about the other nights? Where is that 50%? Which calibration methodology for the entire database?

*Line 24:* To study trends in the UT/LS, please recall the total uncertainty associated with your profiles or refer to the article that assess the performances of the RALMO to measured water vapor in the UT/LS on a routine basis.

---

## Short Comment (SC1) · 24 Nov 2018

I'm missing 1. the reference of our recently published new calibration method for the water-vapor Raman lidar measurements "Calibration of Raman lidar water vapor profiles by means of AERONET photometer observations and GDAS meteorological data" (https://doi.org/10.5194/amt-11-2735-2018) and 2. the discussion of our paper "Comparison of Raman Lidar Observations of Water Vapor with COSMO-DE Forecasts during COPS 2007" (DOI: 10.1175/2011WAF2222448.1) where we took already into account the time-height-dependences of radiosonde data when comparing those data to Raman lidar data. Table 1: I'm wondering about the large variability (about 15-20 %) of the determined calibration constant and that it seems to have no unit . . .

---

## Short Comment (SC2) · 16 Dec 2018

We determined the calibration constant by using radiosonde data at different times (heights) as the radiosonde moves upward with time (DOI: 10.1175/2011WAF2222448.1). Of course, we did not apply the trajectory idea. But we took into account that the air masses might have changed during the uplifting of radiosonde. I think, it would be fair to mention this paper at least in your discussion, too.

---

## Referee Comment (RC2) · Whiteman (Referee) · 29 Jan 2019

Review of Hicks-Jalali et al, "Calibration of a Water Vapour Lidar using a Radiosonde Trajectory method" by David N. Whiteman

**General Comments**

The subject manuscript presents a new twist on the use of radiosonde profiles to determine the Raman water vapor lidar calibration constant. The method defines a vertical column surrounding the lidar and considers radiosonde measurements that fall within this vertical column for use in the calibration. Other criteria for selection of data are applied as well, but the requirement of physical co-location is the principal novelty mentioned by the authors. A concern is that the authors' trajectory technique does not produce significantly different results from the "traditional" technique so at the end of the paper one is left wondering about the value of the approach.

But I think the authors are overlooking a significant novelty of this paper which, with a modest amount of new analysis, could make a significant contribution to the lidar calibration literature. Thanks to the very large efforts of GRUAN, radiosonde data with fully characterized uncertainties are now available at GRUAN launch sites such as Payerne. Thus, in the long history of water vapor lidar-radiosonde intercomparisons for the purposes of calibrating a water vapor Raman lidar (since Melfi et al (1969); so now extending for half a century!), to my knowledge, this is the first effort to use rigorously determined uncertainty values for radiosonde measurements in the process. This novelty permits the radiosonde data to be weighted properly in the calibration regressions. Thus, as mentioned, I suggest a shift of focus away from the radiosonde trajectory technique and toward quantifying how the use of GRUAN sondes for calibration differs from using ordinary sondes. The authors could quantify this effect by comparing the calibration results when using weights in the regression of radiosonde data (appropriate due to the availability of the GRUAN data product) versus when not using weights in the regression. I would expect that this would result in different calibration regions being chosen at least. But there could be other effects as well that are worth documenting. I don't believe that this has been done before and would constitute a valuable addition to the literature. It might be appropriate to consider a title change to something like "Use of rigorously characterized radiosonde data for calibrating a Raman water vapor lidar".

Also, as mentioned in the specific comments, I believe that it is necessary for the authors to account for the influence of aerosols on the differential transmission correction.

**Specific Comments**

Abstract

1. L1 – statement is made that "Lidars are well-suited for trend measurements in the upper troposphere and lower stratosphere, particularly for species such as water vapor."
   1. The measurement requirements for detection of trends in water vapor differ dramatically between the UT and the LS. Paragraph 16 in Whiteman et al, 2011b and the first several paragraphs of the discussion section of Whiteman et al, 2012 detail the argument that Raman water lidar is much better suited to trend detection in the upper troposphere than the lower stratosphere. Also, I might suggest that instead of just saying "Lidars" here, to specify "Carefully calibrated and quality-controlled Raman Lidars..."
2. L4 and beyond – the current technique is improved with respect to the traditional technique but no comparisons are done with respect to other "improved" techniques. It is my hope that we can address that in follow-up research.

3. L5 – Whiteman et al (2006) is cited for a "track-sonde" technique that was used. It is worth noting, however, that the track-sonde technique as used in 2006 did not perform as well as the more simple variable temporal-spatial smoothing routine described in that same publication. More importantly, a significantly more sophisticated technique for performing radiosonde calibration was presented in Whiteman et al, 2012. It does not explicitly track the sonde but the geometrical similarity requirements imposed in that routine, I expect, achieve some of the same collocation benefit that is discussed in the authors' technique. These details should be mentioned.
4. L29- "paralyzation" → "paralysis"

Introduction

5. Statement is made that "instruments with high spatial-temporal resolutions, such as lidars, are uniquely suited to long-term stratospheric and tropospheric water vapour studies". For lidars to provide a good signal-to-noise measurement in the UTLS requires significant temporal and spatial smoothing. So I do not agree that high spatial-temporal resolution measurements make lidars uniquely suited to long-term UTLS studies of water vapor since the temporal and spatial resolution must be degraded to achieve an acceptable S/N in the UTLS.
6. L6 - "of" → "from"
7. Lines 7-9. Statement is made "Lidar measurements are particularly useful for creating statistically significant water vapor trends of the UT and LS region … " and Weatherhead, 1998 and Whiteman 2011b are used to support the claim. I don't believe that Weatherhead et al makes any statement about the suitability of lidar for this task. Also, as stated above, Whiteman et al 2011b expresses doubt that Raman lidar would be suitable for LS trend detection; a claim that is amplified in Whiteman et al, 2012. So I would suggest a statement such as "Carefully processed, stably calibrated Lidar measurements can be particularly useful for creating statistically significant water vapor trends in the UT region ..."
8. Paragraph starting "Internal calibration techniques …
   1. reference is made to Venable et al, 2011 as an example of the white light technique, which is correct. The next sentences, however, refer to the limitation of using a single lamp and the need for multiple lamps or a scanning technique. This is confusing since Venable et al showed the utility of the scanned lamp technique so that work does not suffer from the limitations of the single lamp technique as implied by the current discussion. I suggest revising the paragraph so that the first reference cited is one that makes use of a single lamp.
   2. Later in the same paragraph it is stated that the uncertainty in the knowledge of the ratio of the Raman cross sections is 10% from Penny and Lapp, 1976. The work of Avila et al, 2004 and Venable, 2011 however point toward an uncertainty of this cross section ratio closer to 5%. To support this, Fernandez-Sanchez (the lead of the group in which Avila did his work) has privately communicated with me that 5% is his assessment of the absolute accuracy of their water vapor cross sections and given that the nitrogen cross section uncertainty is in the range of 1-2%, this is consistent with a claim of ~5% uncertainty in the cross section ratio. Venable et al has some text concerning this. So I believe that an assignment of 5% to the uncertainty of the Raman water vapor/nitrogen cross section ratio is justifiable. But at least this more recent work makes the 10% Penny and Lapp uncertainty from 1976 no longer appropriate.
9. P 4, L19. Immler et al, 2010 is used as a reference for the GRUAN RS92 correction technique. Immler et al discusses error characterization in general but does not present the RS92 correction technique. The Dirksen et al, 2014 reference is more appropriate.

10. P9, Lines 8-9. Statement is made that "we do not correct for aerosols as they are considered to have a very small contribution to the overall mixing ratio". I take this to mean that the differential transmission due to aerosols is not accounted for. In the 1992 reference that is cited to support the authors' statement, it is shown that with aerosol optical thickness at 355nm of 1.0 the calculated mixing ratio would change by ~4% as compared to a pure Rayleigh atmosphere. Indeed, AOT of 1.0 is quite a turbid atmosphere but this result also implies that AOT of 0.25 would yield a 1% change in mixing ratio. One's first impression might be that 1% uncertainties are small enough to neglect (I do not agree). But neglecting aerosol differential transmission does not introduce a random uncertainty but rather a systematic one. And surely in a paper that has long term trend detection as a stated goal, elimination of systematic uncertainties that can be up to 4% must really be done. So I strongly encourage that the authors address this deficiency. Note that it is not necessary to calculate aerosol extinction directly from the lidar data to adequately make this correction. One can instead use collocated aerosol optical thickness measurements along with a reasonable estimate of the height of the boundary layer to develop a simple model for calculating the aerosol differential transmission such that the residual uncertainty in the aerosol differential transmission correction is well below 1% even under turbid conditions. This is the technique that we generally use to handle this tricky part of the Raman water vapor lidar analysis.

11. P10, Lines 16-17. "we require … to be correlated to greater than 90%." I assume that by this the authors mean than the correlation coefficient of the linear regression is 0.9 or greater. If so, please restate in terms of correlation coefficient to avoid confusion.

12. P10, last line. I do not see that the results of Aug 8, 2012 showing a 5% offset. Is this something that is apparent from the Table? If not, please clarify that this information cannot be gleaned from the Table.

13. P15, lines 5-10. A qualitative comparison of results of homogeneous and heterogeneous cases is made but the actual standard deviations, for example, are not given. From Table 1, it seems that for the homogeneous cases, the standard deviation of the calibration constants derived using the traditional technique is less than that of the trajectory technique. For the heterogeneous cases, the trajectory technique gives slightly smaller standard deviation as stated. I do suggest giving the actual standard deviation values in the table and discussing the significance of these standard deviation differences since they seem to be rather small.

14. P 17. "Lidar Calibration Uncertainties for Trajectory and Traditional Methods". Three sources of uncertainty are listed: lidar statistical uncertainty, GRUAN radiosonde uncertainty, dead time uncertainty. The "usual" way that radiosondes have been used in a Raman lidar calibration effort (e.g the MOHAVE, AWEX, IHOP, PECAN field campaigns) is to assume that the lidar calibration value has been constant over the duration of a field campaign and that differences in calculated calibration constants relate to statistical uncertainties, collocation uncertainty, etc. Following this procedure, a single calibration constant is determined from all the radiosonde comparisons in a field campaign and that calibration value is used for some period of time until another large intercomparison effort with multiple radiosondes is performed. This is the technique outlined in discussions of the hybrid technique, for example, and in such cases there is another very significant component of the calibration uncertainty, which I call the calibration transfer uncertainty, that is not listed here by the authors since it does not pertain to what they are doing (but it is very significant in the overall discussion of lidar calibration). This systematic uncertainty can be taken to be the standard deviation of the individual calibration constants used to determine the mean calibration constant that is finally used in a field campaign type of study. I understand that the authors are doing things differently and are re-calibrating the lidar with every available radiosonde. In fact, the authors approach is much preferred from the standpoint of developing a time series for trend detection because each time a different calibration constant

is used for the lidar, a step-change systematic uncertainty is introduced into the time series. This is inevitable. So to decrease the influence of these systematic step-changes, frequent calibrations are needed so as to make these systematic uncertainties, in effect, components of the random uncertainty budget in the time series. The authors refer to some of this later in the paper but here is where it should be introduced. Thus to recalibrate the lidar as frequently as possible serves to transform a component of the systematic uncertainty budget (where it can really destroy a trend calculation) into a random uncertainty. Note that the DOE/ARM Raman lidar is recalibrated with respect to microwave radiometer every three days achieving this randomization of calibration constant. However, campaign mode calibration efforts as described in the MOHAVE papers do not achieve this. So … I suggest that the authors clarify this. There is discussion in Whiteman et al, 2011b about the need to randomize components of the systematic uncertainty budget to improve time series for trend detection.

1. It's also in this section where the uniqueness of the use of GRUAN sondes for this calibration task should be highlighted. This is the first time, to my knowledge, that linear regressions of radiosonde/lidar data have been performed with weights that make use of carefully characterized radiosonde uncertainties. This is significant.

15. P18, line 21. The term "scan" is used here and earlier but it is not clear what "scan" means. Please go back in the paper and define how you use this term the first time it appears.

16. P19, line 11 … I chuckled when I read that eq 5 is a simplified version of eq 4. Upon inspection eq 5 is about twice as long as eq 4 so does not appear much simplified. You might just say "With these assumptions, eq 4 becomes ..."

17. P20, line 3. This is where it becomes clear that you are recalibrating the lidar with each radiosonde. You also make the point that this is different than for field campaigns as in the Leblanc and Dionisi references. Good. Now, as mentioned earlier, you can make the point that this approach helps to randomize a component of the systematic uncertainty making the resulting time series more appropriate for trend detection.

18. P20, lines 13-16. I've already commented that ignoring aerosol differential transmission neglects a systematic bias which is a strong concern and goes against the prescription of the BIPM/GUM where all known systematic biases should be corrected (see quote in Whiteman et al, 2012 or go to the GUM itself). Also, though, the way that the sentence reads it is not clear what 5% refers to. Finally I would say that one should perform the calibration of the lidar data in the same way that it is analyzed for trend detection and one would not want to neglect aerosol differential transmission when trying to create trend-detection quality time series of water vapor measurements. So aerosols really do need to be accounted for in this analysis and in the full analysis of the lidar data.

19. P23, lines 3-5. "frequent and accurate lidar calibrations are critical for detecting water vapor trends …" The earlier discussion of randomizing components of the systematic uncertainty budget is the main argument for why this statement is true so you should add a citation here. But I need to repeat that the measurement challenge in the LS is very different than in the UT so that your statement really only applies to the UT. BTW, these are the reasons why trend detection in the UT is so much easier with Raman lidar than in the LS:

1. The natural variability of water vapor in the UT is much higher than in the LS. So the relatively large random uncertainty of Raman water vapor lidar does not deteriorate the time to detect trend by a large fraction in the UT.

2. On the other hand, the natural variability of water vapor in the LS is very low and the random uncertainty of lidar measurements is much, much larger in the LS since it is farther away than the UT and water vapor concentrations are so small in the LS. So the random uncertainty of Raman lidar measurements in the LS typically will swamp the uncertainty

budget and greatly extend the time to detect trend using the methodology of Weatherhead et al, 1998.

3. According to the modeling cited in Whiteman et al, 2011b the anticipated trends in LS water vapor are smaller than those in the UT making trends more difficult to quantify in the LS.

4. Because of much lower S/N lidar measurements in the LS, small sources of systematic bias in the lidar measurements can more easily corrupt the time series. The larger signals in the UT are more resistant to such unknown sources of bias.

20. P23, last paragraph. At the end of the study a conclusion is that the trajectory method does not produce statistically different calibration values than the trajectory method. This does not argue strongly for the technique presented here. I would suggest looking for ways to decrease the standard deviation of the calculated calibration values. In Whiteman et al, 2012 we found that by using the adaptive technique described there we could reduce the variability of the calculated calibration values by requiring that the correlation coefficient between the lidar and radiosonde profile segments be higher. You might try adding that into your algorithm since, as I understand, you already require $R^2 > 0.9$. The point here is that it should be a goal of this work to achieve a more stable calibration constant than that achieved with the traditional technique.

---

## Author Comment (AC2) · 17 Apr 2019

Please see attached response.

Please also note the supplement to this comment:
https://www.atmos-meas-tech-discuss.net/amt-2018-246/amt-2018-246-AC2-supplement.pdf

---

## Author Comment (AC1)

Reply to Dietrich Althausen's Discussion Comments:
The discussion comments are in black, and are followed by our answers in red.

Comment:
I am missing the reference of our recently published new calibration method for the water-vapor Raman lidar measurements "Calibration of Raman lidar water vapor profiles by means of AERONET photometer observations and GDAS meteorological data" (https://doi.org/10.5194/amt-11-2735-2018) and

Thank you for bringing this article to our attention. We originally did not mention calibration using a sun photometer as an external calibration technique, but we agree that it is a good idea to mention it along with the rest of the external calibration methods in the Introduction.

Comment:
I am missing the discussion of our paper "Comparison of Raman Lidar Observations of Water Vapor with COSMO-DE Forecasts during COPS 2007" (DOI: 10.1175/2011WAF2222448.1) where we took already into account the time-height-dependences of radiosonde data when comparing those data to Raman lidar data.

Thank you for sending this article. Our method is different from that given in the above reference in that we make use of trajectories which seem to have not been mentioned in your article. We would be happy to cite your paper along with the other articles using methods which take the time-height-dependence into account, such as that given by Leblanc et al. 2012. We believe that this class of methods is different from our technique because they do not take the movement of the radiosonde with respect to the air mass into account by using trajectories.

Comment:
Table 1: I'm wondering about the large variability (about 15-20 %) of the determined calibration constant and that it seems to have no unit . . .

Thank you for pointing out that we did not put the units of the calibration constant. We considered the calibration constant technically unitless since it is mass/mass, however, it is indeed in units of "g/kg" and could be included in the text.

The large drift of the calibration constant over 10 years is known to occur for RALMO (Simeonov, 2014) and is thought to be the result of the differential aging of the

photomultipliers which causes a large drift over time. The differential aging is due a culmination of factors such as the exposure of the water vapour channel to high count rates (20+ MHz) during the daytime, the lidar's uptime of 50% over 10 years, and the fact that the photomultipliers have never been changed or upgraded. When considering the operational nature of the instrument, it is not surprising that the calibration factor changes with time. The reference below discusses this drift in detail.

Simeonov, V., Fastig, S., Haefele, A. and Calpini, B.: Instrumental correction of the uneven PMT aging effect on the calibration constant of a water vapor Raman lidar, Proceeding SPIE, 9246, 1–9, doi:10.1117/12.2066802, 2014.

---

## Author Response (AR1)

**A General Response to the Editor and to the Reviewers:**

Thank you very much for your detailed comments and the time you took to consider this paper. We have responded to the general comments below. Some of the general comments are repeated in the "specific comments" section, therefore, we have responded to both the general and specific comments but only provided the changed or relevant text in the specific comments section. We have also included a marked-up version of the manuscript showing what sections are removed and what has been added.

We believe that there may have been some confusion as to the goal of this paper and we would like to clarify this for the reviewers and the editor. The goal of this paper was to create a general method to calibrate raman water vapour lidars using a radiosonde's trajectory. This method may be applied to any lidar system which uses radiosondes, provided the radiosondes record wind measurements (speed and direction). Otherwise, there is no other limiting factor in the method's design.

In the conclusions, we state that this method will be used to calibrate the RALMO water vapour measurements for an upcoming trend analysis. We would like to stress that this is future work, and the trend analysis is not the subject of this paper. Therefore, we will not go into detail on the trend analysis here. We believe discussions regarding the calibrations for the trend analysis should be reserved for the following papers. We have other calibration techniques which will be used for trend analysis, however, this paper is not the place to include that discussion. We have answered the reviewer's comments regarding the trend analysis accordingly.

Thank you again for considering this paper and we hope you find the answers to your concerns satisfactory.

**Response to Reviewer #1**

**Response to General Comments:**

Comment: The structure of the paper needs to be reworked to improve clarity and precision. I suggest to include section 3.2 into current section 2.2. I suggest to improve internal structure of Sect. 4 (see major comments n°4), to remove the Summary section (you should share its content into sections 4, 5, the conclusion and avoid repetitions) and to split the current section 7 into a Discussion section (that you need to develop) and a Conclusions section.

We have reworked several of the sections as per your suggestion, particularly sections 4 and the discussions and conclusions. Please see the specific comments for the specific changes.

Comment:

The « traditional » method is not described simply at the beginning of the article, there is just a list of bibliographical references that the reader must read without knowing which one is used precisely. Explanation loops are given as the article progresses but they arrive too late. A short or detailed description of the so-called « traditional » method should be added to the introduction and/or methodological part.

We agree that the definition of the traditional method was not clear. We had hoped that Figure 1 would be sufficient to explain the traditional method, but we do agree that some text is necessary.

We will address this problem in two ways:
1) A new section will be added to discuss the traditional method before the trajectory section. We will include a short introduction to the traditional method in the introductory paragraph to this section but refrain from a full literature review as we believe it has been well discussed in several papers and we do assume the reader is somewhat familiar with the technique if they are interested in reading the paper.
2) The previous sections 3.2 and 3.3 have now been added to the traditional method procedure. The procedure discussed in these two sections is used for both the trajectory and the traditional, therefore, we think it is appropriate to be introduced here. This way the readers have a clear view on how the traditional method is calculated in our study.

Comment: In the description of the lidar measurement and the description of the « improved » methodology, a lot of choice in filtering the data are made based on high or low SNR but it is never quantified. Please be more specific on this point.

We agree that the SNR values should be quantified and have clarified or removed the few cases which were ambiguous. Thank you for pointing this out. We have answered all of the specific comments regarding the SNR values individually where they occur, as this was more efficient.

In general our cutoff SNR value was 2 as lower SNRs have too much of a noise contribution.

Comment:
Sect. 4 presents many issues:
4.1 The structure needs to be reworked so that the reader can have the following elements:
◦ Presentation of data
◦ Methodology to differentiate between nights when water vapour is homogeneous and nights when water vapour is heterogeneous
◦ Presentation of Table 1
◦ General comment (the current last paragraph)
◦ Illustration of the different characteristics with Fig. 5 and 7
◦ Conclusion with Figure 8

We agree that this section does not flow appropriately and have reorganized it as follows:
- Introduction of the calibration constant comparison, Table 1, and definitions of homogeneous and heterogeneous nights.
- Discussion of Homogeneous nights
  - Why are the calibration constants different? Referencing Figure 6 (now Figure 5).
  - Discussion of Figure 5 (now Figure 6) and the example homogeneous nights
  - Analysis of the homogeneous calibration constants
- Discussion of Heterogeneous nights
  - Discussion of Figure 7 and the example heterogeneous nights
  - Analysis of the calibration constants
- Conclusions with Figure 8

The new section includes changes over several pages of text; please see the included mark-up version of the revised manuscript.

4.2 Figures:
◦ they are under used, even not used for specific subplots. Maybe there is too many Figures, the legends need to be shortened (some analyses of the figures are made in the legend whereas it should be done in the text),

We agree that the discussion of the Figures should be expanded and will do so for each of the Figures not discussed in enough detail. Comments regarding specific figures are discussed below. We would prefer not to shorten the captions of the figures because it is often the case where figures are "borrowed" from papers for presentations or discussions and can be often taken without the proper context if it is not clear in the caption. Therefore, we would like to keep some analysis and context in the caption to account for these situations.

4.3 The quantification of biases of « 0 % » in average. I suggest taking absolute values and indicating the sign of the bias.

Thank you for pointing this out. We have changed the way in which we discuss the comparison for the two methods, and believe that this removes the problem of biases around 0. Please refer to the new Section 5 for the changed text.

It is repeated over and over in the summary and discussion/conclusions that the method is « more accurate », in other words but the results on uncertainty does not quantify this improvement.

Perhaps it would be better to be more specific about the use of the term "more accurate" and we should specify in exactly which scenarios we believe this is the case.  Here, we use "accuracy" to mean "closer to the true state". If we assume the radiosonde is the true measurement, then

by "more accurate" we are saying that we produce a profile which is closer to the radiosonde measurements (or the percent difference between them is closer to 0).
In the case of the homogeneous nights, the trajectory method does just as well as the traditional in terms of matching the radiosonde profile and the uncertainties. However, in the heterogeneous nights, the method does improve the differences between the lidar and radiosonde and does more accurately reproduce the radiosonde measurement by up to 20% (Figure 7). In this case, we do not use the term "accurate" to refer to the uncertainty of the method, but with respect to the shape of the radiosonde profile.
With respect to the propagated uncertainties, it is true that the traditional and trajectory methods do produce similar uncertainties, therefore, it would not be correct for us to use the word "more" when referring to the uncertainties of both methods. We will re-examine the wording in these sections to make sure that the discussion is fair to the traditional technique.

The discussion/conclusion about the advantages of the method, it must be thorough:
5.1 Almost no discussion about the limitations of the method: 32 % of night are calibrated (what about the others?) on 6 years (the first 3 years not being calibratable also)
5.2 The benefits presented are based on the theoretical expectations that motivated the implementation of this methodology. The uncertainty is presumably better but this is not reflected in the budget calculation.

We agree that there should be a clear discussion of the limitations of the method. However, we would suggest that removing nights due to bad weather is not a limitation that is inherent of the method, but is due to the signal attenuation of the lidar measurements on cloudy nights which, depending on the height of the cloud, could remove a large amount of the altitudes available for calibration. It is not preferable to calibrated during cloudy conditions because we do not measure the lidar ratios inside clouds, additionally clouds are not uniform and signal level would not be consistent over all scans.

We agree that there should be a clearer discussion of the traditional uncertainties and will include the appropriate references inside the uncertainties discussion, such as Wandinger 2005 and uncertainty discussions from Leblanc et al. 2012. We have answered the specific comments regarding the uncertainties individually in the "Specific Comments" section.

5.3 Discussion should be pushed before opening the perspective of using this dataset to « UTLS climatology over 10 years »: how to do that with 24 nights above 2008-2016, how to calibrate night where this methodology could not be applied? It should have thought because the authors wants to use the whole dataset for trends study.

We agree that the discussion should be expanded to include the limitations of the method. However, we would prefer to refrain from discussing the details and the methodology of the trend analysis in this paper (particularly in the conclusions/future work section) since the subject of this paper is not the trend analysis, and is the calibration technique itself.

RALMO has bi-weekly RS92 radiosondes launches with which this method can be used. We did not include them in this study because these radiosondes do not report uncertainties as a function of altitude and thus, should be considered secondary, not primary calibration opportunities, like the coincident GRUAN sondes. The non- GRAUN sondes will be used along with the GRUAN sondes and other calibration techniques when forming a climatology of the RALMO measurements.

**Specific comments:**

Comment: *Title*
I strongly suggest to refer to RALMO in the title or to find a way to indicate that it is a test of this new methodology of calibration done on one lidar which could potentially be applied to others.

We disagree with this suggestion to add RALMO to the title. There is nothing in this manuscript specific to RALMO. The limiting factor in this method is in fact only due to the radiosonde and whether or not the final data reports wind speed and direction. We will add a sentence in this regard to the Abstract and the Conclusion.

Abstract sentence: This trajectory technique is a general technique which may be used for any lidar, and only requires that the radiosonde report wind speed and direction.

Comment:
Abstract – page 1
Lines 1-4: I would keep theses sentences for an introduction because it is too general. At least,please shorten this part.
The authors disagree. It is necessary for some introduction to understand the context of the study, even for experts in the field.

Lines 5 & 6: Reference citations should not be included in this section unless they are essential. Using radiosondes is the most used technique for calibrating so please select a maximum of one reference, I would suggest Whiteman et al. (2006) which is the closest from the one you will use?. Maybe, the method is even better summarized in Whiteman et al. (2012). -

While we agree that references should be kept to a minimum, all three papers used similar techniques and we feel that it is appropriate to acknowledge their involvement and to make it clear that the work is built off of three different papers. It would not be fair to the other authors to not include them as well.

Line 7 « movement of radiosonde »: I suggest replacing it with « movement of air masses »
We have changed the sentence to: "However, they did not consider the movement of the radiosonde relative to the air mass and frontal boundaries"

Line 12: Precise on which period the calibration has been performed (i.e. 2011-2016). **-**
We have added "from 2011 - 2016" to the sentence on Line 12.

Lines 14-15: The authors use « more accurately » but there is no conclusion in the article that quantifies that the uncertainty associated with the new technique is better than the « traditional » one . I suggest replacing « reproduces more accurately » with « reproduces accurately ».
It is true that the method does more accurately reproduce the radiosonde profile and decreases the differences in the profiles by up to 20%. However, to avoid confusion over the word "accurately", we have changed the sentence to say:

" We show that the trajectory method reduces differences between the radiosonde and lidar by up to 20% when the water vapour field is not homogeneous over a 30 min calibration period."

Lines 16-21: The summary associated with the uncertainty budget is too detailed. Please replace this part by one value (or range of value) quantifying the total uncertainty associated with the calibration.
We have removed the sentence regarding the deadtime uncertainty to shorten the section and only include the most important result, which is that the uncertainty in the calibration due to the uncertainty in the radiosonde is 4% of the calibration value.

**Page 2**
Line 2: Please replace « the primary contributor» with « one of the main contributors ».
We have changed this as requested.

Line 3: Add reference to « ...high temporal and spatial variability »
The authors believe that this a common knowledge statement and is generally stated throughout the water vapour community. However, we can add references to a few papers (Trenberth et al. 2005, Ross and Elliot 1996, Kampfer 2013).

Line 4: I suggest deleting « uniquely ».
 We have replaced "uniquely" with "well".

Line 6: I suggest deleting « more » or please add a reference.
We have removed the word "more".

Line 8: Replace « take » with « make ».
We have changed this as requested.

Line 9: Delete « also »**.**
We have changed this as requested.

Line 10 « Several Raman... external methods »: I would place this sentence in the following

Paragraph.

We would prefer to keep that sentence where it is because it is necessary to explain why a calibration constant is necessary for water vapour lidars and its importance in a trend analysis.

Lines 17 to 22: It's too detailed whereas it is not the main subject of the article. I suggest deleting this part.

We have removed two sentences from this section and condensed them into one sentence.

Line 29: Please add the use of GNSS as an external instrument to calibrate Raman lidars and a reference. I suggest: David, L., Bock, O., Thom, C., Bosser, P., and Pelon, J.: Study and mitigation of calibration factor instabilities in a water vapor Raman lidar, Atmos. Meas. Tech., 10, 2745-2758, https://doi.org/10.5194/amt-10-2745-2017, 2017.

We have added this paper and mentioned GPS satellites to the first sentence on the paragraph detailing standard external measurements.

**Page 3 Comments:**

Lines 1-2 « External...do not contribute »: I suggest deleting this sentence. -

This sentence is necessary for the reader to understand why some lidar stations prefer to calibrate externally rather than internally.

Line 7 « as RS92 radiosondes are the most frequently used calibration radiosondes Immler et al., 2010 ; Dirksen et al., 2014). »: I'm not convinced that the main objective of this article was to correct sondes for the calibration of Raman lidars. Please rephrase.

While the purpose of this article was not to correct sondes, this sentence is necessary to explain why we choose to use GRUAN sondes for calibration instead of the uncorrected RS92 sondes. Additionally, it shows that we are aware of the disadvantage of using radiosondes, but have tried to reduce that problem by using the most accurate product. Therefore, we believe this sentence should not be removed.

Line 8: Replace « Vaisala » with « not corrected »

We have changed this to "uncorrected RS92"

Line 9: Replace « errors » with « uncertainties »

We have changed this as requested.

Lines 9-10 « A portion of... uncertainty »: I suggest moving this sentence in the next paragraph, line 17 before « This paper attempts to... ».

We have moved this sentence as requested.

Lines 12-15: I suggest moving the whole paragraph line 27.

The paragraph starting on Line 28 directly follows from the previous paragraph and moving the paragraph on line 12 would break the flow of the section. Therefore, we would prefer to keep it where it is.

Line 19: Please explain in few sentences (2-3) what is the « traditional » method, references are not sufficient considering that the improvement of this technique is the main subject of this paper.
We agree that adding a few sentences discussing the traditional method would be helpful for the reader. As mentioned in the general comments, we believe that we have answered this sufficiently in two ways:

1) A new section for the Traditional Method has been added.
2) Text introducing the traditional method has been added and is shown below:
   a) New Text: " The ``traditional" method for calibration water vapour lidars is done by integrating a fixed number of lidar profiles as a function of height starting at a time which is coincident with the radiosonde launch and then calculating a linear weighted least-squares fit between the radiosonde and lidar measurements to determine the calibration constant \citep{Melfi1972, Whiteman1992}. The altitudes over which the fit is conducted are either fixed (e.g. always 1 - 5\,km), or the optimal altitude region may be determined by calculating the correlation between the radiosonde and the lidar measurements. For the purposes of this paper, we refer to the traditional method as using 30\,min of integration with a weighted least-squares fit over altitudes determined by maximum correlation. "
3) The previous sections 3.2 and 3.3 have also been added to this section in regards to another comment so that the entire traditional method is clearly discussed and the reader can then better understand our implementation of the method.

Line 20 «as the radiosonde takes approximately 30 min to reach the tropopause »: It depends on which latitude the sonde is launched, it will be larger near the equator. Please specify it or add a location.

We have added the clarifier "at mid-latitudes".

Line 21: Your statement should be supported by a reference or some statistics from your database.

We agree that we should state where these numbers came from. We also thank you for drawing our attention to a wording mistake - it should not say "up to" as this implies a maximum. 4 km was in fact the minimum distance traveled. This statement is based on statistics from our study and will be rephrased as follows:

… during which time radiosondes in this study drifted a minimum of 4 km away from the lidar's field-of-view ….

Line 28 « in order to ensure that the lidar and the radiosonde are measuring the same air »: It was also the goal of the « traditional » method. Please check.

Yes, this was also the goal of the traditional method. We will rephrase the sentence as follows:

Original Sentence:
"In order to ensure that the lidar and radiosonde are measuring the same air, we have developed an improved lidar-radiosonde calibration technique that utilizes the position.."

New Sentence:

As in any atmospheric calibration method, it is important that the instruments involve measure the same air mass. To improve the coincidence for periods where calibration is required but the atmospheric water vapour content is changing, we have developed an improved lidar-radiosonde calibration …"

Line 28: I would suggest deleting « improved ».

We believe that changing the sentence as above fixes this issue.

Line 29: Replace « of the radiosonde and the » with « , ».

This would not be correct since "wind" applies to both the speed and direction and therefore requires an "and" beforehand instead of a comma. We can change the sentence to:
"... we have developed an improved lidar-radiosonde calibration technique that utilizes the position of the radiosonde and the radiosonde wind measurements."

**Page 4 Comments:**

Line 7: Add « in Payerne » at the end of the sentence.
We have changed this as requested.

Line 9 « their respective uncertainties »: The full uncertainty budget for the calibration is not given for the « traditional » method. What about the « representation uncertainty » for example? I will come back to this in more detail in following comments.
We agree that we did not discuss this in enough detail for the traditional method. Indeed, we did calculate the uncertainties in the same way for the traditional method as we did for the trajectory, however, we did not make it clear enough for the reader. We will clarify this in the uncertainty section and in the summary flow chart. We have also added the uncertainties to Table 1 so that the reader can see that they are similar.

Line 13: Does the lidar always start working at 0:00 UTC? Does « bi-weekly » refer to twice a week or one every two weeks? Please specify it in the text.

Since this section is for the radiosonde, it would not be appropriate to put lidar details here. RALMO is an operational lidar and operates 24/7 so long as it is not raining or undergoing repairs/routine maintenance. We will add the sentence to Page 5, Line 3- "As an operational lidar, RALMO runs 24/7 except when the cloud cover is below 800 m". Bi-Weekly refers to every other week, which we have added to the text.

Line 14: Why is their only « a subset » of these radiosonde processed by GRUAN? Please explain it in the text.

Unfortunately, not every RS92 radiosonde launched at the Payerne station is GRUAN compliant and many are launched for only internal studies, which is why only a subset may be processed by GRUAN. We will change the sentence to "Only a subset of these radiosondes are processed by GRUAN because not every RS92 flight before 2019 was GRUAN-compliant."

Line 23: Please specify that the analysis was conducted on an initial set of 76 flights but in the end only 24 of them were used.

We have edited the last sentence of section 2.1 to say: "A total of 76 GRUAN RS92 nighttime flights were initially used to conduct this analysis, however, due to clouds and lack of coincident lidar data, only 24 flights were used in the end for the calibration."

Page 5 Comments:

Section 2.2: Precise that the RALMO is operating day and night. Give the effective measurement time (« 50% » in the conclusion, it should be precised earlier) and what explain that 50% of the time is not exploitable.

We have added this sentence to the end of the first paragraph in section 2.2 to make this more clear: " RALMO runs day and night with an average of 50% uptime from 2008 - 2017. RALMO downtime is due to the presence of clouds below 800 m, or repairs/routine maintenance."

Line 2: The authors explained that the instrument « is designed to be an operational lidar, and as such, needs to have high accuracy, temporal measurement stability, and minimal altitude-based corrections (Dinoev et al., 2013; brocard et al., 2013) ». The study of the instrument's performance made in the bibliographical references and years of operation should determine whether the instrument really has a high accuracy, temporal stability of measurements and profiles that start close to the ground. Please be more specific.

You are correct, this manuscript is part of a broader study of the instrument one of us (SHJ) is performing as part of her PhD thesis work. We have amended this sentence to: "RALMO was designed to be an operational lidar and therefore was designed to have high accuracy, temporal measurement stability, and minimal altitude-based corrections."

Lines 4-6 « RALMO operates...scattering channels »: Move this sentence to the previous paragraph.

We have moved this sentence as requested.

Line 9 « a sufficiently high SNR »: This is a major issue in this paper. The SNR is often used to select some data and reject others, but no threshold based on bibliographic references or empirical tests specific to this study is defined. You need to clarify this.

We agree that this is not well defined throughout the paper and we will fix this by providing specific SNR values, or remove the discussion of SNR where it is not appropriate. In this case, the second half of that sentence is not appropriate for this section and we will remove it. Only a background value threshold of .01 photon counts/bin/s is applied at this step in the process.

Lines 12 & 13 « After the filtering process...small features »: I wonder if this sentence should not be found in the description of the calibration methodology. Or at least move the interpolation part of the radiosonde profiles to section 2.1 and specify here that there is no interpolation for lidar profiles.

We have moved the sentence to the radiosonde section and have now specified that there is no interpolation of the lidar profiles at the end of section 2.2.

Line 15 « 3.1 Tracking air parcels »: I would move this title at the end of the page. Paragraph line 16 to 20 is more an introduction to sect. 3.

We have moved the first paragraph before section 3.1 as we agree that it is more of an introduction to Section 3.

Page 6 – Figure 1
I would use « homogeneous region » instead of « homogeneous cylinder » to be more consistent with the text. I would suggest adding letters or numbers to the different steps and refer to them in the text.

We will change this to agree with the rest of the text in the paper and use "homogeneous lidar region" instead.

**Page 7 Comments**

Line 7: As simple that it could be, you should provide mathematical explanation/an equation to illustrate your calculation.

This conversion is standard and can be easily found online as the local, flat earth approximation which is appropriate for distances smaller than 20 km. We have used the reference website used by NOAA for our calculations: http://www.edwilliams.org/avform.htm#flat. The theory behind the transformation is discussed in
Smart, W. M.: Textbook on Spherical Astronomy 6th Ed., edited by R. M. Green, Cambridge University Press, Cambridge., 1977.
The authors have also verified this through derivation, which if necessary can be added here. We would be happy to include the citation in the text to guide the reader to the appropriate material, however, we don't think it would be appropriate to add the conversions to the paper.

We will add to the text that the conversion can be assumed using the "local, flat Earth conversion for a spherical Earth" which should provide enough context for the reader to find the appropriate conversions.

Line 8: This assumption should be discussed a little more. You might provide a physical discussion about when (or if) this assumption is (would be) realist and its limitation, please cite references that could support this hypothesis or this discussion.

As this method is not used at all in this paper because all of the 2008 - 2011 radiosondes were removed due to the presence of clouds, we will remove this sentence from the paper. However, if you are curious about why this assumption is reasonable  please see pg 47 of :
Daidzic, N. E. (2017). Long and short-range air navigation on spherical Earth. International Journal of Aviation, Aeronautics, and Aerospace, 4(1). https://doi.org/10.15394/ijaaa.2017.1160
Appendix D, demonstrates  that assuming plane trigonometry is acceptable for short distances on Earth (20 km or less). The radiosondes in this study traveled between 4 and 18 km away from the lidar during a 30 minute calibration, therefore, the assumption is valid here. Additionally, since we are calculating the distances between each radiosonde measurement, this is certainly valid at distances on the order of meters.

Line 9: What do you mean by « We do not explicitly consider the vertical movement of the air parcel in this method »? From your description of the methodology, I understand that you do not consider at all the vertical movement. Please be more specific.
We agree that the word "explicitly" is confusing here. We have removed it because we do not consider the vertical movement.

Line 14: For my point of view, the « lidar region » refers more to the ~1 m diameter at 5 km that the radius of 3 km choosen after for the sensitivity test. You might use a name more related with the assumed homogeneity of the water vapor like or its use for calibration, « homogeneous lidar region » or « calibration region » as you call them after in the paper.

Thank you for catching this. That is in fact a typo and should be "homogeneous region". We have redefined this as the "homogeneous lidar region" as suggested.

Lines 14-15: What is the decisional parameter? Is this the SNR? Which threshold?
The decisional parameter in this case was the SNR of the water vapour channel. Radii smaller than 3 km resulted in SNRs smaller than 2 below 7 km and in some cases halved the SNR of the water vapour channel at altitudes below 5 km. Therefore, a radius of 3km was chosen to maximize the SNR in the lower altitudes without creating too much smoothing of features and to maintain SNRs of at least 2 above 7 km in the majority of cases.

We will reword the paragraph as follows for clarity:

"In order to maintain a SNR in the water vapour greater than 2 above 7 km altitude for the majority of the cases, we defined the homogeneous lidar region to be a circle of 3 km radius centered around the lidar. The size of the homogeneous region was chosen by varying the radius from a range of 1 - 25 km and finally increasing it to infinity. Radii below 3 km resulted in SNRs smaller than 2 below 7 km and in some cases halved the SNR of the water vapour channel at altitudes below 5 km, which decreased the altitude coverage for the calibration and increased the noise in the primary calibration region. While radii above 3 km resulted in SNRs larger than 2 above 7 km, the water vapour profiles started to exhibit biases due to using too long integration times at certain altitudes and losing small features which had previously been visible. The 3 km radius provided the most altitude coverage with profiles closest to the radiosonde measurements and was the best compromise."

Lines 16/17/18: You characterized some « very low SNR », a « large enough SNRs » and « the highest SNR ». Please quantify or explain why you define it this way.

The above paragraph is intended to fix both this comment and the previous comment.

**Page 8 Comments**

Figure 2 : Is this a conceptual scheme or a real example? Please precise it in the legend. If it is an example, give the date of the measurement. The second sentence of the legend is also explained in the text. Please replace this sentence with something like « The purple circle corresponds to the lidar region».
This is a conceptual example, which we will specify in the caption. We would prefer to keep that sentence there for readers who choose not to read the text and only go through the figures.

Lines 6-7 « The standard thirty minutes...tropopause »: It sounds quite general in your text but it corresponds to mid-latitudes. It would be a different duration for polar regions or in the tropics. Please be more specific by adding « at mid-latitudes » for example.
We have added the specifier " at mid-latitudes".

Lines 7-8 « Integrating...by radiosonde »: I suggest deleting this sentence.
We have removed this sentence as requested.

Lines 9-10: The analysis of Figure 3 consists in one sentence. Either your analysis is too short, either the figure is not necessary. Please reconcile.

We would be happy to increase the discussion of Figure 3, as we believe that it helps the reader understand how the integration time changes with altitude. We would add the following text.

"The decrease in integration time with altitude will also change depending on the rate at which the radiosonde moves away from the lidar. The majority of nights had integration times less than 5 min above 7 km. However, if the wind is strong at a particular altitude, sharp decreases in integration times may be seen as the radiosonde moves quickly away. It is also possible to see the integration times decrease and then increase again as the radiosonde drifts in and out of the homogeneous lidar region."

**Page 9 Comments**

Figure 3 Legend: « The integration time...analysis. »: Please remove these sentence from the legend. It is already explained in the text or is part of the analysis.

We would prefer to leave these sentences in the caption to aid readers who do not read the text. It is common for readers to use figures from a paper in presentations or discussions, and we would prefer to keep the analysis in the figure caption so that they have it readily available.

Figure 2 – Figure 3 – Figure 4

In this section, Figure 2 seems to be a conceptual method, Figure 3 refers to July 21 and Figure 4 to July 22. It is not explained why these 3 dates are choosen for each example. Considering that Section 3 details the method of calibration by sonde, you might choose the same date to illustrate the different aspects of it. If not, please justify why.

July 21st is a typo, thank you for catching that. Only one date has been used for these figures - July 22nd.

Section 3.2 - Pages 9 and 10
I suggest moving this section in or after Section 2.2 because it is more about the « Lidar measurement » than about the « Radiosonde Trajectory Method ».

We agree that technically this is more related to the lidar measurements section. However, to address your request for a better description of the traditional method, we have created a whole new section for the traditional method and put it there. This way the traditional method is clear and we only discuss the differences introduced by the trajectory method.

**Page 9 Comments**

Lines 10-11: I would rephrase this way: « The central wavelengths of the water vapour and nitrogen channels of the RALMO were choosen to minimize temperature dependence. »
We have changed this as requested.

Line 14: Please add a reference.
We have added the citation of Whiteman 1992 as requested.

**Page 10 Comments**

Lines 4-5 « In RALMO's case, the ratio...2014). »: Please move this sentence to Sect. 2.2. -
We have moved this entire section to the new traditional method calculation Section 3.1.

Line 8 « the corrected signal »: Which correction? Please be more specific.

By "corrected" we mean "transmission-corrected and cloud-filtered lidar signals". However, I think it is not necessary to mention corrected here. To be clearer, the sentence has been changed to : " After calculating the un-calibrated mixing ratio profile from the ratio of the two lidar signals (Section 2.2), we use a …..".

Line 9: Does « the correlated and weighted least squares fitting » correspond to the « traditional » method? If so, why not call it this way in the abstract so that you don't list so many references? It could be included in the introduction in this way as well.

Yes and no … the correlated and weighted least squares fitting does not refer to the traditional method - although our implementation of the traditional method does use these techniques to fairly compare with the trajectory technique. The traditional method refers to the selection of scans used to calibrate the lidar.
 We believe we have fixed this problem by including this section in the new Traditional method section.

Line 14 « low SNRs »: Please specify.
Water vapour scans integrated less than 5 minutes typically have SNRs less than 2.
We will rephrase the sentence as follows:
".... due to SNRs less than 2 …. "

Line 20: I suggest replacing « lidar region » with « calibration region ».

Thank you for your suggestion. We agree that the term needs to be consistent and appropriately descriptive for the reader. However, the term calibration region is not the best because it could refer to the altitudes at which we calibrate, which is not the case. Therefore, we have decided to use the term "Homogeneous lidar region" throughout the paper and will change all accordingly.

Section 3.3 – Pages 10 and 11
It seems to be the traditional method that is mainly described here except that the lidar data are selected as described in Section 3.1. Be careful to distinguish in the text of this part (and even in the entire article) between what is specific to your study and what is traditional. Perhaps the structure you have chosen is a little confusing on this point.

We have added a few sentences in Section 2.2 discussing the traditional method to make sure the differences are clearer to the reader. Thank you for your suggestion regarding this issue.

**Page 11 Comments**
Lines 2-10: I suggest moving this part to Sect. 2.1 or at least be more concise.

We have moved this section to the new traditional method section. Unfortunately, we do not see how it can be made more concise as it is necessary to explain the reason for the average pressure profile and why we believe the assumption is justified.

Line 12 « for July 22, 2015 »: Why did you choose this date instead of July 21 as in Figure 3? That would seem more consistent.

Thank you for catching this error. This is indeed the same date. The time formats were changed from local time to UT time in the paper at the last minute and we missed some of the changes.

Line 15: Please replace « error » with « uncertainty ». - In this case, standard error is a specific term. What to do here?

Some sources will use the term standard error, but since we are following Bevington's derivation, we will replace the term "standard error" and simply use the "uncertainty of the slope" to avoid confusion here and to keep terms consistent throughout.

Line 15: Please add a reference. What was the range of uncertainty found in the litterature for the « traditional » method?

Thank you for pointing this out. We have added three more paragraphs to the uncertainty section discussing the typical way that the uncertainties are calculated for the traditional method and have cited several papers as examples. We had discussed this very briefly towards the end of the section, but the other referee thought it would be better to make this distinction in the beginning as it was not clear to the reader. To answer your question, the typical traditional calibration uncertainty value is between 4 and 5%, but is calculated differently (the details of which are discussed in the new version of the paper).

Figure 4 – Page 11
The date is not specified, please do it. The last sentence is not necessary.

We will specify the date - it is 22 July 2015. The sentence is necessary for readers who may choose to not read the text and it is necessary to explain the uncertainty value shown in Figure 4 b.

**Page 12 Comments**

Line 3: What do you mean by « repairs »?
In this case "repairs" really means routine maintenance such as alignment, changing the flashlamps, etc. We have clarified this in the paper.

Line 4 « abnormaly high »: Please quantify or explain.
We have clarified this sentence to say "abnormally high background values above 0.01 counts/bin/min".

Line 6: I suggest: « and presence of clouds ».
We have changed this as requested.

Lines 6-7 « The filtering process...radiosonde launch »: This sentence seems to say that there were clouds every night of measurement for all the duration of the measurement. Is that what you mean? Yes, that is exactly what we mean.

Does this mean that there are 3 years of measurement that cannot be used because of systematic presence of clouds?
No, there are other radiosondes available with which we can calibrate during those 3 years. However, there are no RS92 radiosondes during that period that were launched on clear days.

Line 8: If I understood correctly, between 2008 and 2016, there are only 76 calibratable nights and they are all condensed over the period 2011 and 2016. What solution for 2008-2010? On the other hand, over the 76 nights, with the implementation of this methodology, only ~30% can be calibrated? What about the other nights of measurements? It is essential to discuss these aspects in Section 7.

 We think there has been some confusion over the goal of this paper. This paper is meant to present a calibration technique, which we have implemented using GRUAN RS92 radiosondes. We cannot calibrate a water vapour lidar on cloudy days, therefore, the weather has removed the majority of the radiosonde flights from *this* study. This is not a problem with the method itself, but an unfortunate happenstance caused by nature. It is possible to calibrate with non-GRUAN RS92 radiosondes and there are also operational radiosondes which are launched from Payerne daily which would increase the number of calibration points, but as we stated earlier, these are secondary sources and not primary sources for calibration and therefore were not included.

**Page 13 Comments**

Table 1: Column « Difference »: Sign of the difference is missing or it should be specified that it is absolute value.
Thank you, we will specify that it is the absolute value.

Legend: I suggest deleting the word « calibration » in the second sentence. The text from « Two nights in the homogeneous » to « variability in the water vapour » should not appear in the legend.
We can remove the word "calibration" in the second sentence. However,  we would like to keep those sentences in the Table captions because they are necessary to explain the comments in column 6.

Line 6 « show a good agreement »: Please quantify or explain.
This sentence is a hypothesis of what we should see, therefore we did not quantify "good agreement". However, we did neglect to follow up with the results for the homogeneous section which were included in the Summary instead. We have moved the last paragraph in the section into this paragraph to help quantify the agreement between the traditional and trajectory methods on homogeneous nights. Hopefully reorganizing the paper in this way will make it more understandable to the reader.

Line 7: I suggest adding « water vapor conditions around the location of the lidar measurement ». We have changed this as suggested.

Line 8 « as « homogeneous » or « stable » nights in Table 1 »: Only « homogeneous » is used in Table 1 in comparison with « heteregenous » , « stable » is used in the « Comments column » for heterogeneous nights. Please reconcile. - We agree that this is not consistent. We will remove the word "stable" since this is confusing and only use "homogeneous".

Line 9 « an average bias of 0 % »: Please used absolute value to calculte the average bias. Maybe you should use the standard deviation and specify if the bias is more positive or negative.

Perhaps the best way to discuss the differences between the trajectory method and the traditional method is not in terms of biases, but in terms how well it removes the large differences between the radiosonde and lidar. For example: In Figure 7, on 2012-07-27, the difference between the radiosonde and the lidar produces sharp differences on the order of +/- 15% - 20%. However, these differences are reduced by 10 - 15% in the trajectory method. By discussing the features individually, and discussing the changes in the profiles, we believe this will clarify the problem of the bias calculation in addition to increasing the discussion of each of the figures. Please refer to the revised manuscript for the added text as there is too much to include here.

**Page 14 Comments**

Line 1 « The bias on that night is reduced when using the trajectory method »: What do you mean? Please rephrase.

We believe our answer to the previous comment has also answered this comment.

Lines 2-3: Figure 5 is composed of 9 subplots and show results for 3 dates but there is only one sentence that refers to it in the text. You need to analyse your figures or do not put them in the paper.

We agree that we should discuss these and all figures in more detail. We have included 2 paragraphs of discussion for Figure 5 in the new manuscript and would refer you there.

Figure 5 – Page 14 Legend - 4th sentence: Why do you mention « White vertical regions » whereas there is none of them in the figure? Please delete this sentence.

Our apologies, that sentence should indeed be removed from the caption. Thank you for catching the error.

Legend - 7th sentence: You should use « magenta » instead of « pink » or change the color of the corresponding line in Figure 5.

We have changed this as requested.

Legend - 8 and 9th sentences: It is part of the analysis and should not appear in the legend. Please delete them.

As stated previously, we would prefer to keep these sentences inside the caption for readers who choose to skim the paper. We will make sure these sentences are also included in the text.

Figure 6 – Page 15
There is no 2012-07-27 measurement in Table 1. Is this the right date? Please check.

Thank you for noticing this discrepancy. It is the right date. The date formats in the paper were changed at the last minute and the dates in the table were not changed to UTC time. The dates in the table have now been updated to the correct format.

Legend – 1st sentence: I suggest deleting « launch at 0 min ».

We have changed this as requested.

**Page 15 Comments**

Lines 1-5: This part explains the method to characterize a night as homogeneous or heterogeneous, it should come earlier in this section; before discussing about the results. Please reconcile.

We agree that this should come earlier in the section and have changed this as requested. This section has been reorganized as per your general comment at the beginning.

Line 3: Figure 7 is quoted before Figure 6 (p16 - l.11) in the text. The order of the figures is not Respected.

Thank you for noticing this. We have rearranged the figures in this section such that Figure 6 now comes before both Figure 5 and Figure 7 as we feel that this is a clearer way of discussing the comparisons.

Line 3: As for other figures in the article, Figure 7 is commented in only one sentence.

We will increase the discussion of this figure in the paper, as you suggested.

Line 8: See my previous comment on the average bias.
Please see our previous answer. We will increase the discussion of Figure 7 to include how the trajectory method reduces the maximum differences between the radiosonde and the traditional method lidar profile.

Line 9: Please refer to a figure or a table.
We will refer to Figure 7.

Line 14: Why « 11 » whereas there are 12 heterogeneous nights in Table 1? - Thank you for seeing catching this error. That was a typo and should have said 12. Note that we have now amended one of the previous heterogeneous nights and moved it to the homogeneous section. Therefore, there are now 11 heterogeneous nights.

**Page 16 Comments:**

Figure 7 – Why did you choose these three nights? You must analyze all three of them, when you add a figure it should bring an supplementary information, otherwise it means that the figure is not necessary.
We agree that these nights could be analyzed in more detail. We chose these nights because they were prime examples of the heterogeneous nights. We have added another paragraph discussing this figure in more detail to the new manuscript.

Legend – 2nd to 6th sentences: This is already explain in legend of Figure 5, please just leave the part on the white vertical regions and refer to legend of Figure 5.

We can refer to Figure 5's caption. Thank you for the suggestion.

Legend – 7 and 8th sentences: Again, this does not have to appear in the legend.

As per our previous comments, we would prefer to keep these sentences inside the caption.

**Page 16 Comments**

Lines 1-2: Shouldn't these nights be described as homogeneous in this case? Does this not question the methodology of homogeneous versus heterogeneous characterization?

These nights were defined as heterogeneous nights because they had features that changed by over 50% at a given altitude over the 30 minutes of lidar measurements. Therefore they were defined as heterogeneous nights. However, the correlation algorithm determined that the regions which had over 90% correlation with the radiosonde where the regions which were homogeneous over 30 minutes. This is not always the case, therefore we would consider the heterogeneous vs. homogeneous distinction valid in general. What it does suggest, is that the regions which we choose for calibration are important.

Line 4: The average bias is around 1 % here whereas it was 0 % earlier (p15 - l. 8). Please reconcile.

In the previous case, the average bias was referring to the plots in Figure 7, not the average for the entire data set, which is the case here. We will clarify the language in both sentences so that it is clear which figures we are referring to.

Line 5: Please specify what threshold you chose to define that the variability increases above 5 km.

Thank you for catching this, it should in fact say 4.5 km and is referring to the increase in the standard deviation of the percent differences in Figure 8. However, it is not clear on the scale of this figure, therefore, we will increase the x axis to 100% STD for clarity.

[Figure]

Line 6: Here you precise the sign of the bias whereas you have not done it earlier, same for the significant number of digits. Please reconcile.

In this case we are discussing the average of the entire data set, where previously we were discussing the bias of a specific date. This is not clear in the paper and we will change the language such that the reader can easily determine which is which.

Line 7: Please rephrase the sentence.
Original:
        The standard deviation of all of the percent difference profiles shows that the trajectory method more accurately fits the radiosonde profile above 2 km on a profile-by-profile basis and will more consistently provide better fits.

Revised:
        The standard deviation of the ensemble of percent difference profiles between both calibration methods and the radiosonde shows that the trajectory method more accurately matches the radiosonde profile above 2 km.

Line 8 « better fits »: Please quantify.

The standard deviation of the trajectory method is on average 10% smaller than the traditional method for altitudes above 2 km and below 4.5 km. This implies that the profiles more accurately fit the radiosonde on a profile-by-profile basis. We believe that the new sentence above fixes this problem.

Line 11: Once again the commentary on the figure does not exceed one sentence. Please develop or remove the sentence.

We apologise that this was not clear, but the next few sentences also referred to Figure 6. We have rectified this in our restructuring of this section.

**Page 17 Comments:**

Figure 8 -I suggest choosing one reference (either the trajectory or the « traditional » method) to avoid the multiplication of the data and to put the average bias and the standard deviation on the same plot.

Thank you for your suggestion. We would prefer to keep both data sets on the plots so that the reader can easily see how they differ from each other.

Starting Page 16 -line 10 to Page 17 – line 7: This is a description of the methodology and should come earlier in the section. Please put this explanation after the introduction of Table 1. We agree that this discussion should come sooner and have moved it as suggested.

Lines 3-4: What do you mean by « the majority »? How many nights? In Table 1, it seems that it is true for 6 nights on 12. This represents half of the data not the majority.
You are correct, the percentage should say 1% to encompass the majority (10 out of 12).

Lines 4-5: Please specify the dates
We have specified the dates as requested.

**Page 18 Comments:**

Lines 2-7: You repeat yourself. This was already mentioned earlier in the article.

We agree that this is a repetition, however, we feel that it is necessary to remind the reader of how these are calculated as they may not remember and it is not useful to refer back to a previous section.

Lines 17-18: Please precise how the fitting uncertainty is calculated.

This is a standard calculation which is easily referenceable. Therefore, we will reference the Bevington textbook so that the reader has a source for the equation. However, we don't feel that it is necessary to include the equation in the paper.

**Page 19 Comments:**

Line 2: Which part is the systematic part versus the statistical part of the uncertainty? This is important to determine given that one of your objectives is to establish trends in the UT/LS.

The radiosonde uncertainties are the total uncertainty resulting from the combined Statistical and Systematic uncertainties for Relative Humidity, Pressure, and Temperature and are calculated by the GRUAN uncertainty algorithms (Dirksen et al. 2014). We will clarify this sentence and change it to say: "The uncertainty of the water vapour calibration constant due to the lidar's random uncertainty and the radiosonde's total uncertainty (both systematic and random) was determined ...."

Line 5: Please explain what do you mean by the « measurement vector » with regard to the lidar Measurement.

The measurement vector is a vector which includes all of the lidar and radiosonde measurements, e.g. - MV = [L1, L2, L3, …. R1, R2, R3…] where Li is a lidar measurement and Ri is a radiosonde measurement.

Line 12: Please correct the indices of the third term of Eq. (5)
We have changed the indices in the third term to $R\_j$ instead of $R\_i+1$. Thank you for catching this discrepancy.

Lines 13-25: The comments of the equation is not clear enough. First you should describe each term,precise the source of each uncertainty (radiosonde measurements, lidar (including lidar's photon counting and deadtime and covariance term),  quantify the uncertainties before (« 8 % » coming to late in the next section, ?, 5 %, calculation?) and after propagation (values of Table 2) and you should use Table 2 to support this section.
   1. We have described each term for this equation directly after the equation, and before the equation since it uses many of the same variables as Equations 3 and 4.
   2. The sources of the lidar and radiosonde uncertainties were previously discussed in the manuscript in the first paragraph of Section 5 (now Section 6). , However, we have not discussed the quantities of each and can add them to the same paragraph.

Actually, Table 2 comes too late in the paper.
   - We would prefer to leave Table 2 as a summary of the uncertainties, however, we will make sure to quantify the uncertainties used in the text more clearly.
 You should also precise how is considered your uncertainty on the calibration constant due to the radiosonde: statistical or systematic?

- The uncertainties for the radiosonde are the combined statistical and systematic uncertainties as calculated in Dirksen et al. 2014 and is what is reported in the GRUAN corrected radiosonde files. We had simply said "total uncertainties" in the paper but we will specify that they are a combination of all uncertainties.

Then you can conclude on the total uncertainty of the calibration constant for which you are found after propagation: an average value of 4%.

Line 16: Why do you mentioned that the fitting uncertainty is the same that lidar's photon counting uncertainty. As you do it l.24 to compare it to the deadtime uncertainty? Is there a physical or metrological explanation? The fitting uncertainty seems to be due to the methodology and is different from these two uncertainties.

The fitting uncertainty is the uncertainty in the fit calculated from the variance of the residuals. The variance is largely dominated by the photon counting uncertainty from the lidar measurements, therefore, the fitting uncertainty and the uncertainty in the calibration constant due to the lidar statistical uncertainty should be the same or very close. Indeed, upon inspection, we found that the two agreed within a tenth of a percent in all cases (page 19, Line 18).

We have changed the wording in the text, starting on Page 18, Line 18 to hopefully clarify the our definition of the fitting uncertainty:

Using the variance of the residuals of the least-squares fit, one can calculate the uncertainty in the fit, or ``fitting uncertainty." This fitting uncertainty is the result of the amount of photon counting noise in the lidar measurements, and can be treated as the uncertainty in the calibration due to the lidar photon counting statistics. The average trajectory method fitting uncertainty is 0.4% of the average calibration constant. The average fitting uncertainty for the traditional method is 0.3% of the average calibration constant. The traditional method has smaller fitting or statistical uncertainties than the trajectory method due to the larger number of scans used per altitude, on average, compared to the trajectory method. The fitting uncertainty does not encompass the entire uncertainty of the calibration constant, since it is largely due only to the photon counting noise. It is also necessary to examine the uncertainty of the calibration constant due to the uncertainties in the radiosonde measurement as well as the deadtime.

Line 23: I suggest not making a line break here and including it in the previous paragraph. '
- We have changed this as suggested.

Line 23 « we assume a deadtime uncertainty () of 5% or .2 ns »: How is estimated this uncertainty: literature? Test?
The 5% came from testing these nights using the OEM retrieval by Sica and Haefele 2016. We found that the standard deviation of the deadtime retrieval value was 5%. The choice was also a result of discussions with Licel who suggested that 5% would be a reasonable value to assume

on a nightly basis. Sica and Haefele 2016 tested various *a priori* uncertainties for dead time and found that they all gave similar retrieved dead times.

Line 25: Please remove « including dead time effects ».
We have removed this as requested.

**Page 20  Comments:**

Line 7 « The calibration time series »: Do you mean the 24 nights? Why don't you show them on a figure?
Yes, we mean the 24 nights used in this study - we have added this clarification to the end of the sentence. We originally had the time series figure to the paper but we felt it did not add enough to the paper and that there were already too many figures. We also thought it would detract from the discussion of the calibration.

Line 8: What do you mean by « de-trended »? Please explain.
It means that a trend line was fit to the time series and then removed from the time series in order to calculate the standard deviation of the calibration factors. We have added this clarification to the paper and removed the word "de-trended".

Line 8 « over ten years »: You work on the 2011-2016 (i.e. 6 years), why do you speak about 10 Years?
You are correct, therefore we have changed this to say 6 years instead of 10.

Line 17 – Page 20 to Line 14 – Page 21: This « Summary » part should not be a part in itself. These information should be in sections 4 and 5 and in the conclusion. It is necessary to rethink the structure at the end of the article, which for the moment consists in two parts: 6 Summary and 7 Discussion and Conclusions (see comments below).
We respectfully disagree with this opinion. We believe that a summary is a necessary component of a paper. We recognize that some of this information should also be included in sections 4 and 5, and that the summary section should reiterate those points. We would prefer not to remove the summary section as we believe that it helps readers who quickly read through the paper. By keeping the summary separate from the conclusions, readers can choose to skip this section if they so choose and continue on.

Line 23 « due to the lidar profiles »: Please rephrase.
We are not sure how you would like this to be rephrased. We have changed the sentence to say " … due to the difference in 1-minute lidar scans selected".

**Page 21 Comments:**

Line 4: Please change « error » into « uncertainty ».
We have changed this as requested.

Line 4 « is negligible »: Do you neglect it in your calculation? This uncertainty does not appear in Table 2. Please specify it in the text.

We have stated that the fitting uncertainty is analogous to the photon counting uncertainty and we have included it in Table 2 as the photon counting uncertainty. We did not think it would be necessary to include both the fitting and the photon counting since they are equivalent to within a tenth of a percent.

The magnitude of the fitting uncertainty/photon counting uncertainty is a tenth of the total uncertainty which is dominated largely by the uncertainty of the radiosonde and has a magnitude of 4%. When considering the components of the uncertainty "budget" of the calibration, the fitting uncertainty is negligible when compared to the uncertainty of the radiosonde.

We have changed this sentence to:
 "... although the fitting uncertainty is negligible relative to the uncertainty of the calibration constant due to the uncertainty of the radiosonde measurements (4% on average).

Line 5: This part is supposed to be a summary but this value « 8 % » appears for the first time. This should have been discussed earlier.

We agree that this should be discussed sooner. We will move it to Section 4.

All Sect. 6: This section contains many repetitions and brings many repetitions in relation to the whole article. There are key elements (such as the quantification of the uncertainty on the radiosonde of 8 % or Table 2) that should appear earlier in the article.

We agree that some of these sentences could be moved to the previous sections, however, as it is a summary section it is necessary to repeat the conclusions made in the previous sections. We would prefer to leave it in this format since it makes the paper more readable.

**Page 22 Comments:**

Line 1: This article lacks a real discussion part, so it is necessary to restructure it as follows:
6 Discussion
7 Conclusions
Line 2 « has several advantages »: What about the limitations? The following points should be discussed: no consideration of the vertical movement of air masses, only 6 years calibratable over 9 and only 32% of the 76 nights with exploitable radiosonde profiles according to the method's prerequisites. A real issue is: which method did you use for all other nights of measurements than the 24 selected? Because otherwise it means that over 9 years of data there are only 24 nights of calibrated and therefore usable data? We can't make trends over 24 nights.

Thank you for pointing out that it is definitely important to discuss the limitations of the method. We agree that a limitation of the method is it does not consider vertical movement. The fact that

only 24 dates out of the 76 original dates were useable was not due to the method itself but the poor weather that was coincident with the sonde launch.

Your concern for the trend analysis is correct, however, I do not believe it is a concern for this manuscript, which is describing a new calibration method which can be used independently of whether the lidar measurements will be used for trend studies. We do in fact have other radiosondes with which to calibrate in Payerne, but chose not to include them in this paper since we wanted to focus on using the GRUAN products which include detailed uncertainties for the radiosonde measurements. Calculating trends is important, and we are currently undertaking that study using the RALMO measurements.

Line 19 « an automatic [...] scheme »: What do you mean? Does this mean that this method has been implemented in a production chain for water vapor profiles of the RALMO lidar?
Yes, the method has been implemented for water vapour calibration for the RALMO lidar and can be done automatically.

**Page 23 Comments:**

 Line 1: If we read the reasons for the exclusion of measurement nights for the calculation of the calibration coefficient on page 12, there does not seem to be more or less nights used with the new method rather than with the « traditional » method. Please reconcile.
We had meant this sentence to imply that if a sonde shows significant disagreement with a lidar measurement on the order of 1 g/kg then it may be the practice of some researchers to not consider that flight for calibration. In this scenario, one might be able to use this calibration technique to compensate for the difference between the two instruments and thereby gain more calibration nights.

Line 5: The uncertainty has not been quantified for the « traditional » method (at least not indicated in the article) or in this article. This is something that is difficult to quantify, but as a result it is difficult to conclude that it has improved it. Indeed from a theoretical point of view it is, but in practice (and in this discussion) you do not relate it to your results. This uncertainty appears in your introduction and in the conclusion part but no word in the section focusing on the uncertainties.

Thank you for pointing it out that this is not said explicitly in the discussion or clearly in the article. To fix this, we have added the uncertainties of both the Traditional and Trajectory calibration values in Table 1. We have also added a discussion at the beginning of the uncertainties section discussing the uncertainties of the traditional method and how others have calculated it in the past.  We did discuss how Leblanc and others calculated the uncertainty by calculating the RMS of the calibration time series, but did not link it to the traditional method.

Line 10 « The method could... »: Yes but it should be presented as a limitation and not only as a perspective because it means that the « representation uncertainty » may not be as negligible as that.

We will discuss how it is a limitation to not include the vertical movement, as was mentioned in your comment for Page 22, Line 2. It is not obvious to us in general, outside of periods of intense convection, that typical vertical air motions over 30 min or less time scale would impact this analysis, especially for the trajectory method which is, by design, making sure the sonde and lidar sample the same region of air.

Line 12: I suggest starting here Sect. 7 Conclusions.
We would prefer to keep the discussion and conclusion together and not separate them.

Line 15: How do you assess that? Reference? Maybe Whiteman et al. (2011b)?
This sentence refers to Figure 8 in this manuscript which we neglected to include. We have now added the reference.

Line 15 « 50 % uptime over ten years »: Why is this only appearing now? 9 years of measurements -> 76 nights that can be calibrated -> 24 nights calibrated in practice, what about the other nights? Where is that 50%? Which calibration methodology for the entire database?
We did not include the operational components of the RALMO lidar because these have already been introduced in Dinoev et al. and Brocard et al.. These are being introduced now to discuss the future work that will be conducted using this study.

Line 24: To study trends in the UT/LS, please recall the total uncertainty associated with your profiles or refer to the article that assess the performances of the RALMO to measured water vapor in the UT/LS on a routine basis.
To date there are no studies assessing RALMO's measurements in the UTLS. RALMO was originally designed as a meteorological lidar with a focus on operational tropospheric measurements. The upcoming study in progress  will discuss RALMO's performance in the UTLS, develop a climatology, and look for trends in that climatology. This work is beyond the scope of the current manuscript which is developing one of the calibration methods which will be used to develop a climatology.

**Response to David Whiteman**

1. L1 – statement is made that "Lidars are well-suited for trend measurements in the upper troposphere and lower stratosphere, particularly for species such as water vapor."

      1. The measurement requirements for detection of trends in water vapor differ dramatically between the UT and the LS. Paragraph 16 in Whiteman et al, 2011b and the first several paragraphs of the discussion section of Whiteman et al, 2012 detail the argument that Raman water lidar is much better suited to trend detection in the upper troposphere than the lower stratosphere. Also, I might suggest that instead of just saying "Lidars" here, to specify "Carefully calibrated and quality-controlled Raman Lidars..."

      Thank you for pointing out the results from these two papers. We have changed the first sentence to include your suggested text.

      Regarding UT vs LS, this work is targeted at both current and future lidar systems, which we anticipate/hope will have the capability to measure routinely in the LS. However, we agree that reaching the UT is more feasible than reaching the LS.

2. L4 and beyond – the current technique is improved with respect to the traditional technique but no comparisons are done with respect to other "improved" techniques. It is my hope that we can address that in follow-up research.

We will be comparing this method with another calibration technique in a follow-up paper which discusses the processing of the RALMO data for trend analysis and combining the radiosonde technique with another independent long-term calibration method. We felt that it was not appropriate to put this comparison in this paper since the trajectory calibration can stand alone and we did not want to move focus away from the GRUAN radiosonde calibrations.

3. L5 – Whiteman et al (2006) is cited for a "track-sonde" technique that was used. It is worth noting, however, that the track-sonde technique as used in 2006 did not perform as well as the more simple variable temporal-spatial smoothing routine described in that same publication. More importantly, a significantly more sophisticated technique for performing radiosonde calibration was presented in Whiteman et al, 2012. It does not explicitly track the sonde but the geometrical similarity requirements imposed in that routine, I expect, achieve some of the same collocation benefit that is discussed in the authors' technique. These details should be mentioned.

We agree that using the method in 2012 should have a similar effect, however, we have separated the 2012 method into a separate category which is discussed in the new traditional section along with Dionisi et al. 2010. We have mentioned your 2012 paper and its goals in the new Section 3.2  . We have added the points you made about the tracking technique in the 2006 paper being superseded with the 2012 study to the discussion.

4. L29- "paralyzation" → "paralysis"

Thank you for catching this.

**Introduction Comments:**

5. Statement is made that "instruments with high spatial-temporal resolutions, such as lidars, are uniquely suited to long-term stratospheric and tropospheric water vapour studies". For lidars to provide a good signal-to-noise measurement in the UTLS requires significant temporal and spatial smoothing. So I do not agree that high spatial-temporal resolution measurements make lidars uniquely suited to long-term UTLS studies of water vapor since the temporal and spatial resolution must be degraded to achieve an acceptable S/N in the UTLS.

We agree that the sentence is a little misleading and to prevent confusion, we think it's best to just delete the sentence. Therefore we have removed it from the paper.

6. L6 - "of" → "from"

We have changed this as suggested.

7. Lines 7-9. Statement is made "Lidar measurements are particularly useful for creating statistically significant water vapor trends of the UT and LS region … " and Weatherhead, 1998 and Whiteman 2011b are used to support the claim. I don't believe that Weatherhead et al makes any statement about the suitability of lidar for this task. Also, as stated above, Whiteman et al 2011b expresses doubt that Raman lidar would be suitable for LS trend detection; a claim that is amplified in Whiteman et al, 2012. So I would suggest a statement such as "Carefully processed, stably calibrated Lidar measurements can be particularly useful for creating statistically significant water vapor trends in the UT region ..."

We had included Weatherhead 1998 because it discusses the uncertainty thresholds necessary to obtain statistically significant trends, which we thought was relevant here. However, it does not discuss lidars specifically. We have removed the reference.
We have incorporated your suggested comment in to the paper but we have not removed the LS from the paper for the reasons we have discussed previously.

8. Paragraph starting "Internal calibration techniques …
      1. reference is made to Venable et al, 2011 as an example of the white light technique, which is correct. The next sentences, however, refer to the limitation of using a single lamp and the need for multiple lamps or a scanning technique. This is confusing since Venable et al showed the utility of the scanned lamp technique so that work does not suffer from the limitations of the single lamp technique as implied by the current discussion. I suggest revising the paragraph so that the first reference cited is one that makes use of a single lamp.

Thank you for pointing this out. Venable et al.2011 was indeed not the appropriate first reference. In fact, the first reference should have been Leblanc et al. 2008. We have made the change and do not think the paragraph needs to be changed as it now matches the reference.

2. Later in the same paragraph it is stated that the uncertainty in the knowledge of the ratio of the Raman cross sections is 10% from Penny and Lapp, 1976. The work of Avila et al, 2004 and Venable, 2011 however point toward an uncertainty of this cross section ratio closer to 5%. To support this, Fernandez-Sanchez (the lead of the group in which Avila did his work) has privately communicated with me that 5% is his assessment of the absolute accuracy of their water vapor cross sections and given that the nitrogen cross section uncertainty is in the range of 1-2%, this is consistent with a claim of ~5% uncertainty in the cross section ratio. Venable et al has some text concerning this. So I believe that an assignment of 5% to the uncertainty of the Raman water vapor/nitrogen cross section ratio is justifiable. But at least this more recent work makes the 10% Penny and Lapp uncertainty from 1976 no longer appropriate.

Thank you for noticing that this was missed in the original literature review. We will change the numbers accordingly and cite Avila 2004 and Venable 2011.

The new sentence is as follows:
"The limiting factor in the white lamp calibration technique is the degree to which we know the molecular cross-sections which are known to have uncertainties on the order of 5% (Avila et al. 2004, Venable et al. 2011)"

9. P4, L19. Immler et al, 2010 is used as a reference for the GRUAN RS92 correction technique. Immler et al discusses error characterization in general but does not present the RS92 correction technique. The Dirksen et al, 2014 reference is more appropriate.

We agree with you and will remove the Immler reference.

10. P9, Lines 8-9. Statement is made that "we do not correct for aerosols as they are considered to have a very small contribution to the overall mixing ratio". I take this to mean that the differential transmission due to aerosols is not accounted for. In the 1992 reference that is cited to support the authors' statement, it is shown that with aerosol optical thickness at 355nm of 1.0 the calculated mixing ratio would change by ~4% as compared to a pure Rayleigh atmosphere. Indeed, AOT of 1.0 is quite a turbid atmosphere but this result also implies that AOT of 0.25 would yield a 1% change in mixing ratio. One's first impression might be that 1% uncertainties are small enough to neglect (I do not agree). But neglecting aerosol differential transmission does not introduce a random uncertainty but rather a systematic one. And surely in a paper that has long term trend detection as a stated goal, elimination of systematic uncertainties that can be up to 4% must really be done. So I strongly encourage that the authors address this deficiency. Note that it is not necessary to calculate aerosol extinction directly from the lidar data to adequately make this correction. One can instead use collocated aerosol optical thickness measurements along with a reasonable estimate of the height of the boundary layer to

develop a simple model for calculating the aerosol differential transmission such that the residual uncertainty in the aerosol differential transmission correction is well below 1% even under turbid conditions. This is the technique that we generally use to handle this tricky part of the Raman water vapor lidar analysis.

We agree that not including aerosols does induce a systematic bias and should be taken care of. We had not done this originally because we do not currently have a lidar extinction product for RALMO, nor was there a co-located instrument capable of measuring AOD during nighttime at that time.We do, however, have an aerosol scattering ratio product and therefore we did not think it was appropriate to use a daytime AOD measurement due to the variability of aerosols over the course of a night. RALMO's total backscatter ratio product is calculated by taking the ratio of the elastic and the sum of the pure rotational raman signals at 355 nm. Therefore, we have used this product and assumed lidar ratios and an angstrom exponent using an angstrom exponent time series in order to estimate aerosol extinction profiles.
Similarly to the method followed in Sica and Haefele et al. 2016, we calculate the extinction profile using the following equation:

$$\alpha_{aer}(z) \ = \ LR(z) * (\beta_{mol}(z) * (BSR(z) \ - 1))$$

Where $\alpha_{aer}(z)$ is the extinction profile which changes with altitude *z*, *LR(z)* is a lidar ratio profile, $\beta_{mol}(z)$ is the molecular backscatter profile taken from the NCEP model, and the *BSR(z)* is the total backscatter ratio profile. The lidar ratio profile is a step function with a constant value in the boundary layer and another constant value for the free troposphere. The height of the boundary layer is estimated using the backscatter ratio profile. We have assumed lidar ratios of 20 for the free troposphere and 50 for the boundary layer using climatological values from the Payerne station. The transmissions for each channel are calculated using the equation below:

$$T_{aer,x}(z) \ = \ exp\{ \ - \ (\lambda_x/\lambda_0)^A * \int \alpha_{aer}(z)dz \ \}$$

Where $T_{aer,x}(z)$ is the aerosol transmission profile for a given molecule *x* (e.g. N2 or H20), $\lambda_x$ is the wavelength for a particular channel, $\lambda_0$ is the reference extinction profile which in this case is 354.7 nm for the elastic channel, and *A* is the angstrom exponent which is assumed constant with altitude. The Ångstrom exponent, *A*, is assumed constant with altitude. The Ångstrom exponent is measured during the daytime using the co-located Precision Filter Radiometer (PFR). We have no measurements of nighttime angstrom exponents, therefore we are forced to use the daytime values, unlike the case of the optical depth calculation where we could use the lidar backscatter ratio. The Ångstrom exponent is not measured daily as it requires stable, cloud-free conditions to get an accurate calculation. Since it is not always available, we fit the sum of a 6 and 12 month sinusoid to the angstrom exponent time series over measurements from 01 January 2012 until 31 December 2015, with 2014 removed due to a faulty sensor. The fitted sinusoid was then used as the values for the angstrom exponents. The standard deviation of the residuals was ± 0.34 and was used as the uncertainty for the angstrom exponents

We have calculated the uncertainty induced by our assumptions by using the uncertainty equation introduced in the paper. We assume the worst case scenario of 100% uncertainty in the extinction profile calculation and the standard deviation of the Angstrom exponent residuals for the uncertainty calculations.

We found that the uncertainty in the calibration constant due to the uncertainty in the extinction profile was much less than 0.01% for all cases. The uncertainty in the calibration constant due to the uncertainty in the angstrom exponent was only an order of magnitude higher and on average of 0.4% +/- 0.5%. Therefore, the radiosonde remains the largest calibration source.

While the uncertainty in the calibration constant is low due to our assumptions, the calibration constants did change by an average of 2% when adding in the differential aerosol transmission to the calibration. On nights with a strong boundary layer (2 cases), we did see a change in 6 - 8% in the calibration constant. The results we get seem to be consistent with the results of Whiteman 2003.

11. P10, Lines 16-17. "we require … to be correlated to greater than 90%." I assume that by this the authors mean than the correlation coefficient of the linear regression is 0.9 or greater. If so, please restate in terms of correlation coefficient to avoid confusion.

You assume correctly, we will change the wording as you suggest so that it is clearer.
We have changed the sentence as follows:
" To ensure that the …. We require the resulting uncalibrated lidar and radiosonde mixing ratio profiles to have a correlation coefficient which minimizes the variance of the fit's residuals and must be above 0.75". We have changed the calculation of the slope to one similar to what you discuss in Whiteman 2012 and have therefore changed the sentence accordingly.
12. P10, last line. I do not see that the results of Aug 8, 2012 showing a 5% offset. Is this something that is apparent from the Table? If not, please clarify that this information cannot be gleaned from the Table.

Thank you for pointing out that this should not be referring to Table 1 and is in fact missing a reference to Figure 6. We have changed the wording in this section of the paper to no longer discuss 5% offset and instead discuss the differences in the features measured by the lidar and the radiosonde. We believe it is more appropriate to discuss the features individually instead of discussing overall biases. We have not included the text here since the entire section has been reorganized and we would refer you to the re-worked Section 4 in the revised paper.

13. P15, lines 5-10. A qualitative comparison of results of homogeneous and heterogeneous cases is made but the actual standard deviations, for example, are not given. From Table 1, it seems that for the homogeneous cases, the standard deviation of the calibration constants derived using the traditional technique is less than that of the trajectory technique. For the heterogeneous cases, the trajectory technique gives slightly smaller standard deviation as

stated. I do suggest giving the actual standard deviation values in the table and discussing the significance of these standard deviation differences since they seem to be rather small.

You are correct that the standard deviations are not given here, they were originally put in the summary which was not the appropriate place. The discussion of the standard deviations has been moved to where we introduce the Table and is much clearer. It is true, that when considering the entire population of both the standard deviations are small and therefore the differences between the two would not be statistically significant. However, when the two extraneous cases were removed from the calculation then the differences between homogeneous and heterogeneous nights become more apparent. Please refer to the new text in Section 4.

14. P 17. "Lidar Calibration Uncertainties for Trajectory and Traditional Methods". Three sources of uncertainty are listed: lidar statistical uncertainty, GRUAN radiosonde uncertainty, dead time uncertainty. The "usual" way that radiosondes have been used in a Raman lidar calibration effort (e.g the MOHAVE, AWEX, IHOP, PECAN field campaigns) is to assume that the lidar calibration value has been constant over the duration of a field campaign and that differences in calculated calibration constants relate to statistical uncertainties, collocation uncertainty, etc. Following this procedure, a single calibration constant is determined from all the radiosonde comparisons in a field campaign and that calibration value is used for some period of time until another large intercomparison effort with multiple radiosondes is performed. This is the technique outlined in discussions of the hybrid technique, for example, and in such cases there is another very significant component of the calibration uncertainty, which I call the calibration transfer uncertainty, that is not listed here by the authors since it does not pertain to what they are doing (but it is very significant in the overall discussion of lidar calibration). This systematic uncertainty can be taken to be the standard deviation of the individual calibration constants used to determine the mean calibration constant that is finally used in a field campaign type of study. I understand that the authors are doing things differently and are re-calibrating the lidar with every available radiosonde. In fact, the authors approach is much preferred from the standpoint of developing a time series for trend detection because each time a different calibration constant is used for the lidar, a step-change systematic uncertainty is introduced into the time series. This is inevitable. So to decrease the influence of these systematic step-changes, frequent calibrations are needed so as to make these systematic uncertainties, in effect, components of the random uncertainty budget in the time series. The authors refer to some of this later in the paper but here is where it should be introduced. Thus to recalibrate the lidar as frequently as possible serves to transform a component of the systematic uncertainty budget (where it can really destroy a trend calculation) into a random uncertainty. Note that the DOE/ARM Raman lidar is recalibrated with respect to microwave radiometer every three days achieving this randomization of calibration constant. However, campaign mode calibration efforts as described in the MOHAVE papers do not achieve this. So … I suggest that the authors clarify this. There is discussion in Whiteman et al, 2011b about the need to randomize components of the systematic uncertainty budget to improve time series for trend detection.

This comment is very helpful, and while we tried to discuss exactly this concern towards the end of the uncertainty section, we did not stress how our method differs from typical practice enough or its implications for trend analysis. We agree that it would be better to discuss the typical methods of calculating calibration uncertainty at the beginning of the section instead of the end and would lead to a natural transition to our preferred method. Therefore, we will remove the first few sentences in that last paragraph and add the following text to the beginning of the uncertainty calculation section :

> The standard practice for determining the uncertainty of the calibration constant has been to conduct extensive calibration campaigns and assume that the calibration value does not change over the campaign period and then measure the variability of the constant \citep{Ferrare1995,Turner2002,Whiteman2006,Leblanc2008,David2017}. The variability of the constant is then assumed to be the uncertainty and the calibration constant is not changed until the next campaign when multiple radiosondes are available for calibration. The assumption that the calibration constant does not change over long periods of time introduces another source of uncertainty into water vapour measurements, which is often unknown until the next calibration period. Uncertainties calculated in this way vary between 4 and 5\% of the calibration constant during the calibration period, but do not account for the individual sources of contribution nor do they typically account for the variability in the calibration constant beyond the campaign period.

> Accounting for drift or changes in the calibration constant is extremely important for long term trend analyses, since such a drift/change could easily be larger than the uncertainty of the calculated trend and would render the analysis invalid if it was not considered \citep{Whiteman2011b}. Many systems have now taken this into account by conducting daily or semi-daily calibration measurements either using an internal, hybrid, or external calibration. Taking more frequent calibration measurements with uncertainties calculated for each calibration then turns a systematic uncertainty component of a trend analysis into a random uncertainty component, particularly if the uncertainty of the calibration constant is recalculated with each calibration.

> While it is possible to calculate the uncertainty budget of a calibration constant based on the lidar's measurements and components, often the largest unknown uncertainty is the uncertainty of the reference instrument \citep{Leblanc2008}. It was not until recently that such detailed information was available for radiosondes. The GRUAN radiosondes are the first radiosondes to have a published uncertainty budget as a function of altitude for each measurement \citep{Dirksen2014}. By using the GRUAN radiosondes, we are now able to calculate the uncertainty in the calibration constant due to the radiosonde's uncertainties.

14a. It's also in this section where the uniqueness of the use of GRUAN sondes for this calibration task should be highlighted. This is the first time, to my knowledge, that linear

regressions of radiosonde/lidar data have been performed with weights that make use of carefully characterized radiosonde uncertainties. This is significant.

You are certainly correct that these points have not been stated with enough clarity in the paper and need to be more heavily emphasized. We believe the last paragraph in the response to the previous comment makes the distinction more clearly.

15. P18, line 21. The term "scan" is used here and earlier but it is not clear what "scan" means. Please go back in the paper and define how you use this term the first time it appears.

Thank you for pointing this out - scan seems to be a colloquial term within this research group. A scan refers to a 1 minute or 1800 shot raw measurement profile. We have added this definition in the first instance where "scan" appears.

16. P19, line 11 … I chuckled when I read that eq 5 is a simplified version of eq 4. Upon inspection eq 5 is about twice as long as eq 4 so does not appear much simplified. You might just say "With these assumptions, eq 4 becomes ..."

Perhaps "reduced form" might have been the better wording. We will change it as you suggest.

17. P20, line 3. This is where it becomes clear that you are recalibrating the lidar with each radiosonde. You also make the point that this is different than for field campaigns as in the Leblanc and Dionisi references. Good. Now, as mentioned earlier, you can make the point that this approach helps to randomize a component of the systematic uncertainty making the resulting time series more appropriate for trend detection.

Thank you, we did try to make this distinction albeit not very well. We have added new text to the beginning of the uncertainty section which now explains the difference between our approach and the standard field campaign approach.

18. P20, lines 13-16. I've already commented that ignoring aerosol differential transmission neglects a systematic bias which is a strong concern and goes against the prescription of the BIPM/GUM where all known systematic biases should be corrected (see quote in Whiteman et al, 2012 or go to the GUM itself). Also, though, the way that the sentence reads it is not clear what 5% refers to. Finally I would say that one should perform the calibration of the lidar data in the same way that it is analyzed for trend detection and one would not want to neglect aerosol differential transmission when trying to create trend-detection quality time series of water vapor measurements. So aerosols really do need to be accounted for in this analysis and in the full analysis of the lidar data.

We agree that aerosols should have been included. We have answered this concern after your previous comment and have tried, to the best of our ability, to include them and account for the uncertainty in our assumptions.

19. P23, lines 3-5. "frequent and accurate lidar calibrations are critical for detecting water vapor trends …" The earlier discussion of randomizing components of the systematic uncertainty budget is the main argument for why this statement is true so you should add a citation here. But I need to repeat that the measurement challenge in the LS is very different than in the UT so that your statement really only applies to the UT. BTW, these are the reasons why trend detection in the UT is so much easier with Raman lidar than in the LS:

     1. The natural variability of water vapor in the UT is much higher than in the LS. So the relatively large random uncertainty of Raman water vapor lidar does not deteriorate the time to detect trend by a large fraction in the UT.

     2. On the other hand, the natural variability of water vapor in the LS is very low and the random uncertainty of lidar measurements is much, much larger in the LS since it is farther away than the UT and water vapor concentrations are so small in the LS. So the random uncertainty of Raman lidar measurements in the LS typically will swamp the uncertainty budget and greatly extend the time to detect trend using the methodology of Weatherhead et al, 1998.

     3. According to the modeling cited in Whiteman et al, 2011b the anticipated trends in LS water vapor are smaller than those in the UT making trends more difficult to quantify in the LS.

     4. Because of much lower S/N lidar measurements in the LS, small sources of systematic bias in the lidar measurements can more easily corrupt the time series. The larger signals in the UT are more resistant to such unknown sources of bias.

     We will add a citation here for Whiteman 2011 b.

     We agree that calculating trends in the UT is undeniably *easier* than LS, but we would argue that for future lidars, or even for the latest improvements, that measurements in the LS and trends in the LS should be possible. As stated previously, we would prefer not to limit the discussion to only the UT since we hope that this paper will serve as a reference for future lidars which may be built specifically for the purpose of studying the LS and will have the ability to detect trends at those heights.

20. P23, last paragraph. At the end of the study a conclusion is that the trajectory method does not produce statistically different calibration values than the trajectory method. This does not argue strongly for the technique presented here. I would suggest looking for ways to decrease the standard deviation of the calculated calibration values. In Whiteman et al, 2012 we found that by using the adaptive technique described there we could reduce the variability of the calculated calibration values by requiring that the correlation coefficient between the lidar and radiosonde profile segments be higher. You might try adding that into your algorithm since, as I understand, you already require $R2 > 0.9$. The point here is that it should be a goal of this work to achieve a more stable calibration constant than that achieved with the traditional technique.

We agree that we haven't written a strong enough conclusion here and that the goal should be to achieve a stable calibration constant. The conclusion will be revised to make the following important points:

1. The trajectory method does improve the differences between the radiosonde and lidar, particularly on the heterogeneous nights.
2. This is the first paper to use the GRUAN sondes for a nightly calibration uncertainty analysis.
3. The height dependent uncertainties reported by GRUAN allow us to calculate the uncertainty of each calibration constant.
4. This method should allow one to do more frequent calibrations using radiosondes launched farther away from the observatory which in turn will help randomize the calibration uncertainty in any trend analysis.

We have done as you suggested and implemented the variable correlation method you used in Whiteman et al. 2012. With one difference where we do choose the calibration constant directly from the slope. This is because we propagate our uncertainties directly from the least squares fitting equation. We have changed the paper accordingly and updated it with the new calibration values. Implementing the variable correlation did not significantly change the final value of the calibration constant - at most there was a shift in 0.5% of the calibration value. However, adding in the aerosol component does seem to have decreased the variability of the constant.

[revised manuscript text omitted]

---

## Author Response (AR2)

**Response to Referee # 1**

**General Comments:**

Comment 1:
It is difficult to conclude with this section which value should be statistically retained and this "20%" in the abstract seems to be overstated. On the other hand, a 4% uncertainty (p1, l 23) is presented for the trajectory technique but there is no uncertainty stated for the traditional method for comparison. While I have noted that the authors have made it clear to me that the term "more accurate" does not refer to the comparative uncertainties of the two methods, this conclusion in the abstract appears directly after quantifying the uncertainty of the trajectory (p1, l 23). This might confuse the reader.

While only one example is shown, we have other examples (not shown) in our study which exhibit similarly large features. The average difference is roughly 10% therefore we will change the abstract to say "an average of 10% difference between the radiosonde and lidar profiles."

Comment 2:
Therefore, it seems that the authors overstate a little bit the added value of this methodology in the abstract and when it comes to concluding. I think that this could be reviewed by the authors. That being said, all the quantified data are present in the article, it could also be up to the readers to evaluate their contributions. But the authors take the risk that the reader will share this view.

It is up to the reader to decide the value of our methodology based on their site. The differences between the trajectory technique and traditional for RALMO were not statistically different. However, a different site which doesn't have co-located radiosondes may find it makes a larger difference. We believe that we have been fair in discussing the value of the method in the conclusion, but would be willing to add a sentence to this effect in the abstract as well.

We have changed the last sentence in the abstract to:

This trajectory method showed small improvements for RALMO's calibration but would be more useful for stations in different climatological regions, or where non-co-located radiosondes are the only available calibration source.

**Specific comments:**

Page 1 Abstract Line 10: Could you clarify the term "frontal boundaries"?

We have changed "frontal boundaries" to "fronts" to be consistent with the AMS Meteorology Glossary.

Page 1 Line 19: You should specify what this "up to 20%" corresponds to so that this value can be found in the text.

We will now change this to say " an average of 10%" which is more consistent with the figures. However, we have already stated that it is the difference between the radiosonde and lidar profiles which indicated when the water vapour field is inhomogeneous. This is sufficient clarification to guide the reader to the section which includes the discussion on homogeneous vs heterogeneous nights.

Page 2 Line 27: I suggest replacing the word "GPS" with "GNSS".

We will change this as suggested.

Page 3 Lines 4 and 5: Do not the references only refer to RS92S? In this case, this should be specified.

You are correct, they only studied the uncertainties of the RS92 sonde. We will change this sentence to:

"Measurement uncertainties for the Vaisala RS92 relative humidity measurements vary between 5 to 15% uncertainty depending on the time of day (Miloshevich 2009, Dirksen 2014)."

Page 3 Lines 22 "travelled" and 23 "traveled": Please standardize.

Thank you, we have changed this to "traveled".

Page 4 Line 8: Please add a reference.

We will add the reference for Dirksen 2014 as this is the most relevant to our calibration. The new text is:

The GRUAN sondes represent the best characterized sonde measurements available in terms of calibration and uncertainty budget (Dirksen 2014).

Page 5 Line 16 "a fog,clouds": A space is missing.

Thank you we have fixed this.

Page 8 Lines 6 and 7 "The radiosonde relative humidity [… ] 2014).": This should appear in Sect. 2.1.

It is in section 2.1 - the Hyland and Wexler 1983 formulae are used for the standard WMO conversion. However, the language is not consistent and is not clear. We will change this to say that the conversion is done using the Hyland and Wexler 1983 formulae instead so as not to confuse the reader.

Page 14 Line 11 "30 minute": I would have written 30-minute.

We will change this to 30 min to be consistent with the rest of the manuscript.

Page 15  Line 4 "will produce": Did not you mean "should"?

We have changed this to "should".

Page 15 Line 4 "homogeneous nights":

In this case it is not correct to put a colon after homogeneous nights because the sentence begins with "While" and uses a subordinate conjunction, so we will leave the comma.

Figure 5 shows the 2012-07-27 time series. According to Table 1, this night corresponds to a heterogeneous night. Please reconcile.

Apologies that this was confusing - we used this night only as an example to illustrate how the methods can choose different scans and not as an example homogeneous night. We understand that this could be confusing for the reader. We will change the figure to be a homogeneous night to avoid confusion. The only wording in the caption that will change is the date of the radiosonde launch.

[Figure]

Figure 5: Lidar water vapour mixing ratio measurements on 2013-06-05 00:00 UTC. The time axis is measured relative to the radiosonde launch. The traditional method uses all scans between the two red dashed lines. The trajectory method uses all measurements between the magenta dots. The white "x" markers show the height of the radiosonde with time.

Page 16 Lines 7 to 9 "is reduced to oscillate around 0 %": I do not understand what you mean. Please clarify.

We agree that this sentence was poorly worded and does not agree with the figure. The sentence will be changed as follows:

On 24 April 2013, there is a large and increasing difference between the radiosonde and lidar measurements above 4 km, with a slope of roughly 5% difference per kilometer altitude. This increasing difference between the radiosonde and lidar measurements is reduced in the trajectory method to a constant bias of 5%. However, the variability of the difference between the sonde and lidar profile is larger than for the traditional method.

Page 16 Line 14: Please refer to Table 1 after "1 %".

We have changed this as suggested.

Page 19 Line 6 "less variability": Please quantify

We will change this sentence to say " …. 10 - 15% less variable than the traditional method."

Page 19 Line 7: I suggest adding a small conclusion on this comparative part (Sect. 5). The improvements shown refers to the altitude range 2.5 - 4 km and highlight the values that quantify it.

We have added the following text:
In summary, the trajectory method shows a similar absolute bias to the radiosonde but with the opposite sign compared to the traditional method. The variability of the differences between the lidar and the radiosonde is 10 - 15\% smaller in the trajectory method than it is in the traditional method between 2 and 4.5\,km altitude, but is the same below 2\,km and above 4.5\,km.

Page 21 Line 10 "mixing ratio ,": There is a space to delete.

Thank you for catching this.

Page 21 Line 20 ".1 and .3 K": I think that some zeros are missing. Please correct it.

You are correct, it should be 0.1  and 0.3 K.

Page 22 Lines 6 and 7: There are '' and ".Please standardize.

Thank you we have fixed the formatting.

Page 23 Line 15: The average total accuracy is stated to be 4 % for both method. The dominant uncertainty is the radiosonde uncertainty of 4%. Table 2 shows that there are other uncertainties. Are you sure that the total is 4%.

You are correct, that is confusing. We have replaced the above with a new sentence as follows.

        The average total uncertainty of both the trajectory and traditional calibration constant is 4.5\% with the majority from the radiosonde's contribution.

Page 24 Lines 26 to 28: Why the discussion on these two uncertainties is at this point when the other uncertainties are discussed in the list that begins l 4 on p25?

We can include them in the summary list if that is preferable.

Page 25 Line 2 "by 5-10 %": Would not it be better to use this 5-10% value in the abstract instead of "up to 20 %"?

We will change this to say an average of 10% as mentioned previously.

Page 25 Line 8: You should precise the average difference for the traditional method.

Our apologies that this sentence was not clear - it is stating the average difference between the traditional and trajectory calibration constants.

The new sentence will be as follows:

For these cases the calibration constants calculated by the trajectory method resulted in an average of \textcolor{black}{$1.92 \pm 0.93\%$} difference with the traditional method calibration constants.

Page 25 Line 10: This should be precised that it is between 2 and 4 km.

We will change this to say between 2 and 4 km.

The trajectory method produces a smaller average bias between the radiosonde and the lidar than the traditional method between 2 and 4\,km \textcolor{black}{(Fig.~\ref{fig:both_pdiff})}. Adding points above 4\,km does not change the calibration constant significantly as the photon counting uncertainty becomes large at these altitudes.

Page 25 Line 12: In Sect. 6, 4.5% corresponds to the standard deviation (p24, l 9) and 4% to the total uncertainty on calibration (p23, l 15). In this summary and Table 1, it is the uncertainty on lidar photo counting and radiosonde mixing ratios whereas the authors find a total uncertainty of 4% (p23, l 7). Please reconcile.

The above is addressed in our previous comment. The average total uncertainty is 4.5%. The uncertainty of 4% did not include the uncertainty contribution from the Angstrom exponent.

Page 25 Line 20: I have the impression that these values appear for the first time in Sect. 7, why were they not brought into the dedicated part, namely Sect. 6?

These values were included in the last paragraph of Section 6, but described as the standard deviation of the residuals. RMS is sometimes used interchangeably with standard deviation, however, it is not best practice to do so as they mean different things. The wording here has now been changed to maintain consistency with Section 6.

The new sentence is as follows:
The uncertainties calculated by the standard deviation of the trajectory and traditional method time series were both 4.5%, which is consistent with the total uncertainties calculated using Eq.~\ref{eqn:leblanc}.

Page 26 Table 2: You should put the total uncertainty. You could differentiate the average uncertainties according to the calibration method.

We can add the total uncertainty. The uncertainty ranges are the same for both methods, since the only difference in the methods is what lidar data is used. We will make sure this is clear by stating this in the table caption and the text.

Page 27 Lines 12 to 14: You say that the trajectory method "removes the representative uncertainty" and then that this uncertainty is considered "to be small". This means that it is not deleted. Please reconcile.

You are correct that these two sentences are contradictory. The representative uncertainty should be greatly reduced. The first sentence will be removed and the second will remain.

Page 27 Line 19 & line 24: Be careful you have inverted the letters from GRUAN to "GRAUN".

Thank you for catching this mistake.

Page 28 Line 5: This should be precised that it is between 2.5 and 4 km and not only "below 4 km".

The improved region was between 2 and 4 km; we will change that.

Page 28 Line 12: I am puzzled by the term "reference instrument". For me, the reference instrument would be a third instrument considered as the reference that would be used to validate this method. The profile calibrated with RS92S would then be compared to the profile measured by this reference instrument. I suggest replacing this term with another that would mean: the instrument used to calibrate.

We have defined the term "reference instrument" throughout the paper and in the abstract and therefore would prefer not to change the terminology for the entire paper.

Response to Dr. Whiteman (June 2019)

General Comment 1:
 "As contained in the detailed comments, the authors still are using text that cannot at the current time be supported by available references when they discuss the ability of Raman water vapor lidar to detect trends in the UTLS. They state that such instruments are "well-suited for trend measurements in the upper troposphere and lower stratosphere" and that "lidar measurements are particularly useful for creating statistically significant water vapour trends of the Upper Troposphere and Lower Stratospheric region". Such language seems to imply that studies have been performed using Raman lidar data that have revealed significant trends in water vapor. It is my sincere hope that such studies will be successful (and I know the authors are working on that currently) but to my knowledge there is presently no such published account. Therefore I think it is necessary to modify the language they use in this context. Perhaps the authors could mention that both NDAAC and GRUAN have designated Raman lidars as potential candidates for trend studies and that for such studies improved calibration techniques are needed. That then can be the context for this paper as they also look toward performing such studies themselves."

You are correct that there is no text to support these assertions. We will remove the mention of the lower stratosphere. We will add the sentence that you have suggested to the introduction instead.

Text added:
"Raman lidars have been designated as potential candidates for trend studies by NDACC and GRUAN; however, for such studies improved calibration techniques are needed as well as careful consideration of the calibration uncertainties."

General Comment 2:
"The authors find that the trajectory technique works better than the traditional below 4 km but not so above 4 km. They state that the reason perhaps for the result above 4 km is the reduced number of comparison points. What is happening, it seems, is that the radiosonde is floating farther away and thus leaving the 3 km radius column around the lidar. One wonders, therefore, if an acceptance cone instead of a column might make more sense. The other tests of similarity of profiles would still be in effect insuring that the profiles come from the same atmosphere. Having a cone could increase the number of comparisons and improve the statistics above 4 km. I would like the authors to add some small discussion for why a column was chosen versus a cone."

We did try  a cone early on in our study, but it was not used. We had originally used a cone whose size was determined by the horizontal correlation length of water vapor using the wind speeds in each layer. Unfortunately this method didn't work because the winds were too variable and changed rapidly on the order of meters per second. Smoothing the cone did not help. We also tried to use a cone which was the same width as the lidar field of view, which was

too small. We did not test a cone with arbitrary lengths, however, we did test cylinders of various radii. Increasing the radii of the column above 4 km did not increase the number of comparisons by a significant amount and did not change the shape of the original 3 km profile. However, it would be a reasonable avenue to pursue in further studies and might improve the signal-to-noise ratios at those heights. However, as the lidar statistical uncertainty is such a small component of the calibration uncertainty, it would have a small effect on the overall uncertainty, and would likely not produce significantly different results for RALMO. We would encourage other sites to test a cone to see if it provides similar or better results for them.

We propose to add the following text to the Discussion Section:

Future studies using this technique could possibly show an improvement using a cone instead of a cylinder for the homogeneous lidar region. The trajectory method works better between 2 and 4 km but worse than the traditional method above. Using a cone could increase the integration time for the higher altitudes where less trajectories tend to intersect the homogeneous lidar region cylinder, improving the comparison. We initially tested a cone, and set its radius using the horizontal correlation lengths of water vapour using the wind speeds measured by the radiosonde. However, this scheme produced a cone which was too variable in size to be useful for calibration. We explored using a cone the same size as the lidar's field-of-view, but it allowed so few trajectories to intersect the cone that no calibration could be performed. Varying the cylinder size did not significantly change the shape of the profile above 4 km, but reduced the noise. Reducing the noise of the profile is important; however, as the lidar measurements only contribute 0.01% of the uncertainty to the calibration constant, it would not provide much benefit for RALMO. Using a cone could prove advantageous for other sites which exhibit higher wind speeds than Payerne. However, for sites which use radiosondes which are not co-located with the lidar, it would not be as beneficial as a cylinder. We would encourage others who might implement this method to try a cone to see if it significantly improves their results.

Detailed Comments:

Comment 1:
First line of abstract states "Carefully calibrated and quality-controlled Raman lidars are well-suited for trend measurements in the upper troposphere and lower stratosphere, particularly for species such as water vapour." This is a modified version of the original as shown by the blue text but does not address the primary concern. To my knowledge Raman water vapor lidar measurements have never been demonstrated to produce a dataset from which a statistically significant trend in water vapor can be discerned. I know that the authors are pursuing this goal and I sincerely hope they are successful. But, for now, no such demonstration has been made. Furthermore, it has been demonstrated that revealing a trend in lower stratospheric water vapor concentration using Raman lidar measurements would be significantly more difficult than for the upper troposphere. So, since trend detection has not been

accomplished with Raman water vapor lidar in either the UT or the LS and it will clearly be more difficult to accomplish in the LS this opening sentence really must be revised or removed.

We agree with your statement and will change the sentence as follows:

"Raman lidars have been designated as potential candidates for trend studies by NDACC and GRUAN; however, for such studies improved calibration techniques are needed as well as careful consideration of the calibration uncertainties."

Comment 2:
Lines 11-13 state "Lidar measurements are particularly useful for creating statistically significant water vapour trends of the Upper Tropospheric and Lower Stratospheric (UTLS) region, as they are able to take make long term and frequent measurements (Weatherhead et al., 1998; Whiteman et al., 2011b) (Whiteman et al., 2011b)."

1. Again, this statement is not justified by any work that I am aware of. Note that the 2011b reference cited states that Raman water vapor lidar "could contribute usefully to monitoring trends in upper tropospheric water vapor although it will be necessary to have stringent quality control procedures in place to guard against errors [Whiteman et al., 2006; Leblanc et al., 2008; Boers and van Meijgaard, 2009] in upper tropospheric Raman lidar water vapor measurements. It should be noted in this context, though, that a large random error could mask the presence of a small systematic error making its detection more difficult." So I do not believe that the cited reference justifies the sentence that it refers to.

We will remove the mention of Lower Stratospheric and change the sentence to:
"Lidar measurements are particularly useful for creating statistically significant water vapour trends throughout the troposphere, as they are able to make long term and frequent measurements."

2. For both items 1 and 2 in these comments, I suggest that the authors adopt a different perspective on trend detection using Raman lidar. They can instead talk about the potential for doing so and that their future work is aimed in exactly that direction. The current work, therefore, is helping to lay the ground work for future efforts in trend detection using Raman water vapor lidar. If the text in the manuscript adopted this kind of tone, I think there would be no issues with reviewers.

We agree with you and will change the tone of the manuscript. We will change the instances which focus on the UTLS and use "throughout the troposphere" when appropriate and avoid stating that Raman lidars are suitable for LS trend studies.

3. I suggest that upper troposphere and lower stratosphere not be capitalized although with the abbreviation UTLS it is customary to do so.

We agree and have removed the capitalized letters in the response.

Comment 3:
P 6, line 6 – "...backgrounds above 0.01 photon counts." Unit: Per bin? Per second? Those units are used earlier and should be repeated here.

The units are photon counts/bin/s - the units will be added.

Comment 4:
Section 3, Traditional method. Line 12: calibration calibrating

Thank you we will change this.

Comment 5:
Section 3, line 20. Presumably the saturation correction comes before the background subtraction so I would reverse the order of those processes in this sentence.

You are correct - we will reverse them.

Comment 6:
P. 7, line 23. "...exponent is small." Can you quantify this? It is important to convey that fact that making such approximations yields a useful correction to the data without greatly increasing the uncertainty budget.

Yes, we can add the result here. We had originally left it later in the paper since it was a result, but we agree that it would be helpful here.

Comment 7:
With respect to Eq 4, the 2003 reference is more suitable than the 1992 reference due to the presence of the temperature dependent terms.

We will change the reference, thank you.

Comment 8:
P 9, L 25 "… each of the scans is ..."

Thank you for catching the mistake.

Comment 9:
P10, Line 8-10. "First, we use the latitude and longitude of the radiosonde, as calculated by the on-board GPS system, as the initial position for air parcel tracking. The air parcel is then assumed to have traveled in a straight line until the time it was measured by the radiosonde." I'm confused by this sentence, should the last word here be "lidar"? Or is the lat/lon referred to that of the sonde upon launch and not at altitude?

The word should be radiosonde since we are taking *back* trajectories. The lat/lon refers to the position of the sonde upon launch. We will reword this as:

The air parcel is then tracked backwards from the radiosonde position and is assumed to have traveled in a straight line.

Comments 10-15: Grammar changes

Comments 10 - 15 have been changed as suggested.

Comment 16
Fig 8 Suggest binning at lower vertical resolution, perhaps 100m or more, to reduce some of high frequency noise.

We have smoothed the results with a running average of 20 bins (75 m) and added them to the figure as dashed lines to the original that way both the raw results and the smoothed are visible for the reader.

[Figure]

Comment 17
P23 line 9. You could add something like " which is to be expected since the traditional technique tends to sample the same volume of air as the trajectory in the lower altitudes as shown in Fig 5."

We will add the suggested sentence thank you.

Comments 18 -20:
We have changed these as suggested.

[revised manuscript text omitted]